# Principled Long-Tailed Generative Modeling via Diffusion Models

**Pranoy Das, Kexin Fu, Abolfazl Hashemi, Vijay Gupta**
School of Electrical and Computer Engineering, Purdue University, W. Lafayette, IN, 47906
{das211,fu448,abolfazl,gupta869}@purdue.edu

## Abstract

Deep generative models, particularly diffusion models, have achieved remarkable success but face significant challenges when trained on real-world, long-tailed datasets- where few "head" classes dominate and many "tail" classes are underrepresented. This paper develops a theoretical framework for long-tailed learning via diffusion models through the lens of deep mutual learning. We introduce a novel regularized training objective that combines the standard diffusion loss with a mutual learning term, enabling balanced performance across all class labels, including the underrepresented tails. Our approach to learn via the proposed regularized objective is to formulate it as a multi-player game, with Nash equilibrium serving as the solution concept. We derive a non-asymptotic first-order convergence result for individual gradient descent algorithm to find the Nash equilibrium. We show that the Nash gap of the score network obtained from the algorithm is upper bounded by $\mathcal{O}(\frac{1}{\sqrt{T_{train}}} + \beta)$ where $\beta$ is the regularizing parameter and $T_{train}$ is the number of iterations of the training algorithm. Furthermore, we theoretically establish hyper-parameters for training and sampling algorithm that ensure that we find conditional score networks (under our model) with a worst case sampling error $\mathcal{O}(\epsilon + 1), \forall \epsilon > 0$ across all class labels. Our results offer insights and guarantees for training diffusion models on imbalanced, long-tailed data, with implications for fairness, privacy, and generalization in real-world generative modeling scenarios.

## 1 Introduction

Successful integration of deep learning models into society requires working with real-world data. This comes with many challenges: data quality issues such as inaccurate data, data bias, ethical issues such as breach in privacy, transparency, technical issues such as data integration, generalization, scalability, etc. Furthermore, real world class-labeled datasets are not uniform, but follow a skewed or sometimes referred to as "long-tailed" distribution. It is characterized by a "head" classes that occurs with high probability while the probability of the rest of the classes, often referred to as "tail" classes fall off very quickly. It is well known that the performance of traditional deep learning ([14, 18]) and generative models ([30]) suffer significantly when trained on *long-tailed* distributions.

One might be curious to ask, *"Should deep learning or generative models be concerned with class labels which occur with very low frequency?"* The answer is Yes! Even though individually each class occurs with low frequency, collectively these classes may occur with high probability. Diffusion models, which are the focus of this work, are no exception to this phenomenon. Diffusion models are latent variable generative models which learn diffusion process for a given dataset, such that the process can generate new elements that are distributed similarly as the original dataset (See section 3 for more details). They have become popular techniques in image generation beating traditional models such as GANs [3, 7, 38], natural language processing [39], time series forecasting [26] and in fields of applied chemistry [1], biology [9] and medicine [15] to name a few. However, the study of diffusion models for long-tailed learning is limited. [23] showed that when the traditional conditional Diffusion Denoising Probabilistic Model (DDPM) is trained on a long-tailed distribution,

the conditional DDPM model as shown in [23, Figure 1]", "generates head class images with satisfying performance, whereas conversely, the generated images on tail classes are very likely to show unrecognizable semantics". Moreover, there might be privacy and ethical concerns if the model overfits (memorize) to the tail class label data and replicate them during generation.

Motivated by this, we develop a theory of *Long-Tailed Learning* for diffusion models in a mathematically rigorous manner through the perspective of *Deep Mutual Learning*. Our main results are:

1. Under a suitable metric (KL- divergence) that captures the distance between the learnt class distribution and the ground truth distribution, we derive an upper bound on the worst case distance across all class labels. To do so, we employ Deep Mutual Learning along with the score based diffusion model objective in literature [29, 32]. We present the formulation as a game across conditional score networks and propose Nash equilibrium as the appropriate solution concept.

2. Borrowing ideas from [13] on Deep Mutual Learning, we derive a non-asymptotic first order convergence result for the individual gradient descent algorithm to find the Nash Equilibrium of the proposed game. We show the Nash gap of the score network obtained from the algorithm is upper bounded by $\mathcal{O}(\frac{1}{\sqrt{T_{train}}} + \beta)$ where $\beta$ is the regularizing parameter and $T_{train}$ is the number of iterations of the training algorithm. Finally, we show we can find hyper-parameters for training and sampling such that the score networks obtained from the algorithm enjoys a worst case error bound of $\mathcal{O}(\epsilon + 1)$ for any $\epsilon > 0$ for any class, tail and otherwise.

## 2 Related Works

**Long-Tailed Learning for Diffusion Models.** To tackle the issue with *long-tailed* distributions, diverse techniques have been proposed such as re-sampling [27], re-weighting [27], transfer learning [23], and feature augmentation [10]. The closest work to ours is that of [23] titled "Class- Balancing Diffusion Models" or CBDM and its followups [33, 35]. The paper proposes a distribution adjustment regularizer as a solution along with the usual DDPM objective. This represents a modification in the training phase. Their experiments show that the images generated by CBDM exhibit greater diversity and quality in both quantitative and qualitative ways when trained on CIFAR100/CIFAR100LT datasets. As mentioned in [33], *"CBDM [23] represents an inaugural inquiry into the performance of DDPM within the context of long-tailed data scenarios"*. Motivated by CBDM and contrastive learning, [33] propose adding a penalty function to demarcate the distribution boundaries of different data categories. However, the derivation of the distribution adjustment regularizer in [23, 24, Proposition 2, Appendix A] relies on strong assumptions. They follow a traditional machine learning framework that optimizes over a single objective function with a single neural network and give empirical verification of their method's performance. On the other hand, we define a game across conditional score networks and propose the Nash equilibrium of this game as the egalitarian solution to learn a fair score function for equally good generation over all classes. Furthermore, our framework and analysis do not rely on the strong assumptions made in [23, 24, Proposition 2, Appendix A].

**Deep Mutual Learning.** Deep Mutual Learning (DML) [36] is a knowledge distillation process that allows the transfer of knowledge from a highly powerful model to a smaller faster efficient model. In DML, an ensemble of students (models) learn collaboratively and teach each other throughout the training process. DML has shown promise in visual object tracking [37], metric learning [22], multi-modal recommender systems [16], and classification tasks trained on *Long-Tailed* distributions [21]. The theoretical performance guarantees for models trained with DML are scarce. [13] gives a non-asymptotic first order convergence result for training models for classification task using DML. Deep Mutual Learning literature proposes various methods for optimizing Deep Mutual Learning objectives without specifying the solution concept they seek. In contrast, we show that the individual gradient descent (one method for DML) is seeking a Nash equilibrium of underling multiplayer game. While this result of ours could be of independent interest, in this work we further leverage this result to obtain a generalization result for diffusion models for long-tailed generation.

**Training and Sampling of Score-based (Conditional) Diffusion Models.** The performance of score-based diffusion models have been rigorously studied in the literature of generative modeling. [11, 17, 32] provided a full error analysis of training and sampling from a diffusion model. [11, 17]

parametrize the score network using a random feature model and use gradient flow to train the model. [11] leverages Neural Tangent Kernels to obtain an approximation and generalization error for diffusion models. [32] parametrize the score network by a deep neural network and prove exponential convergence of its gradient descent training dynamic on the empirical loss function. For conditional diffusion models, [8] provides data- dependent approximation bounds of the conditional score function by multi-layered neural network and also give an expected sampling error of the approximated distribution over all class labels. Compared to [8, 32], we consider a finite label class and make no assumption on how the data is distributed. While our result can readily be extended to deeper neural network in line with [32], we parametrize the score function using a two-layer ReLU network (as in [11, 17]) due to the nice properties it induces in the proposed game.

# 3 Basics of Score-Based Diffusion Generative Models

**Notation** Let $\|.\|$ denote the $\ell_2$ norm for vectors and matrices, $\|.\|_F$ be the Frobenius norm. For the discrete time points, we use $t_i$ to denote time point for forward dynamics and $t_i^{\leftarrow}$ for backward dynamics. $\sigma(x)$ where $x \in \mathbb{R}^d$ refers to the ReLU activation function applied element-wise while $\bar{\sigma}_t$ refers to the variance of the forward dynamics. $\tau \in [T_{train}]$ represents the iteration of the training algorithm, which in our case is gradient descent. $\theta_y$ is the training parameter for score for label $y \in \mathcal{Y}$ while $\theta_{-y}$ is the training parameter for score for all label $y' \in \mathcal{Y} - \{y\}$. Given two distributions $p$ and $q$, the KL divergence from $q$ to $p$ is defined as $D_{KL}(p\|q) = \int_{R^d} p(x) \log \frac{p(x)}{q(x)} dx$.

In subsequent sections, we introduce the basics of diffusion model training and generation. Denote the initial conditional distribution as $P_0(X_0 = x|y), \forall x \in \mathcal{A} \subset \mathbb{R}^d$, $\mathcal{A}$ is a compact set of all possible features and $y \in \mathcal{Y}$ where $\mathcal{Y}$ is the finite set of class labels.

## 3.1 Forward and Backward Processes

The use of diffusion model in generative modeling involves two processes:

1. **Forward Process:** The forward process pushes an initial distribution $P_0(.|y)$ to Gaussian by adding noise progressively to $X_0$, and is usually described as an Ornstein-Uhlenbeck (OU) process,

$$dX_t = -f_t X_t dt + g_t dW_t, \text{ with } X_0 \sim P_0(.|y), \quad \forall y \in \mathcal{Y}, \tag{1}$$

where $f_t, g_t$ are functions of $t \in [0, T]$ and $dW_t$ is the incremental Brownian motion or Wiener process, $X_t$ is a $d-$ dimensional random variable with $X_t \sim P_t(.|y)$. The choice of $f_t, g_t$ results in various diffusion model schemes such as Variance Preserving (VP), Variance Exploding (VE) SDE (see [29] for more details).

2. **Backward Process:** To generate a new sample, the forward dynamics can be reversed conditioned on the final distribution $X_T^{\leftarrow} \sim P_0(.|y)$ to get the backward or reverse diffusion process defined as:

$$dX_t^{\leftarrow} = \left[ f_{T-t} X_t^{\leftarrow} + g_{T-t}^2 \nabla_x \log p_{T-t}(X_t^{\leftarrow}|y) \right] dt + g_{T-t} d\bar{W}_t, X_0^{\leftarrow} \sim P_T(.|y), \tag{2}$$

where $X_0^{\leftarrow} \sim P_T$ and $p_t$ is the density of $P_t$. Then $X_{T-t}^{\leftarrow}$ and $X_t$ have the same distribution with density $P_t(.|y)$, which means that the dynamics will push near-Gaussian distributions back to the initial distribution $P_0(.|y), \forall y \in \mathcal{Y}$.

## 3.2 Training via Denoising Score Matching

From (2), to generate samples conditionally, one needs access to $\nabla_x \log p_{T-t}(X_t^{\leftarrow}|y)$, the conditional score function, which is unknown. Let $s_{t,\theta}(x, y)$ be an estimator of $\nabla_x \log p_t(x|y)$. To estimate the conditional score function, a natural loss function to train a model would be the following objective:

$$\mathcal{L}_{conti}(\theta) = \mathbb{E}_y[\mathcal{L}_{conti}^y(\theta)] := \mathbb{E}_y \left[ \frac{1}{2} \int_{t_0}^T \lambda(t) \mathbb{E}_{x(t)} \left[ \left\| \nabla_{x(t)} \log p_t(x(t)|y) - s_{t,\theta}(x(t), y) \right\|_2^2 \right] \right]$$

$$:= \frac{1}{2} \int_{t_0}^T \lambda(t) \mathbb{E}_{(x(t),y)} \left[ \left\| \nabla_{x(t)} \log p_t(x(t)|y) - s_{t,\theta}(x(t), y) \right\|_2^2 \right] dt.$$

Once the conditional score function is learnt, a datum from class label $y \in \mathcal{Y}$ is sampled using the reverse diffusion process given below:

$$d\tilde{X}_t^\leftarrow = \left[ f_{T-t}\tilde{X}_t^\leftarrow + g_{T-t}^2 s_{t,\theta}(x(T-t), y) \right] dt + g_{T-t}d\bar{W}_t, \text{ with } \tilde{X}_0^\leftarrow \sim \mathcal{N}(0, I). \quad (3)$$

To measure how well the learnt score function approximates the ground truth distribution, KL-divergence is employed as the metric. To assess the goodness of the learnt score function through the optimization of $\mathcal{L}_{conti}^y(\theta)$, we have to relate the KL-divergence between the learnt distribution and the ground truth to the training objective. Informally, the KL divergence between the learned distribution and the ground truth distribution is bounded by the score based diffusion model objective $(\mathcal{L}_{conti}(\theta))$ as (see [28, Theorem 1] or [8, Appendix D] for detailed proof)

$$\mathbb{E}_y \left[ \mathcal{D}_{KL}(P_0(.|y)||P_{0,\theta}(.|y)) \right] \lesssim \mathcal{L}_{conti}(\theta) \quad (4)$$

Achieving a bound on the expectation as [8, Theorem 4.1] gives no insights into the worst case sampling error over all $y \in \mathcal{Y}$. In this work, we provide a methodology to achieve an upper bound on $\max_{y \in \mathcal{Y}} \mathcal{D}_{KL}(P_0(.|y)||P_{0,\theta}(.|y))$, thereby addressing the long-tailed issue in generative modeling.

# 4 Long-Tailed Learning

## 4.1 Egalitarian Solution Concept

Previous work in conditional diffusion models [8] have focused on optimizing the following objective

$$\mathcal{L}_{conti}(\theta) = \mathbb{E}_y \left[ \mathcal{L}_{conti}^y(\theta) \right] = \sum_{y \in \mathcal{Y}} p(y)\mathcal{L}_{conti}^y(\theta) \quad (5)$$

for classifier guided sampling [29] or the unconditional score function along with the conditional score function from 5 for classifier free guidance. The above objective is sound when the marginal density of the classes $p(y)$ itself is uniformly distributed. Observe that when optimizing $\mathcal{L}_{conti}(\theta)$ (eq. 5), an optimization algorithm will give more weight towards reducing $\mathcal{L}^y(\theta)$ for head classes (classes with high $p(y)$, appearing with higher frequency in the data). Thus, the trained model overfits the head class, while performing poorly on the tail classes. One way of ensuring that each class label is equally weighted during the training process is to re-weigh each class objective function by a factor inversely proportional to the class marginal density $p(y)$. This ensures that both head and tail classes receive equal weighting during the training process.

$$\mathcal{L}_{conti,balanced}(\theta) = \mathbb{E}_y \left[ \frac{1}{p(y)} \mathcal{L}_{conti}^y(\theta) \right] = \sum_{y \in \mathcal{Y}} \mathcal{L}_{conti}^y(\theta). \quad (6)$$

However, in many real world scenarios the marginal density $p(y)$ is unknown and hence such an accurate reweighting is not possible. For *Long-Tailed Learning*, as we desire to perform well (in terms of generation quality) for every class label, the natural objective would be to minimize $\max_{y \in \mathcal{Y}} \mathcal{D}_{KL}(P_0(.|y)||P_{0,\theta}(.|y))$, that is, minimize the worst-case KL divergence over all $y \in \mathcal{Y}$. Suppose $\mathcal{L}_{conti}^y(\theta)$ is convex in the training parameter $\theta$, then so is $f(\theta) = \max_{y \in \mathcal{Y}} \mathcal{L}_{conti}^y(\theta)$ as maximum of finite convex functions is again convex. $f(\theta)$ may not be differentiable even if $\mathcal{L}_{conti}^y(\theta)$ are differentiable in $\theta$ for all $y \in \mathcal{Y}$. One could use sub-gradient methods to optimize the worst case class loss $\max_y \mathcal{L}^y(\theta)$. However, in practice one has to work with the empirical version of these losses which might be noisy and lead to parameters that are sub-optimal with respect to the population loss.

## 4.2 Nash Equilibrium as a Solution Concept

To enable diffusion models for *Long-tailed* learning, we modify the DM objective to add the mutual learning objective defined as

$$\mathcal{L}_{conti,mut}^y(\theta_y, \theta_{-y}, \omega(.)) = \frac{1}{2} \int_{t_0}^T \omega(t)\mathbb{E}_{x(t) \sim p_t}\mathbb{E}_{y' \sim Q} \left[ \left\| s_{t,\theta_y}(x(t)) - s_{t,\theta_{y'}}(x(t)) \right\|_2^2 \right] dt, \quad (7)$$

to obtain a regularized version of the DM objective function denoted as $\mathcal{L}^y_{cont,reg}(\theta_y, \theta_{-y})$. In the setting of Mutual Learning, the distribution $\mathcal{Q}$ is uniform. But, the distribution can be a hyperparameter over which one could optimize. From now on, we will drop the weighting arguments $\lambda(.), \omega(.)$ in the objective functions, leading to the following regularized objective for each class:

$$\mathcal{L}^y_{conti,reg}(\theta_y, \theta_{-y}) = \mathcal{L}^y_{conti}(\theta_y) + \beta \mathcal{L}^y_{conti,mut}(\theta_y, \theta_{-y}). \tag{8}$$

Learning the score $\nabla_{x(t)} \log p_t(x(t)|y)$ is difficult as it is intractable. Conditioning on $X_0$ and using law of iterated expectation, one can rewrite the objective function as (see [32, Appendix A] for detailed proof) with discretized time points as $0 < t_0 < t_1 < \cdots t_N = T$ to get the training objective

$$\mathcal{L}^y_{reg}(\theta_y, \theta_{-y}) = \mathcal{L}^y(\theta_y) + \beta \mathcal{L}^y_{mut}(\theta_y, \theta_{-y})$$

$$= \frac{1}{2} \sum_{j=1}^N \lambda(t_j)(t_j - t_{j-1}) \mathbb{E}_{X_0} \mathbb{E}_{X_{t_j}|X_0} \left[ \left\| \nabla_{x(t_j)} \log p_t(x_i(t_j)|x_0) - s_{t_j,\theta_y}(x_i(t_j)) \right\|_2^2 \right] +$$

$$\bar{C}(y) + \beta \frac{1}{2} \sum_{j=1}^N \omega(t_j)(t_j - t_{j-1}) \mathbb{E}_{X_0} \mathbb{E}_{X_{t_j}|X_0} \mathbb{E}_{y' \sim Q} \left[ \left\| s_{t_j,\theta_y}(x_i(t_j)) - s_{t,\theta_{y'}}(x_i(t_j)) \right\|_2^2 \right], \tag{9}$$

where $\bar{C}(y) = \frac{1}{2} \sum_{j=1}^N \lambda(t_j)(t_j - t_{j-1}) C_{t_j}(y)$ and $C_t(y) = \mathbb{E}_{X_t} \left\| \nabla \log p_t(.|y) \right\|^2 - \mathbb{E}_{X_0} \mathbb{E}_{X_t|X_0} \left\| \nabla \log p_t(x_t|x_0,y) \right\|^2$. [32, Remark 1] point out that $C(y) < 0$ and hence the first summand in Eq. 9 is always bound below by $-C(y)$. $\bar{C}(y)$ along with the entire first summation in 9 correspond to $\mathcal{L}^y(\theta_y)$ while the third term is $\mathcal{L}^y_{mut}(\theta_y, \theta_{-y})$. As $\bar{C}(y)$ doesn't depend on $\theta$, we can ignore it for the purpose of training. But we note that $C(y)$ will appear in our final worst case sampling error. When the drift and diffusion coefficient of the forward dynamics satisfy some nice properties, the distribution of $p_t(x_t|x_0)$ is normally distributed, whose mean and variance ($\bar{\sigma}_t$) can be explicitly computed. Exploiting this knowledge, one can rewrite the objective function in eq 9 as (See Appendix B.3 for details)

$$\bar{\mathcal{L}}^{n_y}_{reg}(\theta_y, \theta_{-y}) = \bar{\mathcal{L}}^{n_y}(\theta_y) + \beta \bar{\mathcal{L}}^{n_y}_{mut}(\theta_y, \theta_{-y})$$

$$= \frac{1}{2n_y} \sum_{i=1}^{n_y} \sum_{j=1}^N \frac{\lambda(t_j)(t_j - t_{j-1})}{\bar{\sigma}_{t_j}} \left[ \left\| \bar{\sigma}_{t_j} s_{t_j,\theta_y}(x_i(t_j)) + \xi_{ij} \right\|_2^2 \right.$$

$$\left. + \beta \omega(t_j)(t_j - t_{j-1}) \mathbb{E}_{y' \sim Q} \left[ \left\| s_{t_j,\theta_y}(x_i(t_j)) - s_{t,\theta_{y'}}(x_i(t_j)) \right\|_2^2 \right] \right] \tag{10}$$

where $\bar{\mathcal{L}}^{n_y}_{reg}(\theta_y, \theta_{-y})$ is the empirical version of $\mathcal{L}^y_{reg}(\theta_y, \theta_{-y})$ with $n_y$ samples, $\{x_i\}_{i=1}^{n_y}$ with $x_i \sim P_0(.|y)$ denotes the initial data, $\{\xi_{ij}\}_{j=1}^N$ where $\xi_{ij} \sim \mathcal{N}(0, I_d)$ denotes the noise and input data of the neural network is $\{t_j, x_i(t_j)\}_{i=1,j=1}^{n_y,N}$, where $x_i(t_j) \sim P_{t_j}(.|y)$ is obtained from the forward diffusion process.

### 4.2.1 Neural Network Architecture for Score Parametrization

The approximation power of two-layer ReLU network with randomly sampled input layer are well understood from numerous works [12, 25] and has been used to study the generalization properties of Diffusion Models in [11, 17]. We also parametrize the score function $s_{t,\theta_y}$ for each label $y \in \mathcal{Y}$ using a random feature model

$$s_{t,\theta_y}(x) := \frac{1}{m} A_y \sigma(W_y x + U_y e(t)) = \frac{1}{m} \sum_{i=1}^m a_{y,i} \sigma(w_{y,i}^T x + u_{y,i}^T e(t)) \tag{11}$$

where $\sigma(\cdot) = \max\{0, \cdot\}$ is the ReLU activation function, $A_y = (a_{y,1}, \cdots, a_{y,m}) \in \mathbb{R}^{d \times m}$ is the trainable parameter, $W_y = (w_{y,1}, \cdots, w_{y,m})^T \in \mathbb{R}^{m \times d}$ and $U_y = (u_{y,1} \cdots, u_{y,m})^T \in \mathbb{R}^{d \times d_e}$ are randomly initialized embedding matrices that are frozen during training, $e : \mathbb{R}_{\geq 0} \to \mathbb{R}^{d_e}$ is the embedding function for the time. The above model represents a neural network with one hidden layer with $m$ neurons and a $d-$ dimensional vector as an output. Suppose $a_{y,i}, w_{y,i}$ and $u_{y,i}$ are i.i.d. sampled from an underlying distribution $\rho$. Then as $m \to \infty$, we can view

$$s_{t,\theta_y}(x) \to \bar{s}_{t,\bar{\theta}_y}(x) := \mathbb{E}_{a_y,w_y,u_y} \left[ a_y \sigma(w_y^T x + u_y^T e(t)) \right] = \mathbb{E}_{w,u} \left[ a_y(w,u) \sigma(w_y^T x + u_y^T e(t)) \right],$$

---

**Algorithm 1** Individual Gradient Descent(IGD)

---

*Input parameters:* Learning rate $\eta_\tau$
*Initialize:* $(W_y, U_y)_{y \in \mathcal{Y}}$ and $\theta_y^0, \forall y \in \mathcal{Y}$
**for** $\tau = 0...T_{train}$ **do**
    **for** $y = 0...|\mathcal{Y}|$ **do**
        $\theta_y^{\tau+1} \leftarrow \theta_y^\tau - \eta_\tau \nabla_{\theta_y} \bar{\mathcal{L}}_{reg}^{n_y}(\theta_y^\tau, \theta_{-y}^\tau)$
    **end for**
**end for**
*Output:* $(\theta_y, \theta_{-y}) = \min_{\tau \in [T_{train}]} \text{NE-gap}(\theta_y^\tau, \theta_{-y}^\tau)$

---

with $a_y(w, u) := \frac{1}{\rho_0(w,u)} \int_{\mathbb{R}^d} a_y \rho(a, w, u) da_y$ and $\rho_0(w, u) := \int_{\mathbb{R}^d} \rho(a, w, u) da$. The above relation represents $s_{t,\theta_y}(x)$ as an approximation of the continuous version $\bar{s}_{t,\bar{\theta}_y}(x)$, which can be viewed as a neural network with infinite width, i.e., infinite number of neurons in the hidden layer ($m \to \infty$). Furthermore, we assume the embedding matrices $W_y$ and $U_y$ are sampled independently for every $y \in \mathcal{Y}$ from a set with bounded support.

Having defined our loss function, we define the strategy space as $\Theta_y = \{A_y \in \mathbb{R}^{d \times m} : \|A_y\|_F \leq B\}, \forall y \in \mathcal{Y}$. Now, consider the $|\mathcal{Y}|$-player game $\langle \mathcal{Y}, (\bar{\mathcal{L}}_{reg}^{n_y})_{y \in \mathcal{Y}}, (\Theta_y)_{y \in \mathcal{Y}} \rangle$,

**Definition 1** (Nash Gap). *Let $B_y : \Theta_{-y} \to \Theta_y$ represent the best response function for label $y \in \mathcal{Y}$ defined as $B_y(\theta_{-y}) \in \text{argmin}_{\theta \in \Theta_y} \bar{\mathcal{L}}_{reg}^{n_y}(\theta, \theta_{-y})$. Using the best response function, we define the Nash gap of a strategy profile $(\theta_y)_{y \in \mathcal{Y}} \in \times_{y \in \mathcal{Y}} \Theta_y$ as:*

$$\text{NE-gap}((\theta_y)_{y \in \mathcal{Y}}) = \max_{y \in \mathcal{Y}} \bar{\mathcal{L}}_{reg}^{n_y}(\theta_y, \theta_{-y}) - \bar{\mathcal{L}}_{reg}^{n_y}(B_y(\theta_{-y}), \theta_{-y}). \tag{12}$$

**Definition 2** (Nash Equilibrium). *A strategy $(\theta_y')_{y \in \mathcal{Y}} \in \times_{y \in \mathcal{Y}} \Theta_y$ is an $\epsilon$- Nash equilibrium of the game $\langle \mathcal{Y}, (\bar{\mathcal{L}}_{reg}^{n_y})_{y \in \mathcal{Y}}, (\Theta_y)_{y \in \mathcal{Y}} \rangle$ if $\text{NE-gap}((\theta_y')_{y \in \mathcal{Y}}) \leq \epsilon$. When $\text{NE-gap}((\theta_y')_{y \in \mathcal{Y}}) = 0$, then $(\theta_y^*)_{y \in \mathcal{Y}}$ is a Nash equilibrium.*

The ability to find an $\epsilon$- Nash equilibrium of the game $\langle \mathcal{Y}, (\bar{\mathcal{L}}_{reg}^{n_y})_{y \in \mathcal{Y}}, (\Theta_y)_{y \in \mathcal{Y}} \rangle$ is crucial in our analysis to bound the worst case sampling error.

### 4.3 Algorithm

In this section, we propose the individual gradient descent algorithm 1 to find an approximate Nash equilibrium of the game $\langle \mathcal{Y}, (\bar{\mathcal{L}}_{reg}^{n_y})_{y \in \mathcal{Y}}, (\Theta_y)_{y \in \mathcal{Y}} \rangle$. The input parameter for the algorithm is the step-size $\eta_\tau$ where $\tau$ is the $\tau^{th}$ step of the individual gradient descent algorithm. The initialization step samples the embedding matrices and fixes an initial condition for the training parameter $(W_y, U_y, \theta_y^{(0)})_{y \in \mathcal{Y}}$. The individual gradient proceeds for $T_{train}$ steps and within each step an individual gradient update is performed by computing the gradient $\nabla_{\theta_y} \bar{\mathcal{L}}_{reg}^{n_y}(\theta_y^\tau, \theta_{-y}^\tau)$.

**The complexity of finding Nash equilibrium:** One of the most celebrated results in game theory [6] proved that the computational complexity of the problem of computing of a Nash equilibrium in an arbitrary game lies in the complexity class PPAD. So far, there does not exist an polynomial time algorithm that can find an approximate or exact solution to problems in PPAD. The game $\langle \mathcal{Y}, (\bar{\mathcal{L}}_{reg}^{n_y})_{y \in \mathcal{Y}}, (\Theta_y)_{y \in \mathcal{Y}} \rangle$ is a convex minimization game (See B.5). [20] showed that concave maximization games (convex minimization games) also lie in the class PPAD. We present a positive result that in our game $\langle \mathcal{Y}, (\bar{\mathcal{L}}_{reg}^{n_y})_{y \in \mathcal{Y}}, (\Theta_y)_{y \in \mathcal{Y}} \rangle$, individual gradient descent finds an approximate Nash equilibrium whose NE-gap is bounded by $\mathcal{O}(\frac{m^2}{\sqrt{T_{train}}} + \beta)$.

## 5 Main Result

We now present the main result of the capability of diffusion models in long-tailed learning through deep mutual learning. We derive a data-independent worst-case bound for $D_{KL}(p_0(.|y)||p_{0,\theta_{y,t}}(.|y))$. Let $\theta_y^* = \text{argmin}_{\theta_y} \mathcal{L}^y(\theta_y), \forall y \in \mathcal{Y}$ and let $\bar{\theta}_y^*$ be the optimal solution when the true score function

$s_{t,\theta_y}(x)$ is replaced in the class-label objective function $\mathcal{L}^y(\theta_y)$ (equation 9) by its approximation $\bar{s}_{t,\bar{\theta}_y}(x)$. We make one assumption on the support of data distribution (justified in Remark 1).

**Assumption 1.** *We assume that the target distribution $P_0(x|y)$ is continuously differentiable in $x$ and has compact support for every $y \in \mathcal{Y}$. Let for any $y \in \mathcal{Y}$, $x \in \mathcal{A} \subset \mathbb{R}^d$, $\|x\|_\infty \leq K$*

**Generation Algorithm.** We consider the DDPM sampling scheme. Under this scheme $f_t = 1$ and $g_t = \sqrt{2}$ in Eq. 1. Denote the backward time schedule as $\{t_j^\leftarrow\}_{0 \leq j \leq N}$ such that $0 = t_0^\leftarrow < t_1^\leftarrow < \cdots, t_N^\leftarrow = T - \alpha$. To simulate the backward SDE, we use the exponential integrator scheme [34] which can be piecewise expressed as a continuous-time SDE: for any $t \in [t_j^\leftarrow, t_{j+1}^\leftarrow)$. .

$$d\bar{Y}_t = (\bar{Y}_t + 2s_{T-t_j^\leftarrow, \theta_y}(\bar{Y}_{t_j^\leftarrow}))dt + \sqrt{2}d\bar{W}_t. \tag{13}$$

Denote $q_t(.|y) := \text{Law}(\bar{Y}_t), \forall t \in [0, T - \alpha]$. $\gamma_k = t_{k+1}^\leftarrow - t_k^\leftarrow$ and assume there exists $\kappa > 0$ such that $\gamma_k \leq \kappa \min\{1, T - t_{k+1}^\leftarrow\}$. Let $u_2^2$ be such that $\mathbb{E}_{x_0 \sim P_0(.|y)}[\|x\|^2] \leq u_2^2 < \infty, \forall y \in \mathcal{Y}$.

**Remark 1.** *Assumption 1 ensure the data belong to a bounded set and the score is well defined. This also ensures the second moment of the data distribution are bound which is necessary for convergence of forward SDE. Some works [4, 32] do not require the existence of score function for the data distribution $P_0(.|y)$. These works employ early stopping of the reverse (sampling) process. They do so because for non-smooth data distributions $\nabla \log q_t$ can blow up as $t \to T$. This means that the model will approximate $q_{T-\alpha}$ rather than $q_T = P_0(.|y)$, which is acceptable since for small $\alpha$ the distance (e.g. in Wasserstein-p metric) between $q_{T-\alpha}$ and $P_0(.|y)$ is small [4].*

We now present the main result of the paper.

**Theorem 1.** *Given Assumption 1, for $0 < \delta \ll 1$, we have with probability $1 - N(\sum_{y \in \mathcal{Y}} n_y)\delta$ that*

1. *The empirical loss functions $\bar{\mathcal{L}}_{reg}^{n_y}(\theta_y, \theta_{-y})$, are $\frac{L_y}{m^2}$ smooth w.r.t to their own parameter $\theta_y \forall y \in \mathcal{Y}$ (See Lemma 3 in Appendix B.5)*

2. *If one runs individual gradient descent with step-size $\eta_\tau \leq \frac{m^2}{\max_{y \in \mathcal{Y}} L_y \sqrt{T_{train}}}$ for $T_{train}$ iterations and selects the parameter from $(\theta_y^\tau, \theta_{-y}^\tau)_{\tau \in [T_{train}]}$ that minimizes the Nash Gap of the game $\langle \mathcal{Y}, (\bar{\mathcal{L}}_{reg}^{n_y})_{y \in \mathcal{Y}}, (\Theta_y)_{y \in \mathcal{Y}} \rangle$ and samples according to Eq.13, the sampling error*

$$\max_{y \in \mathcal{Y}} D_{KL}(P_\alpha(.|y) \| q_{T-\alpha}(.|y)) \lesssim \max_{y \in \mathcal{Y}} \mathcal{L}^y(\bar{\theta}_y^*) + \tilde{\mathcal{O}}\left(\frac{m^2}{\sqrt{T_{train}}} + \beta\right) +$$

$$\tilde{\mathcal{O}}(\frac{1}{\sqrt{mn^*}} + \frac{1}{m}) + C_0(\kappa^2 N u_2^2 + \kappa T u_2^2 + \exp(-2T)u_2^2) + \bar{\mathcal{C}} \tag{14}$$

*where $n^* = \min_{y \in \mathcal{Y}} n_y$, $\bar{\mathcal{C}} = \max_{y \in \mathcal{Y}} -\bar{C}(y)$ as in Eq. 9, $\kappa^2 N u_2^2 + \kappa T u_2^2$ is an upper bound on the discretization error due to the reverse SDE, $\exp(-2T)u_2^2$ is the error due to the convergence of the forward SDE and constant $C_0$ is some constant. $\tilde{\mathcal{O}}$ hides the $\log \frac{1}{\delta}$ factors, $|\mathcal{Y}|^2$ and bounds on strategy space, embedding matrices and other constants.*

**Corollary 1** (Full Error Analysis). *Fix $\epsilon > 0$ arbitrarily. If $T \geq 1, \alpha < 1$ and $N > \log \frac{1}{\alpha}$, then there exists $0 = t_o < t_1 < \cdots t_N = T - \alpha$ such that for some $\kappa = \Theta(\frac{T + \log \frac{1}{\alpha}}{N})$ and $\gamma_k \leq \kappa \min\{1, T - t_k + 1\} \forall k = 0, 1, \cdots, N - 1$. If we take $T = \frac{1}{2} \log \frac{d}{\epsilon}$, $N = \Theta(\frac{d(T + \log \frac{1}{\alpha})^2}{\epsilon})$, $\beta = \tilde{\Theta}(\epsilon)$, $T_{train} = \tilde{\Theta}(\frac{1}{\epsilon^6})$ and $m = \tilde{\Theta}(\frac{1}{\epsilon^2})$, then under similar conditions as Theorem 1, we achieve*

$$\max_{y \in \mathcal{Y}} D_{KL}(P_{\alpha(.|y)} \| q_{T-\alpha}(.|y)) \lesssim \max_{y \in \mathcal{Y}} \mathcal{L}^y(\bar{\theta}_y^*) + \bar{\mathcal{C}} + \epsilon \tag{15}$$

*where $\tilde{\mathcal{O}}, \tilde{\Theta}$ and $\lesssim$ hides the polynomial of $\log \frac{1}{\delta}$, $|\mathcal{Y}|^2$ and bounds on strategy space, embedding matrices and other constants. $\mathcal{L}^y(\bar{\theta}_y^*)$ is the universal approximation error of approximating the score with two layer network with random ReLUs.*

Corollary 1 gives us the range of hyper-parameters such as width of hidden-layer, number of training steps, discretization of sampling, etc. to achieve worst case sampling error of $\mathcal{O}(\epsilon + 1)$. The $\mathcal{O}(1)$ term $C(y)$ in Eq. 9, can be viewed as the error incurred due to diffusion model's nature in approximating $\nabla \log p_t(x_t|y)$ which is intractable by $\nabla \log p_t(x_t|x_0, y)$ with reverse SDE.

## 5.1 Proof sketch of Theorem 1

We provide a sketch for the proof and defer the details to the Appendix. We use a slight variant of [4, Theorem 1] (See Appendix B for more details) to upper bound the KL-divergence between the distribution approximated by our model and the ground truth to get

$$\max_{y\in\mathcal{Y}} D_{KL}(P_\alpha(.|y)||q_{T-\alpha}(.|y)) \leq \max_{y\in\mathcal{Y}} \mathcal{L}^y(\theta_y) + C_0(\kappa^2 N u_2^2 + \kappa T u_2^2 + \exp(-2T)u_2^2). \quad (16)$$

We then perform the following decomposition for $\max_{y\in\mathcal{Y}} \mathcal{L}^y(\theta_y)$ (See Appendix B.1), where

$$\min_{\tau\in[T_{train}]} \max_{y\in\mathcal{Y}} \mathcal{L}^y(\theta_y^\tau) \leq \max_{y\in Y} \mathcal{L}^y(\theta_y^*) + +2\max_{y\in Y} \sup_{(\theta_y,\theta_{-y})} | \mathcal{L}^y_{reg}(\theta_y,\theta_{-y}) - \bar{\mathcal{L}}^{n_y}_{reg}(\theta_y,\theta_{-y}) | \\ + \min_{\tau\in[T_{train}]} \text{NE-gap}(\theta_y^\tau,\theta_{-y}^\tau) + \beta\max_{y\in Y} \sup_{(\theta_y,\theta_{-y})} \mathcal{L}^y_{mut}(\theta_y,\theta_{-y}). \quad (17)$$

**Proposition 1** (Training and bounding the Nash Gap). *Suppose $\bar{\mathcal{L}}^{n_y}_{reg}(\theta_y^\tau,\theta_{-y}^\tau)$ is $\frac{L_y}{m^2}$ smooth for all $y \in \mathcal{Y}$. Then by selecting a constant learning rate $\eta_\tau \leq \frac{\eta}{\sqrt{T_{train}}} \leq \frac{m^2}{\max_{y\in\mathcal{Y}} L_y \sqrt{T_{train}}}$ that depends on the total iteration $T_{train}$, and using $\tilde{\mathcal{O}}$ to hide the $\log\frac{1}{\delta}$ factors, we have*

$$\min_{\tau\in[T_{train}]} NE\text{-}gap(\theta_y^\tau,\theta_{-y}^\tau) \lesssim \min_{\tau\in[T_{train}]} \max_{y\in\mathcal{Y}} \left\|\nabla_{\theta_y}\bar{\mathcal{L}}^{n_y}_{reg}(\theta_y^\tau,\theta_{-y}^\tau)\right\|^2 = \tilde{\mathcal{O}}(\frac{m^2}{\sqrt{T_{train}}} + \beta). \quad (18)$$

The proof is presented in Appendix B.6. Proposition 1 gives a non-asymptotic first order convergence of individual gradient descent. When no further assumption on the gradient mapping (e.g., (strong) monotonicity of the game $\langle\mathcal{Y}, (\bar{\mathcal{L}}^{n_y}_{reg})_{y\in\mathcal{Y}}, (\Theta_y)_{y\in\mathcal{Y}}\rangle$) is considered, this is the best we can hope for. The iterate at which the minimum Nash Gap is achieved can be tracked by storing the parameters $(\theta_y^\tau, \theta_{-y}^\tau)$ for which the $\max_{y\in\mathcal{Y}} \left\|\nabla_{\theta_y}\bar{\mathcal{L}}^{n_y}_{reg}(\theta_y^\tau,\theta_{-y}^\tau)\right\|^2$ is the least.

**Monte-Carlo Estimate.** To bound $\max_{y\in Y} \mathcal{L}^y(\theta_y^*)$, we employ ideas from [17, Lemma 6]. Informally (See Prop 2 in Appendix B.7 ), for $0 < \delta \ll 1$, with probability $1 - 2N|\mathcal{Y}|\delta$, we achieve

$$\max_{y\in Y} \mathcal{L}^y(\theta_y^*) \lesssim \max_{y\in\mathcal{Y}} \mathcal{L}^y(\bar{\theta}_y^*) + \tilde{\mathcal{O}}\left(\frac{1}{m}\right), \quad (19)$$

where $\tilde{\mathcal{O}}$ hides the $\log\frac{1}{\delta}$ factors. $\mathcal{L}^y(\bar{\theta}_y^*)$ is the error associated with approximating the score of the data using a two layer networks of random ReLUs.

**Rademacher Complexity.** Finally, we bound the generalization error (See Lemma 9 in Appendix B.8 for the derivation) by the Rademacher Complexity

$$\max_{y\in Y} \sup_{(\theta_y,\theta_{-y})} | \mathcal{L}^y_{reg}(\theta_y,\theta_{-y}) - \bar{\mathcal{L}}^{n_y}_{reg}(\theta_y,\theta_{-y}) | = \tilde{\mathcal{O}}\left(\frac{1}{\sqrt{mn^*}}\right) + \bar{C} \quad (20)$$

where $n^* = \min_{y\in\mathcal{Y}} n_y$, $\bar{C} = \max_{y\in\mathcal{Y}} -\bar{C}(y)$. $\tilde{\mathcal{O}}$ hides the $\log\frac{1}{\delta}$ factors, $|\mathcal{Y}|^2$ and bounds on strategy space, embedding matrices and other constants.

**Bound on Mutual Learning Loss** The final term in Eq. 17 $\max_{y\in Y} \sup_{(\theta_y,\theta_{-y})} \mathcal{L}^y_{mut}(\theta_y,\theta_{-y})$ is $\mathcal{O}(1)$ (See Lemma 6 in Appendix B.8).

## 5.2 Interpretation of the Main Result and Implications for Long-tailed Learning

Firstly, when the training objective function are nice, Proposition 1 shows that individual gradient descent employed in Deep Mutual Learning literature is seeking a Nash Equilibrium of an underlying game across different models. Second, when diffusion models are employed for long-tailed generation, Theorem 1 shows that a Nash equilibrium of an underlying game across conditional score network achieves an egalitarian solution w.r.t to sampling error. Our result give insight into the bottleneck process in diffusion generative modeling when faced with limited computing resources and long-tailed data. To the best of our knowledge, our result is the first to provide a comprehensive view of Deep Mutual Learning and long-tailed generation(learning) with diffusion models.

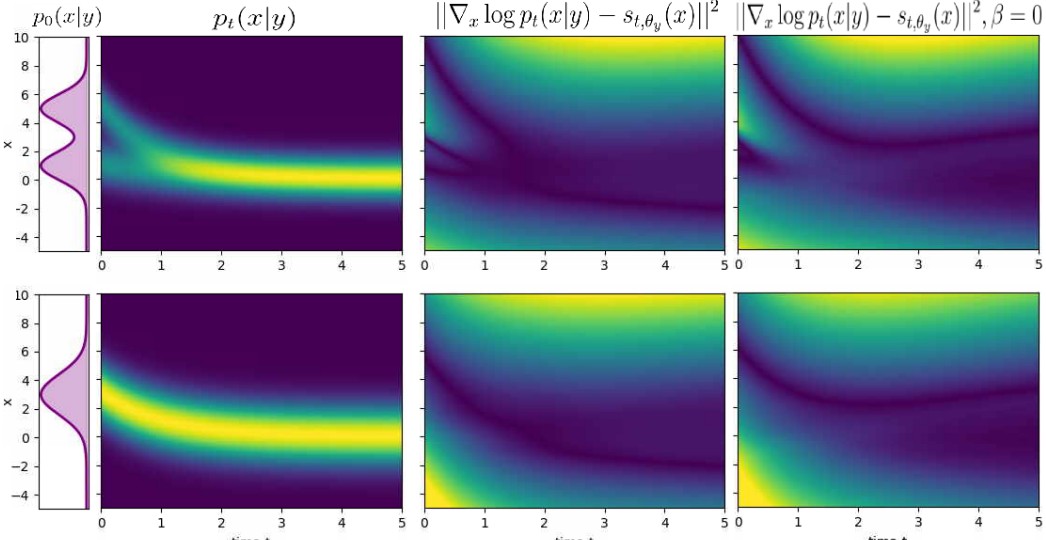

*Figure 1:* Fitting error on a toy demo with and without mutual learning. Top row represents the tail class and bottom row represents head class. The middle column represents mutual learning and the right column represents without mutual learning ($\beta = 0$). Lighter areas represent higher probability region (left column) and larger fitting error (middle and right column)

# 6 Numerical Experiments

## 6.1 Toy Model

Diffusion models are trained to learn the score function $\nabla \log p_t(x_t|x_0, y)$ of the forward process which is then used in the reverse SDE to sample. From Figure 1 (left column), the score model works well when $p_t(x|y)$ is large but suffers from large error when $p_t(x|y)$ is small. This observation can be explained by examining the training loss on Figure 1(middle and right columns). Since the training data is sampled from $p_t(x|y)$, in regions with a low $p_t(x|y)$ value, the learned score network is not expected to work well due to the lack of training data. As a consequence, to ensure $\bar{Y}_0$ is close to $x_0$, one need to make sure $\bar{Y}_t$ stays in the high $p_t(x|y)$ region $\forall t \in [0, T]$. The mutual learning term aligns the score network for the tail classes with the high confidence scores of head classes at the high noise regime area ($t \gg 0$) decreasing the fitting error. This can be seen from a comparison of the heatmap of the tail class (top row middle column) with mutual learning having larger portion of area with low fitting error compared to the case with no mutual learning (top row right column). [1]

## 6.2 Real World Datasets

**Datasets** We perform empirical validation of our theoretical findings with the widely used CIFAR10 dataset in the domain of image synthesis, specifically its long-tailed versions CIFAR10LT. The construction of CIFAR10LT follows from [5], where the size decreases exponentially with its class label index according to the imbalance factor $imb = 0.01$. We also perform experiments on synthetic dataset such as Gaussian Mixture Model and include them in Appendix C.

| Method | FID($\downarrow$) | IS($\uparrow$) |
|---|---|---|
| Vanilla DDPM ($\beta = 0$) | 16.58 | $8.78 \pm 0.15$ |
| Mutual Learning | 14.58 | $8.92 \pm 0.19$ |
| CBDM | 15.28 | $8.11 \pm 0.14$ |

*Table 1:* Best Performance for Various Methods

**Implementation Details** We take the code from [23] and modify the training procedure according to individual gradient descent. The Neural network Architecture employed is U-net as in [23]. To

---

[1]The code is available at `https://github.com/pranoydas51/IGD-ML`

be able to make direct comparisons to DDPM and a rudimentary comparison to CBDM, we modify the code of CBDM and employ error networks for mutual learning (individual gradient descent) instead of score networks as above. We run both CBDM and Individual Gradient Descent(IGD) for $T_{train} = 60k$ training steps. We generate $15k$ samples per class and make the comparison at the $60k$ training step mark. We provide FID, IS across various parameter settings in Appendix C.2.

**Comparison with baselines** The baseline model for us is DDPM models trained individually on each class label dataset ($\beta = 0$). We also make a comparison of mutual learning with Class Balancing Diffusion Models. While empirical experiments on CIFAR10LT shows Mutual Learning perform better than CBDM, we do not make any claim such as mutual learning outperforms CBDM. Since our contribution is theoretical in nature, comprehensive numerical comparison with CBDM is left as a future direction.

# 7  Discussion

**Choice of** $\lambda(t)$ **and** $\omega(t)$**.** We choose $\omega(t)$ as an increasing function of $t$ (as in [23]) and $\lambda(t)$ such that $\frac{\lambda(t)}{\bar{\sigma}_t}$ is non-increasing in $t$. The motivation behind this is to ensure that the training process gives more weight to fitting to the data distribution for smaller $0 < t < T$ and give more weight to the mutual learning objective for high noise regions i.e. larger $0 < t < T$ of the forward diffusion process. There might exist a better weighting function. Our analysis doesn't involve the investigation of an optimal weighting function. We leave this as a future direction to pursue.

**Bound on Approximation Error** $\mathcal{L}^y(\bar{\theta}_y^*)$**.** Given universal approximation results for two-layer networks of Random ReLUs such as [11, Theorem 3.6] and assuming $\nabla \log p_t(x_t|y)$ to be Lipschitz continuous w.r.t. $x_t$, we can follow [11] to achieve an upper bound for $\max_{y \in \mathcal{Y}} \mathcal{L}^y(\bar{\theta}_y^*)$. This bound can be made arbitrarily small by controlling hyperparameters such as bound on RKHS norm of $\bar{s}_{t,\bar{\theta}_y}$ and $0 < \delta \ll 1$.

**Extension of Theoretical Results to Deeper Neural Networks** Following [2], which proves that stochastic gradient descent (SGD) can find global minima in Deep Neural Networks (DNN) in polynomial time (given that the inputs are non-degenerate and the network is over-parameterized), and [32], which extends [2] to determine the training complexity for diffusion models and determine the generalization error of sampling with DNNs, our theoretical analysis can be extended to Deeper Neural Network architecture in three steps. First, we can use [31] to obtain the generalization bound using Rademacher Complexity for DNNs with ReLU activation function. Then, using the fact that $\bar{\mathcal{L}}_{reg}^{n_y}(\theta_y, \theta_{-y}) = \bar{\mathcal{L}}^{n_y}(\theta_y) + \beta \bar{\mathcal{L}}_{mut}^{n_y}(\theta_y, \theta_{-y})$, we observe that [32, Lemma 9] proves the semi-smoothness of $\bar{\mathcal{L}}^{n_y}(\theta_y)$ with high probability. Thus, we can use [2, Theorem 3] to obtain the semi-Smoothness of $\bar{\mathcal{L}}_{mut}^{n_y}(\theta_y, \theta_{-y})$. The only thing that one needs to compute are the various hyperparameter dependent constants. The final step would be to derive a PL like inequality as in [2, Theorem 3] [32, Lemma 1(Appendix D.1)] with high probability. Proving whether $\bar{\mathcal{L}}_{reg}^{n_y}(\theta_y, \theta_{-y})$ satisfies a PL like inequality is challenging. [32, Lemma 1(Appendix D.1)] considers the case without mutual learning. Even though $\bar{\mathcal{L}}^{n_y}(\theta_y)$ and $\bar{\mathcal{L}}_{mut}^{n_y}(\theta_y, \theta_{-y})$ individually satisfy a PL like inequality, their sum may not. We leave this as a conjecture for future work.

**Application to Federated Learning.** Consider the following scenario, each class label $y \in \mathcal{Y}$ is thought of as a client that holds private training data with variable number of training sample points. Individual Gradient Descent then represents local training of score network with global sharing of updated score network parameters while preserving the privacy of local client data. This allows fair learning and generalization among all classes and prevents overfitting (memorization) for class labels with low training data frequency.

**Limitations.** While we achieve a bound on the worst case generalization (sampling) error, the current analysis should be extended to provide insight into whether the performance of the head class score networks is preserved upon adding the mutual learning loss. Further, we set $\mathcal{Q} = Uniform(\mathcal{Y})$ and further investigation is warranted on the effect of the distribution $\mathcal{Q}$ on the worst-case sampling error. It is worth examining if generalization (sampling) error can be made arbitrarily small (also noted in [32, Section 3.3]) i.e. the $\mathcal{O}(1)$ bias be removed. Finally, while we support our analysis with empirical experiments, validating our findings on larger real world datasets CIFAR100LT and a detailed comparison with CBDM [23] could further strengthen the approach.

## Acknowledgments and Disclosure of Funding

We thank Mainak Pal for his help with the numerical simulations. The first author was partially supported by ARO grant W911NF2310266, the second by ONR grant 13001274, the third by NSF under Grants CNS-2313109 and DMS-2502560, and the fourth by ONR grant N000142312604.

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

## A  Appendix / supplemental material

| Notations | |
|---|---|
| Symbol | Meaning |
| $m$ | Number of neurons in hidden-layer of score network |
| $C_{w_y, u_y}$ | Upper bound on $\|w_{y,i}\|_1$, $\|u_{y,i}\|_1$ |
| $F_T^2$ | Upper bound on $\mathbb{E}_{X_0} \mathbb{E}_{\xi_j} \left[ \|\sigma(Wx(t) + Ue(t))\|_2^2 \right], 0 \leq t \leq T$ |
| $L_y$ | $\bar{\mathcal{L}}_{reg}^{n_y}(\theta_y, \theta_{-y})$ is $\frac{L_y}{m^2}$ smooth w.r.t. $\theta_y$ |
| $\phi_y$ | Lipschitz constant of $\bar{\mathcal{L}}_{mut}^{n_y}(\theta_y, \theta_{-y})$ w.r.t. $\theta_y$ |
| $\sigma_y$ | $\bar{\mathcal{L}}_{reg}^{n_y}(\theta_y, \theta_{-y})$ is $\sigma_y$ Lipschitz in $\theta_y$ |
| $B$ | Upper Bound of the Frobenius norm of $A_y$ |

## B  Sampling

Denote the backward time schedule as $\{t_j^\leftarrow\}_{0 \leq j \leq N}$ such that $0 = t_0^\leftarrow < t_1^\leftarrow < \cdots, t_N^\leftarrow = T - \alpha$. Lower case $p_t$ represents the density of $P_t$. We consider the exponential integrator scheme for simulating the backward SDE with

The generation algorithm can be expressed as a piecewise continuous-time SDE: for any $t \in [t_j^\leftarrow, t_{j+1}^\leftarrow)$.

$$d\bar{Y}_t = (\bar{Y}_t + 2s_{T-t_j^\leftarrow, \theta_y}(\bar{Y}_{t_j^\leftarrow}))dt + \sqrt{2}d\bar{W}_t \tag{21}$$

Denote $q_t := \text{Law}(\bar{Y}_t), \forall t \in [0, T - \delta]$.

**Theorem 2.** *[4, Theorem 1] Let Assumption 1 hold. Then there exists a numerical constant $C_0 > 0$, such that*

$$D_{KL}(p_\alpha(.|y) \| q_{T-\alpha}(.|y)) \leq C_0(E_S + E_D + E_F) \tag{22}$$

*where $E_D \leq \kappa^2 N u_2^2 + \kappa T u_2^2$ is the discretization error due to the reverse SDE, $E_F \leq \exp(-2T)u_2^2$ is the error due to the convergence of the forward SDE and $E_S$ is the score estimation error*

$$E_S(\theta_y) = \sum_{j=0}^{N-1} \gamma_j \mathbb{E}_{x \sim p_{T-t_j^\leftarrow}} \left[ \left\| \nabla \log p_{T-t_j^\leftarrow}(x|y) - s_{T-t_j^\leftarrow, \theta_y}(x) \right\|_2^2 \right] \tag{23}$$

*where $\gamma_j := t_{j+1}^\leftarrow - t_j^\leftarrow, \forall j = 0, 1, \cdots N - 1$ is the step-size of the generation algorithm.*

When the training is done over the forward discretization given by $(t_{N-j} = T - t_j^\leftarrow)_{j=0}^{N-1}$, we have

$$E_S = \sum_{j=0}^{N-1} \frac{\bar{\sigma}_{t_{N-j}} \lambda(t_{N-j})}{\lambda(t_{N-j}) \bar{\sigma}_{t_{N-j}}} (t_{N-j} - t_{N-j-1}) \mathbb{E}_{X_0} \mathbb{E}_{X_{t_{N-j}}|X_0} \left\| \bar{\sigma}_{t_{N-j}} s_{t_{N-j}, \theta_y}(X_{t_{N-j}}) + \xi \right\|^2$$

$$+ \sum_{j=0}^{N-1} \frac{\bar{\sigma}_{t_{N-j}}}{\lambda(t_{N-j})} \lambda(t_{N-j})(t_{N-j} - t_{N-j-1}) C_{t_{N-j}}$$

$$\leq 2 \max_j \frac{\bar{\sigma}_{t_{N-j}}}{\lambda(t_{N-j})} \mathcal{L}^y(\theta_y)$$

where

$$\mathcal{L}^y(\theta_y) = \frac{1}{2} \sum_{j=0}^{N-1} \mathbb{E}_{X_0} \mathbb{E}_{X_{t_{N-j}}|X_0} \left[ \lambda(t_{N-j})(t_{N-j} - t_{N-j-1}) \right.$$

$$\left\| \nabla_{x(t_{N-j})} \log p_{t_{N-j}}(x(t_{N-j})|x_0) - s_{t_{N-j}, \theta_y}(x(t_{N-j})) \right\|_2^2 \right] \tag{24}$$

$$+ \frac{1}{2} \sum_{j=0}^{N-1} \lambda(t_{N-j})(t_{N-j} - t_{N-j-1}) C_{t_{N-j}}(y)$$

**Theorem 3.** *(Appendix B and [4, Theorem 1]) Let Assumption 1 hold. Then there exists a numerical constant $C_0 > 0$, such that*

$$D_{KL}(p_\alpha(.|y)||q_{T-\alpha}(.|y)) \leq C_0(\mathcal{L}^y(\theta_y) + E_D + E_F) \tag{25}$$

*where $E_D \leq \kappa^2 N u_2^2 + \kappa T u_2^2$ is the discretization error due to the reverse SDE, $E_F \leq \exp{(-2T)}u_2^2$ is the error due to the convergence of the forward SDE.*

## B.1    Decomposition of $\mathcal{L}^y(\theta_y)$

Let $\theta_y^* = \operatorname{argmin}_{\theta_y} \mathcal{L}^y(\theta_y)$. We further decompose $\mathcal{L}^y(\theta_y)$ as

$$
\begin{aligned}
\max_{y \in Y} \left( \mathcal{L}^y(\theta_y) - \mathcal{L}^y(\theta_y^*) \right) \leq{}& \max_{y \in Y} \left( \mathcal{L}^y(\theta_y) + \beta\mathcal{L}_{mut}^y(\theta_y, \theta_{-y}) - (\mathcal{L}^y(\theta_y^*) + \beta\mathcal{L}_{mut}^y(\theta_y^*, \theta_{-y})) \right. \\
& \left. + \beta(\mathcal{L}_{mut}^y(\theta_y^*, \theta_{-y}) - \mathcal{L}_{mut}^y(\theta_y, \theta_{-y})) \right) \\
\overset{(a)}{\leq}{}& \max_{y \in Y} \left( \mathcal{L}^y(\theta_y) + \beta\mathcal{L}_{mut}^y(\theta_y, \theta_{-y}) - (\mathcal{L}^y(B(\theta_{-y})) \right. \\
& \left. + \beta\mathcal{L}_{mut}^y(B(\theta_{-y}), \theta_{-y})) \right) + \beta\max_{y \in Y} \mathcal{L}_{mut}^y(B(\theta_{-y}), \theta_{-y}) \\
\leq{}& \max_{y \in Y} \left( \mathcal{L}_{reg}^y(\theta_y, \theta_{-y}) - \mathcal{L}_{reg}^y(B(\theta_{-y}), \theta_{-y}) \right) \\
& + \beta\max_{y \in Y} \sup_{(\theta_y, \theta_{-y})} \mathcal{L}_{mut}^y(\theta_y, \theta_{-y})
\end{aligned}
$$

where $(a)$ follows from the fact that $\mathcal{L}_{reg}^y(B(\theta_{-y}), \theta_{-y}) \leq \mathcal{L}_{reg}^y(\theta_y, \theta_{-y})$. We further decompose this to obtain an upper bound on $\max_{y \in Y} \min_t \mathcal{L}^y(\theta_y^t)$

$$
\begin{aligned}
\max_{y \in Y} \mathcal{L}^y(\theta_y) \leq{}& \max_{y \in Y} \mathcal{L}^y(\theta_y^*) + \max_{y \in Y} \left( \mathcal{L}_{reg}^y(\theta_y, \theta_{-y}) - \mathcal{L}_{reg}^y(B(\theta_{-y}), \theta_{-y}) \right) \\
& + \beta\max_{y \in Y} \sup_{(\theta_y, \theta_{-y})} \mathcal{L}_{mut}^y(\theta_y, \theta_{-y}) \\
\overset{(a)}{\leq}{}& \max_{y \in Y} \mathcal{L}^y(\theta_y^*) + \max_{y \in Y} |\; \mathcal{L}_{reg}^y(\theta_y, \theta_{-y}) - \bar{\mathcal{L}}_{reg}^{n_y}(\theta_y, \theta_{-y}) \;| \\
& + \max_{y \in Y} |\; \bar{\mathcal{L}}_{reg}^{n_y}(\theta_y, \theta_{-y}) - \bar{\mathcal{L}}_{reg}^{n_y}(B(\theta_{-y}), \theta_{-y}) \;| \\
& + \max_{y \in Y} |\; \mathcal{L}_{reg}^y(B(\theta_{-y}), \theta_{-y}) - \bar{\mathcal{L}}_{reg}^{n_y}(B(\theta_{-y}), \theta_{-y}) \;| \\
& + \beta\max_{y \in Y} \sup_{(\theta_y, \theta_{-y})} \mathcal{L}_{mut}^y(\theta_y, \theta_{-y}) \\
\overset{(b)}{\leq}{}& \max_{y \in Y} \mathcal{L}^y(\theta_y^*) + 2\max_{y \in Y} \sup_{(\theta_y, \theta_{-y})} |\; \mathcal{L}_{reg}^y(\theta_y, \theta_{-y}) - \bar{\mathcal{L}}_{reg}^{n_y}(\theta_y, \theta_{-y}) \;| \\
& + \text{NE-gap}(\theta_y, \theta_{-y}) + \beta\max_{y \in Y} \sup_{(\theta_y, \theta_{-y})} \mathcal{L}_{mut}^y(\theta_y, \theta_{-y}) \\
\implies \min_{\tau \in [T_{train}]} \max_{y \in \mathcal{Y}} \mathcal{L}^y(\theta_y^\tau) \overset{(c)}{\leq}{}& \max_{y \in Y} \mathcal{L}^y(\theta_y^*) + \\
& + 2\max_{y \in Y} \sup_{(\theta_y, \theta_{-y})} |\; \mathcal{L}_{reg}^y(\theta_y, \theta_{-y}) - \bar{\mathcal{L}}_{reg}^{n_y}(\theta_y, \theta_{-y}) \;| \\
& + \min_{\tau \in [T_{train}]} \text{NE-gap}(\theta_y^\tau, \theta_{-y}^\tau) + \beta\max_{y \in Y} \sup_{(\theta_y, \theta_{-y})} \mathcal{L}_{mut}^y(\theta_y, \theta_{-y})
\end{aligned}
$$

where $(a)$ follows from adding and subtracting the empirical losses $\bar{\mathcal{L}}_{reg}^{n_y}(\theta_y, \theta_{-y})$ and $\bar{\mathcal{L}}_{reg}^{n_y}(B_y(\theta_{-y}), \theta_{-y})$ and using triangle inequality of the max norm, $(b)$ follows from the gradient domination property for strongly convex functions, $(c)$ follows from taking the minimum over the iterates of the algorithm.

## B.2 Boundedness of Forward Dynamics

**Lemma 1.** *Consider the forward diffusion process with linear drift coefficients. For any $\delta > 0, \delta \ll 1$, w.p. (with probability) of atleast $1 - \delta$. we have*

$$\|x(t)\|_\infty \leq C_T \left( \|x(0)\|_\infty + \sqrt{\log \frac{2}{\pi\delta^2}} \right) \tag{26}$$

*where $C_T := \max_{t\in[0,T]} r(t), r(t)v(t)$.*

*Proof:* The proof is similar to [17, Lemma 1] When the drift coefficient $f(.,t) : \mathbb{R}^d \to \mathbb{R}^d$ is linear in $x$ i.e. $f(x,t) = -f(t)x$, the transition kernel $p_{t|0}$ has a closed form

$$p_{t|0}(x(t)|x(0)) = \mathcal{N}(x(t); \mu(t)x(0), \bar{\sigma}^2(t)I_d) \tag{27}$$

where $\mu(t) := \exp\left(\int_0^t f(\xi)d\xi\right), \bar{\sigma}^2(t) := 2\int_0^t \exp\left(2\mu_s - 2\mu_t\right)\sigma_s^2 ds$. Together we get,

$$x(t) = \mu(t)x(0) + \bar{\sigma}(t)z, z \sim \mathcal{N}(0, I_d) \tag{28}$$

For any $\epsilon \sim \mathcal{N}(0,1), c > 1$, we have

$$\mathbb{P}\{\epsilon : |\epsilon| > c\} = 2\int_c^\infty \frac{1}{\sqrt{2\pi}} e^{-\frac{x^2}{2}} dx \leq \frac{1}{\sqrt{2\pi}} \int_c^\infty 2xe^{-\frac{x^2}{2}} dx = \frac{1}{\sqrt{2\pi}} \int_{c^2}^\infty e^{-\frac{x}{2}} dx = \sqrt{\frac{2}{\pi}} e^{-\frac{c^2}{2}} \tag{29}$$

Let $\delta = \sqrt{\frac{2}{\pi}} e^{-\frac{c^2}{2}}$, then

$$\mathbb{P}\{\epsilon : |\epsilon| \leq \sqrt{\log \frac{2}{\pi\delta^2}}\} \geq 1 - \delta \tag{30}$$

Hence, for any $\delta \in (0,1)$ with $\delta \ll 1$, w.p. at least $1 - \delta$, we have

$$\|x(t)\|_\infty \leq C_T \left( \|x(0)\|_\infty + \sqrt{\log \frac{2}{\pi\delta^2}} \right) \tag{31}$$

where $C_T := \max_{t\in[0,T]}\{\mu(t), \bar{\sigma}(t)\}$. Let $C_{T,\delta} = C_T(K + \sqrt{\log \frac{2}{\pi\delta^2}})$

## B.3 Boundedness of Loss function $\bar{\mathcal{L}}_{reg}^{n_y}(\theta_y, \theta_{-y})$

In this section, study some properties of the game defined by $\langle \mathcal{Y}, (\bar{\mathcal{L}}_{reg}^{n_y})_{y\in\mathcal{Y}}, (\Theta_y)_{y\in\mathcal{Y}}\rangle$. From Eq. 8, we have

$$\mathcal{L}_{conti,reg}^y(\theta_y, \theta_{-y}) = \mathcal{L}_{conti}^y(\theta_y) + \beta\mathcal{L}_{conti,mut}^y(\theta_y, \theta_{-y}) \tag{32}$$

where

$$\mathcal{L}_{conti,mut}^y(\theta_y, \theta_{-y}, \omega(.)) = \frac{1}{2}\int_{t_0}^T \omega(t)\mathbb{E}_{x(t)\sim p_t}\mathbb{E}_{y'\sim Q}\left[\left\|s_{t,\theta_y}(x(t)) - s_{t,\theta_{y'}}(x(t))\right\|_2^2\right]dt$$

and

$$\mathcal{L}_{conti}^y(\theta_y, \theta_{-y}) = \frac{1}{2}\int_{t_0}^T \lambda(t)\mathbb{E}_{(x(t),y)}\left[\left\|\nabla_{x(t)}\log p_t(x(t)|y) - s_{t,\theta}(x(t),y)\right\|_2^2\right]dt$$

Conditioning on $X_0$ and using law of iterated expectation, we can write [32, Appendix A], we get

$$\mathcal{L}_{conti,reg}^y(\theta_y, \theta_{-y}) = \frac{1}{2}\int_{t_0}^T \mathbb{E}_{X_0}\mathbb{E}_{X_t|X_0,y}\left[\lambda(t)\left\|s_{t,\theta_y}(x(t)) - \nabla_{x(t)}\log p_t(x(t)|x_0)\right\|_2^2\right.$$

$$\left. + \beta\omega(t)\mathbb{E}_{y'\sim Q}\left[\left\|s_{t,\theta_y}(x(t)) - s_{t,\theta_{y'}}(x(t))\right\|_2^2\right]\right]dt + \frac{1}{2}\int_{t_0}^T \left[\lambda(t)C_t(y)\right]dt$$

where $C_t(y) = \mathbb{E}_{X_t} \|\nabla \log p_t(X_t|y)\|^2 - \mathbb{E}_{X_0} \mathbb{E}_{X_t|X_0} \|\nabla \log p_t(X_t|X_0, y)\|^2$ to learn the score $\nabla_{x(t)} \log p_t(x(t)|x_0, y)$.

Furthermore, we discretize the time points $0 = t_0 < t_1 < \cdots < t_N = T$ to the objective function

$$\mathcal{L}_{reg}^y(\theta_y, \theta_{-y}) = \mathcal{L}^y(\theta_y) + \beta \mathcal{L}_{mut}^y(\theta_y, \theta_{-y})$$

$$= \frac{1}{2} \sum_{j=1}^{N} \lambda(t_j)(t_j - t_{j-1}) \mathbb{E}_{X_0} \mathbb{E}_{X_{t_j}|X_0} \left[ \left\| \nabla_{x(t_j)} \log p_t(x_i(t_j)|x_0) - s_{t_j, \theta_y}(x_i(t_j)) \right\|_2^2 \right] +$$

$$+ \bar{C}(y) + \beta \frac{1}{2} \sum_{j=1}^{N} \omega(t_j)(t_j - t_{j-1}) \mathbb{E}_{X_0} \mathbb{E}_{X_{t_j}|X_0} \mathbb{E}_{y' \sim Q} \left[ \left\| s_{t_j, \theta_y}(x_i(t_j)) - s_{t, \theta_{y'}}(x_i(t_j)) \right\|_2^2 \right]$$

(33)

where $\bar{C}(y) = \frac{1}{2} \sum_{j=1}^{N} \lambda(t_j)(t_j - t_{j-1}) C_{t_j}(y)$ From [32, Appendix A], we have $X_t|X_0 \sim \mathcal{N}(e^{-\mu_t} X_0, \bar{\sigma}_t^2 I)$ and its density function is

$$p_t(x|x_0) = (2\pi \bar{\sigma}_t^2)^{-\frac{d}{2}} \exp(-\frac{\|x - e^{-\mu_t} x_0\|}{2\bar{\sigma}_t^2})$$

Then,

$$\Delta = \mathbb{E}_{X_0} \mathbb{E}_{X_t|X_0} \left\| s_{t_j, \theta_y}(x_i(t_j)) - \nabla_{x(t_j)} \log p_t(x(t_j)|x_0) \right\|$$

$$= \mathbb{E}_{X_0} \mathbb{E}_{X_t|X_0} \left\| s_{t_j, \theta_y}(x_i(t_j)) - \nabla_x \left( - \frac{\|X_t - e^{-\mu_t} X_0\|}{2\bar{\sigma}_t^2} \right) \right\|^2$$

$$= \mathbb{E}_{X_0} \mathbb{E}_{X_t|X_0} \left\| s_{t_j, \theta_y}(x_i(t_j)) + \frac{X_t - e^{\mu_t} X_0}{\bar{\sigma}_t^2} \right\|^2$$

$$= \mathbb{E}_{X_0} \mathbb{E}_{\epsilon_t} \left\| s_{t_j, \theta_y}(x_i(t_j)) + \frac{\epsilon_t}{\bar{\sigma}_t^2} \right\|^2$$

Let $\xi = \frac{\epsilon_t}{\bar{\sigma}_t} \sim \mathcal{N}(0, I)$

$$\Delta = \frac{1}{\bar{\sigma}_t} \mathbb{E}_{X_0} \mathbb{E}_\xi \left\| \bar{\sigma}_t s_{t_j, \theta_y}(x_i(t_j)) + \xi \right\|^2$$

(34)

Finally putting all of it together, we get the empirical loss function

$$\bar{\mathcal{L}}_{reg}^{n_y}(\theta_y, \theta_{-y}) = \frac{1}{2n_y} \sum_{i=1}^{n_y} \sum_{j=1}^{N} \frac{\lambda(t_j)(t_j - t_{j-1})}{\bar{\sigma}_{t_j}} \left[ \left\| \bar{\sigma}_{t_j} s_{t_j, \theta_y}(x_i(t_j)) + \xi_{ij} \right\|_2^2 \right.$$

$$\left. + \beta \omega(t_j)(t_j - t_{j-1}) \mathbb{E}_{y' \sim Q} \left[ \left\| s_{t_j, \theta_y}(x_i(t_j)) - s_{t, \theta_{y'}}(x_i(t_j)) \right\|_2^2 \right] \right]$$

(35)

We will show that the empirical loss function for the label $y \in \mathcal{Y}$, $\bar{\mathcal{L}}_{reg}^{n_y}(\theta_y, \theta_{-y})$ that is optimized is convex and smooth in $\theta_y$ with high probability.

**Lemma 2.** *For $\delta > 0, \delta \ll 1, wp.1 - n_y N \delta$, the empirical loss function*

$$\bar{\mathcal{L}}_{reg}^{n_y}(\theta_y, \theta_{-y}) = \frac{1}{2n_y} \sum_{i=1}^{n_y} \sum_{j=1}^{N} \frac{\lambda(t_j)(t_j - t_{j-1})}{\bar{\sigma}_{t_j}} \left[ \left\| \bar{\sigma}_{t_j} s_{t_j, \theta_y}(x_i(t_j)) + \xi_{ij} \right\|_2^2 \right.$$

$$\left. + \beta \omega(t_j)(t_j - t_{j-1}) \mathbb{E}_{y' \sim Q} \left[ \left\| s_{t_j, \theta_y}(x_i(t_j)) - s_{t, \theta_{y'}}(x_i(t_j)) \right\|_2^2 \right] \right]$$

(36)

*is bounded i.e.*

$$\bar{\mathcal{L}}_{reg}^{n_y}(\theta_y, \theta_{-y}) = \mathcal{O}(\sum_{j=1}^{N} \frac{\lambda(t_j)(t_j - t_{j-1})}{\bar{\sigma}_{t_j}} + \beta \omega(t_j)(t_j - t_{j-1}))$$

**Proof:** From Lemma 1, we have $\delta > 0, \delta \ll 1$

$$\mathbb{P}\{|\xi_{ij}| > \sqrt{\frac{2}{\pi\delta^2}}\} \leq \delta \tag{37}$$

Thus, w.p. $1 - n_y N\delta$, we have $|\xi_{ij}| \leq \sqrt{\frac{2}{\pi\delta^2}}$ and hence we have $\|x(t_j)\|_\infty \leq C_{t_N,\delta}, \forall i = 1, \cdots, n_y$ and $j = 1, ,\cdots, N$

Thus, w.p. $1 - n_y N\delta$

$$
\begin{aligned}
\bar{\mathcal{L}}_{reg}^{n_y}(\theta_y, \theta_{-y}) =& \frac{1}{2n_y} \sum_{i=1}^{n_y} \sum_{j=1}^{N} \frac{\lambda(t_j)(t_j - t_{j-1})}{\bar{\sigma}_{t_j}} \Bigg[ \left\| \bar{\sigma}_{t_j} s_{t_j, \theta_y}(x_i(t_j)) + \xi_{ij} \right\|_2^2 \\
&+ \beta\omega(t_j)(t_j - t_{j-1}) \mathbb{E}_{y' \sim Q} \Bigg[ \left\| s_{t_j, \theta_y}(x_i(t_j)) - s_{t, \theta_{y'}}(x_i(t_j)) \right\|_2^2 \Bigg] \Bigg] \\
\leq& \frac{1}{n_y} \sum_{i=1}^{n_y} \sum_{j=1}^{N} \frac{\lambda(t_j)(t_j - t_{j-1})}{\bar{\sigma}_{t_j}} (\bar{\sigma}_{t_j}^2 \left\| s_{t_j, \theta_y}(x_i(t_j)) \right\|_2^2 + \|\xi_{ij}\|^2) \\
&+ \beta\omega(t_j)(t_j - t_{j-1})(\left\| s_{t_j, \theta_y}(x_i(t_j)) \right\|_2^2 + \max_{y' \in \mathcal{Y}} \left\| s_{t_j, \theta_{y'}}(x_i(t_j)) \right\|_2^2)
\end{aligned}
$$

For a bound on $\left\| s_{t_j, \theta_y}(x(t_j)) \right\|_2$

$$\left\| s_{t_j, \theta_y}(x(t_j)) \right\|_2 = \left\| \frac{1}{m} \sum_{i=1}^{m} a_{y,i} \sigma(w_{y,i}^T x(t_j) + u_{y,i}^T e(t_j)) \right\|_2 \tag{38}$$

$$\overset{(a)}{\leq} \frac{1}{m} \sum_{i=1}^{m} \|a_{y,i}\|_2 |\sigma(w_{y,i}^T x(t_j) + u_{y,i}^T e(t))| \tag{39}$$

$$\overset{(b)}{\leq} \frac{1}{m} \sum_{i=1}^{m} \|a_{y,i}\|_2 (\|w_{y,i}\|_1 \|x(t_j)\|_\infty + \|u_{y,i}\|_1 \|e(t))\|_\infty) \tag{40}$$

$$\leq \frac{1}{m} \sum_{i=1}^{m} \|a_{y,i}\|_2 (C_{t_N,\delta} \|w_{y,i}\|_1 + \max_j \|e(t_j)\|_\infty \|u_{y,i}\|_1) \tag{41}$$

$$\overset{(c)}{\leq} (C_{t_n,\delta} + C_{t_N,e}) C_{w_y, u_y} B \tag{42}$$

where $(a)$ follows from triangle inequality for norms, $(b)$ follows from the fact that the ReLu function satisfies $|\sigma(x)| \leq |x|$ and Holder inequality and $(c)$ follows from the bounds on the embeddings and $x(t_j)$ with $\|w_{y,i}\|_1, \|u_{y,i}\|_1 \leq C_{w_y, u_y}, \forall i \in [m]$. Thus, for $\delta > 0, \delta \ll 1$, we have w.p. $1 - n_y N\delta$

$$\bar{\mathcal{L}}_{reg}^{n_y}(\theta_y, \theta_{-y}) \leq C_1 \sum_{j=1}^{N} \frac{\lambda(t_j)(t_j - t_{j-1})}{\bar{\sigma}_{t_j}} + \beta\omega(t_j)(t_j - t_{j-1}) \tag{43}$$

where $C_1 = (\bar{\sigma}_{t_N}^2 + 2)(C_{t_n,\delta} + C_{t_N,e})^2 C_{w_y, u_y}^2 B^2 + \frac{2}{\pi\delta^2}$. Since $\bar{\sigma}_{t_j}$ is non-decreasing in $j$, so $\max_j \bar{\sigma}_{t_j} = \bar{\sigma}_{t_N}$.

## B.4 Boundedness of Gradient of Loss function $\bar{\mathcal{L}}_{reg}^{n_y}(\theta_y, \theta_{-y})$

$$\left\|\nabla_{A_y}\bar{\mathcal{L}}_{reg}^{n_y}(\theta_y, \theta_{-y})\right\|_F^2 = \sum_{k=1}^{d}\left\|\nabla_{(A_y)_k}\bar{\mathcal{L}}_{reg}^{n_y}(\theta_y, \theta_{-y})\right\|^2$$

$$= \sum_{k=1}^{d}\left\|\frac{1}{n_y}\sum_{i=1}^{n_y}\sum_{j=1}^{N}\lambda(t_j)(t_j - t_{j-1})(\bar{\sigma}_{t_j}s_{t_j,\theta_y}(x(t_j)) + \xi_{ij})_k\sigma(W_yx(t_j) + U_ye(t_j))\right.$$

$$\left.+\beta\omega(t_j)(t_j - t_{j-1})\mathbb{E}_{y'}[(s_{t_j,\theta_y}(x(t_j)) - s_{t_j,\theta_{y'}}(x(t_j)))_k\sigma(W_yx(t_j) + U_ye(t_j))]\right\|^2$$

$$\leq 2\sum_{k=1}^{d}\left\|\frac{1}{n_y}\sum_{i=1}^{n_y}\sum_{j=1}^{N}\lambda(t_j)(t_j - t_{j-1})(\bar{\sigma}_{t_j}s_{t_j,\theta_y}(x(t_j)) + \xi_{ij})_k\sigma(W_yx(t_j) + U_ye(t_j))\right\|^2$$

$$+2\beta^2\sum_{k=1}^{d}\left\|\frac{1}{n_y}\sum_{i=1}^{n_y}\sum_{j=1}^{N}\omega(t_j)(t_j - t_{j-1})\mathbb{E}_{y'}[(s_{t_j,\theta_y}(x(t_j)) - s_{t_j,\theta_{y'}}(x(t_j)))_k\cdot\right.$$

$$\left.\sigma(W_yx(t_j) + U_ye(t_j))]\right\|^2$$

$$\leq 2\frac{N}{n_y}\sum_{i=1}^{n_y}\sum_{j=1}^{N}\lambda(t_j)^2(t_j - t_{j-1})^2\sum_{k=1}^{d}\left\|\bar{\sigma}s_{t_j,\theta_y}(x(t_j)) + \xi_{ij}\right\|^2\left\|\sigma(W_yx(t_j) + U_ye(t_j))\right\|^2$$

$$+2\beta^2\frac{N}{n_y}\sum_{i=1}^{n_y}\sum_{j=1}^{N}\omega(t_j)^2(t_j - t_{j-1})^2\cdot$$

$$\sum_{k=1}^{d}\mathbb{E}_{y'}[\left\|s_{t_j,\theta_y}(x(t_j)) - s_{t_j,\theta_{y'}}(x(t_j))\right\|^2]\left\|\sigma(W_yx(t_j) + U_ye(t_j))\right\|^2$$

$$\leq 4Nd\left\|\sigma(W_yx(t_j) + U_ye(t_j))\right\|_2^2\max_j\{\lambda(t_j)(t_j - t_{j-1})\bar{\sigma}_{t_j}, \beta\omega(t_j)(t_j - t_{j-1})\}\bar{\mathcal{L}}_{reg}^{n_y}(\theta_y, \theta_{-y})$$

$$\leq 4Nd^2(C_{t_N,\delta} + C_{t_N,e})^2C_{w_y,u_y}^2\max_j\{\lambda(t_j)(t_j - t_{j-1})\bar{\sigma}_{t_j}, \beta\omega(t_j)(t_j - t_{j-1})\}\bar{\mathcal{L}}_{reg}^{n_y}(\theta_y, \theta_{-y})$$

Since $w.p.1 - n_yN\delta$ the empirical loss function $\bar{\mathcal{L}}_{reg}^{n_y}(\theta_y, \theta_{-y})$ is bounded, $\left\|\nabla_{A_y}\bar{\mathcal{L}}_{reg}^{n_y}(\theta_y, \theta_{-y})\right\|_F^2$ is bounded with the same probability.

This also shows that for fixed $\theta_{-y}, (W_y, U_y)_{y\in\mathcal{Y}}, w.p.1-n_yN\delta, \bar{\mathcal{L}}_{reg}^{n_y}(\theta_y, \theta_{-y})$ is a Lipschitz function in $\theta_y$ with Lipschitz constant $\sigma_y$ such that $\sigma_y^2 = 4C_1Nd^2(C_{t_N,\delta} + C_{t_N,e})^2C_{w_y,u_y}^2\max_j\{\lambda(t_j)(t_j - t_{j-1})\bar{\sigma}_{t_j}, \beta\omega(t_j)(t_j - t_{j-1})\}\left(\sum_{j=1}^{N}\frac{\lambda(t_j)(t_j-t_{j-1})}{\bar{\sigma}_{t_j}} + \beta\omega(t_j)(t_j - t_{j-1})\right)$

## B.5 Smoothness of Loss Function $\bar{\mathcal{L}}_{reg}^{n_y}(\theta_y, \theta_{-y})$

**Lemma 3.** *Let* $(W_y, U_y)_{y\in\mathcal{Y}}, \theta_{-y}, \{t_j\}_{j=1}^{N}$ *be fixed.* *Let* $L_y = d(C_{t_N,\delta} + C_{t_N,e})^2C_{w_y,u_y}^2\sum_{j=1}^{N}\left(\lambda(t_j)(t_j - t_{j-1})\bar{\sigma}_{t_j} + \beta\omega(t_j)(t_j - t_{j-1})\right)$. *Then for* $\delta > 0, \delta \ll 1$, *w.p.* $1 - n_yN\delta, \bar{\mathcal{L}}_{reg}^{n_y}(\theta_y, \theta_{-y})$ *is* $\frac{L_y}{m^2}$ *smooth and convex in* $\theta_y$.

**Proof** We have,

$$\bar{\mathcal{L}}_{reg}^{n_y}(\theta_y, \theta_{-y}) = \frac{1}{2n_y}\sum_{i=1}^{n_y}\sum_{j=1}^{N}\frac{\lambda(t_j)(t_j - t_{j-1})}{\bar{\sigma}_{t_j}}\left[\left\|\bar{\sigma}_{t_j}s_{t_j,\theta_y}(x(t_j)) + \xi_{ij}\right\|_2^2\right.$$

$$\left.+\beta\omega(t_j)(t_j - t_{j-1})\mathbb{E}_{y'\sim Q}\left[\left\|s_{t_j,\theta_y}(x(t_j)) - s_{t,\theta_{y'}}(x(t_j))\right\|_2^2\right]\right] \tag{44}$$

To show smoothness, we will show that the function $f(\theta_y) = \left\|\bar{\sigma}_{t_j} s_{t_j,\theta_y}(x(t_j)) + \xi_{ij}\right\|_2^2$ and $g(\theta_y) = \left\|s_{t_j,\theta_y}(x_{t_j}) - s_{t_j,\theta_{y'}}(x(t_j))\right\|_2^2$ are individually smooth. Once we prove this, it is easy to show $\bar{\mathcal{L}}_{reg}^{n_y}(\theta_y, \theta_{-y})$ is smooth as the linear combination of smooth functions is again smooth. To show smoothness, we need to show that $\left\|\nabla_{\theta_y}^2 f(\theta_y)\right\|$ and $\left\|\nabla_{\theta_y}^2 g(\theta_y)\right\|$ have a bounded norm. Recall that $s_{t,\theta_y}(x) = \frac{1}{m} A_y \sigma(W_y x(t) + U_y e(t))$. Let $h_1(x,t) := \sigma(W_y x + U_y e(t))$, $h_2(x,t) := s_{t,\theta_{y'}}(x)$, $h_3(i,j) = \xi_{ij}$, we have

$$f(\theta_y) = \left\|\bar{\sigma}_{t_j} s_{t_j,\theta_y}(x(t_j)) + \xi_{ij}\right\|^2 \tag{45}$$

$$= \frac{\bar{\sigma}_{t_j}^2}{m^2} h_1^T(x(t_j), t_j) A_y^T A_y h_1(x(t_j), t_j) - 2\bar{\sigma}_{t_j} h_3^T(i,j)\left(\frac{A_y}{m}\right) h_1(x(t_j), t_j) \tag{46}$$

$$+ h_3^T(i,j) h_3(i,j) \tag{47}$$

$$\overset{a}{=} \frac{\bar{\sigma}_{t_j}^2}{m^2}\text{trace}(A_y^T A_y B_1) - \frac{2\bar{\sigma}_{t_j}}{m}\text{trace}(A_y B_3) + \text{constant} \tag{48}$$

$$\overset{b}{=} \frac{\bar{\sigma}_{t_j}^2}{m^2} vec(A_y)^T (B_1 \bigotimes I) vec(A_y) - \frac{2\bar{\sigma}_{t_j}}{m} vec(B_3^T)^T vec(A_y) + \text{constant} \tag{49}$$

where $(a)$ follows from the identity $x^T A y = \text{trace}(B y x^T)$, $(b)$ follows from the following identities

$$\text{trace}(A^T A B) = \text{trace}(A B A^T) = vec(A)^T (B \bigotimes I) vec(A)$$

$$\text{trace}(AB) = vec(A)^T vec(B^T)$$

and $B_3 = h_1(x(t_j), t_j) h_3^T(i,j)$.

Similarly, we have for $g(\theta_y)$

$$g(\theta_y) = \left\|s_{t,\theta_y}(x(t_j)) - s_{t,\theta_{y'}}(x(t_j))\right\|^2$$

$$= \frac{1}{m^2} h_1^T(x(t_j), t_j) A_y^T A_y h_1(x(t_j), t_j) - 2h_2^T(x(t_j), t_j)\left(\frac{A_y}{m}\right) h_1(x(t_j), t_j)$$

$$+ h_2^T(x(t_j), t_j) h_2(x(t_j), t_j)$$

$$\overset{a}{=} \frac{1}{m}\text{trace}(A_y^T A_y B_1) - \frac{2}{m}\text{trace}(A_y B_2) + \text{constant}$$

$$\overset{b}{=} \frac{1}{m^2} vec(A_y)^T (B_1 \bigotimes I) vec(A_y) - \frac{2}{m} vec(B_2^T)^T vec(A) + \text{constant}$$

where $B_1 := h_1(x(t_j), t_j) h_1^T(x(t_j), t_j)$ and $B_2 := h_1(x(t_j), t_j) h_2^T(x(t_j), t_j)$. Thus,

$$\frac{1}{\bar{\sigma}_{t_j}^2}\nabla_{\theta_y}^2 f(\theta_y) = \nabla_{\theta_y}^2 g(\theta_y) = \nabla_{vec(A_y)}^2 g(\theta_y) = \frac{2}{m^2}(B_1 \bigotimes I) \tag{50}$$

The eigenvalues of $(B_1 \bigotimes I)$ is the same as $B_1$ with multiplicity. Thus, to show smoothness, we need to bound the maximum eigenvalues of $B_1$. For any $v \in \mathbb{R}^m$

$$0 \le v^T B_1 v = (v^T h_1(x(t_j), t_j))^2 \le d \left\|h_1(x(t_j), t_j)\right\|_\infty^2 v^T v \tag{51}$$

Now,

$$\left\|\sigma(W_y x(t_j) + U_y e(t_j))\right\|_\infty = \max_{i=1,\cdots,m} \sigma(w_{y,i}^T x(t_j) + u_{y,i}^T e(t_j)) \tag{52}$$

$$\le \max_{i=1,\cdots,m} |w_{y,i}^T x(t_j) + u_{y,i}^T e(t_j)| \tag{53}$$

$$\le \max_{i=1,\cdots,m} \left\|w_{y,i}\right\|_1 \left\|x(t_j)\right\|_\infty + \left\|u_{y,i}\right\|_1 \left\|e(t_j)\right\|_\infty \tag{54}$$

$$\le (C_{t_N,\delta} + C_{t_N,e}) C_{w_y,u_y} \tag{55}$$

Thus, we have for any $v \in \mathbb{R}^m$

$$0 \leq v^T B_1 v \leq d(C_{t_N,\delta} + C_{t_N,e})^2 C_{w_y,u_y}^2 v^T v \tag{56}$$

Since w.p. $1 - n_y N\delta$ we have $\{\|x_{ij}\|_\infty \leq C_{t_N,\delta}\}_{i=1,j=1}^{n_y,N}$, we have with the same probability $f(\theta_y)$ and $g(\theta_y)$ are smooth in $\theta_y$ for every $W_y, U_y, x(t_j), \theta_{-y}$.

Thus, $\bar{\mathcal{L}}_{reg}^{n_y}(\theta_y, \theta_{-y})$ is $L_y \frac{1}{m^2} = \frac{1}{m^2} d(C_{t_N,\delta} + C_{t_N,e})^2 C_{w_y,u_y}^2 \sum_{j=1}^N \left( \lambda(t_j)(t_j - t_{j-1})\bar{\sigma}_{t_j} + \beta\omega(t_j)(t_j - t_{j-1}) \right)$ smooth.

## B.6 Proof: First order convergence of the algorithm

**Proof** Our proof follows closely along the lines of [13]. Let $\bar{\mathcal{L}}_{reg}^{n_y}(\theta_y, \theta_{-y})$ be the empirical version of $\mathcal{L}_{reg}^y(\theta_y, \theta_{-y})$ with $n_y$ samples. By $L_y$ smoothness of $\bar{\mathcal{L}}_{reg}^{n_y}(\theta_y, \theta_{-y})$ we have, for any $y \in \mathcal{Y}$,

$$\bar{\mathcal{L}}_{reg}^{n_y}(\theta_y^{\tau+1}, \theta_{-y}^\tau) \leq \bar{\mathcal{L}}_{reg}^{n_y}(\theta_y^\tau, \theta_{-y}^\tau) + \langle \nabla_{\theta_y} \bar{\mathcal{L}}_{reg}^{n_y}(\theta_y^\tau, \theta_{-y}^\tau), \theta_y^{\tau+1} - \theta_y^\tau \rangle$$
$$+ \frac{L_y}{2} \left\| \theta_y^{\tau+1} - \theta_y^\tau \right\|^2 \tag{57}$$

$$\implies \bar{\mathcal{L}}_{reg}^{n_y}(\theta_y^{\tau+1}, \theta_{-y}^\tau) \leq \bar{\mathcal{L}}_{reg}^{n_y}(\theta_y^\tau, \theta_{-y}^\tau) - \eta_\tau \left\| \nabla_{\theta_y} \bar{\mathcal{L}}_{reg}^{n_y}(\theta_y^\tau, \theta_{-y}^\tau) \right\|^2$$
$$+ \frac{L_y}{2} \eta_\tau^2 \left\| \nabla_{\theta_y} \bar{\mathcal{L}}_{reg}^{n_y}(\theta_y^\tau, \theta_{-y}^\tau) \right\|^2 \tag{58}$$

$$\implies \eta_\tau \left\| \nabla_{\theta_y} \bar{\mathcal{L}}_{reg}^{n_y}(\theta_y^\tau, \theta_{-y}^\tau) \right\|^2 \leq \bar{\mathcal{L}}_{reg}^{n_y}(\theta_y^\tau, \theta_{-y}^\tau) - \bar{\mathcal{L}}_{reg}^{n_y}(\theta_y^{\tau+1}, \theta_{-y}^\tau)$$
$$+ \frac{L_y}{2} \eta_\tau^2 \left\| \nabla_{\theta_y} \bar{\mathcal{L}}_{reg}^{n_y}(\theta_y^t, \theta_{-y}^t) \right\|^2 \tag{59}$$

$$\implies \sum_{\tau=1}^{T_{train}} \eta_\tau \left\| \nabla_{\theta_y} \bar{\mathcal{L}}_{reg}^{n_y}(\theta_y^\tau, \theta_{-y}^\tau) \right\|^2 \leq \bar{\mathcal{L}}^{n_y}(\theta_y^t) - \bar{\mathcal{L}}^{n_y}(\theta_y^{\tau+1}) + \beta \sum_{\tau=1}^{T_{train}} \psi(\theta_y^{\tau+1}, \theta_y^\tau, \theta_{-y}^\tau)$$
$$+ \sum_{\tau=1}^{T_{train}} \frac{L_y}{2} \eta_\tau^2 \left\| \nabla_{\theta_y} \bar{\mathcal{L}}_{reg}^{n_y}(\theta_y^\tau, \theta_{-y}^\tau) \right\|^2 \tag{60}$$

$$\implies \sum_{\tau=1}^{T_{train}} \eta_\tau \left\| \nabla_{\theta_y} \bar{\mathcal{L}}_{reg}^{n_y}(\theta_y^\tau, \theta_{-y}^\tau) \right\|^2 \leq \bar{\mathcal{L}}^{n_y}(\theta_y^1) - \bar{\mathcal{L}}^{n_y}(\theta_y^{\tau+1}) + \sum_{\tau=1}^{T_{train}} \frac{L_y}{2} \eta_\tau^2 \sigma_y^2$$
$$+ \beta \sum_{\tau=1}^{T_{train}} \psi(\theta_y^{\tau+1}, \theta_y^\tau, \theta_{-y}^\tau) \tag{61}$$

$$\implies \min_{\tau \in [T_{train}]} \left\| \nabla_{\theta_y} \bar{\mathcal{L}}_{reg}^{n_y}(\theta_y^\tau, \theta_{-y}^\tau) \right\|^2 \leq \frac{\bar{\mathcal{L}}^{n_y}(\theta_y^1) - \bar{\mathcal{L}}^{n_y}(\theta_y^*) + \frac{L_y}{2}\sigma_y^2 \sum_{\tau=1}^{T_{train}} \eta_\tau^2}{\sum_{\tau=1}^{T_{train}} \eta_\tau}$$
$$+ \beta \frac{\sum_{t=1}^{T_{train}} \psi(\theta_y^{\tau+1}, \theta_y^\tau, \theta_{-y}^\tau)}{\sum_{\tau=1}^T \eta_\tau} \tag{62}$$

where $\psi(\theta_y^{\tau+1}, \theta_y^\tau, \theta_{-y}^\tau) = \bar{\mathcal{L}}_{mut}^{n_y}(\theta_y^\tau, \theta_{-y}^\tau) - \bar{\mathcal{L}}_{mut}^{n_y}(\theta_y^{\tau+1}, \theta_{-y}^\tau)$.

### B.6.1 Analyzing the Bias Term

**Lemma 4.** *Suppose $\theta_{-y}, (W_y, U_y)_{y \in \mathcal{Y}}$ are fixed. Let $\phi_y = d^{1.5} N(C_{t_N,\delta} + C_{t_N,e})^2 C_{w_y,u_y}^2 B \max_j \omega(t_j)(t_j - t_{j-1})$. Then for $\delta > 0, \delta \ll 1, w.p.1 - n_y N\delta$, we have*

$$\bar{\mathcal{L}}_{mut}^{n_y}(\theta_y, \theta_{-y}) = \frac{1}{2n_y} \sum_{i=1}^{n_y} \sum_{j=1}^N \omega(t_j)(t_j - t_{j-1})\mathbb{E}_{y' \sim Q}\left[ \left\| s_{t_j,\theta_y}(x(t_j)) - s_{t,\theta_{y'}}(x(t_j)) \right\|_2^2 \right] \tag{63}$$

*is $\phi_y$ Lipschitz in $\theta_y$.*

**Proof:**

$$\left\|\nabla_{A_y}\bar{\mathcal{L}}^{n_y}_{reg,mut}(\theta_y,\theta_{-y})\right\|^2_F = \sum_{k=1}^{d}\left\|\nabla_{(A_y)_k}\bar{\mathcal{L}}^{n_y}_{mut}(\theta_y,\theta_{-y})\right\|^2$$

$$= \sum_{k=1}^{d}\left\|\frac{1}{n_y}\sum_{i=1}^{n_y}\sum_{j=1}^{N}\omega(t_j)(t_j-t_{j-1})\mathbb{E}_{y'}[(s_{t_j,\theta_y}(x(t_j))-s_{t_j,\theta_{y'}}(x(t_j)))_k\sigma(W_y x(t_j)+U_y e(t_j))]\right\|^2$$

$$\leq \frac{N}{n_y}\sum_{i=1}^{n_y}\sum_{j=1}^{N}\omega(t_j)^2(t_j-t_{j-1})^2.$$

$$\sum_{k=1}^{d}\mathbb{E}_{y'}[\left\|s_{t_j,\theta_y}(x(t_j))-s_{t_j,\theta_{y'}}(x(t_j))\right\|^2]\left\|\sigma(W_y x(t_j)+U_y e(t_j))\right\|^2$$

$$\leq \left\|\sigma(W_y x(t_j)+U_y e(t_j))\right\|^2 Nd\max_j\omega(t_j)(t_j-t_{j-1})\bar{\mathcal{L}}^{n_y}_{mut}(\theta_y,\theta_{-y})$$

$$\leq 4d^2 N(C_{t_N,\delta}+C_{t_N,e})^2 C^2_{w_y,u_y}\max_j\omega(t_j)(t_j-t_{j-1})\bar{\mathcal{L}}^{n_y}_{mut}(\theta_y,\theta_{-y})$$

$$\leq d^3 N(C_{t_N,\delta}+C_{t_N,e})^4 C^4_{w_y,u_y}B^2\max_j\omega(t_j)(t_j-t_{j-1})\sum_{j=1}^{N}\omega(t_j)(t_j-t_{j-1})$$

Since $\bar{\mathcal{L}}^{n_y}_{mut}(\theta_y,\theta_{-y}) \leq \bar{\mathcal{L}}^{n_y}_{reg}(\theta_y,\theta_{-y})$ and w.p.$1-n_y N\delta$, $\bar{\mathcal{L}}^{n_y}_{reg}(\theta_y,\theta_{-y})$ is bounded. Thus, $\left\|\nabla_{A_y}\bar{\mathcal{L}}^{n_y}_{mut}(\theta_y,\theta_{-y})\right\|^2_F$ is bounded and hence $\bar{\mathcal{L}}^{n_y}_{mut}(\theta_y,\theta_{-y})$ is Lipschitz in $\theta_y$ with $\phi_y = d^{1.5}N(C_{t_N,\delta}+C_{t_N,e})^2 C^2_{w_y,u_y}B\max_j\omega(t_j)(t_j-t_{j-1})$ Here,

$$\psi(\theta_y^{\tau+1},\theta_y^{\tau},\theta_{-y}^{\tau}) = \left|\bar{\mathcal{L}}^{n_y}_{mut}(\theta_y^{\tau},\theta_{-y}^{t})-\bar{\mathcal{L}}^{n_y}_{mut}(\theta_y^{\tau+1},\theta_{-y}^{\tau})\right| \tag{64}$$

$$\leq\phi_y\left\|\theta_y^{\tau}-\theta_y^{\tau+1}\right\| \tag{65}$$

$$\leq\phi_y\eta_t\left\|\nabla_{A_y}\bar{\mathcal{L}}^{n_y}_{reg}(\theta_y^{\tau},\theta_{-y}^{\tau})\right\| \tag{66}$$

$$\leq\phi_y\eta_{\tau}\sigma_y \tag{67}$$

By taking $\eta_{\tau}\leq\frac{m^2}{\max_{y\in\mathcal{Y}}L_y\sqrt{T_{train}}},\forall y\in\mathcal{Y}$

$$\max_{y\in\mathcal{Y}}\min_{\tau\in[T_{train}]}\left\|\nabla_{\theta_y}\bar{\mathcal{L}}^{n_y}_{reg}(\theta_y^{\tau},\theta_{-y}^{\tau})\right\|^2 = \max_{y\in\mathcal{Y}}\mathcal{O}\left(\frac{2(\bar{\mathcal{L}}^{n_y}(\theta_y^0))-\bar{\mathcal{L}}^{n_y}(\theta_y^*))}{\max_{y\in\mathcal{Y}}L_y\sqrt{T_{train}}}+\frac{\sigma_y^2}{\sqrt{T_{train}}}+\beta\phi_y\sigma_y\right) \tag{68}$$

$$= \mathcal{O}\left(\frac{m^2}{\sqrt{T_{train}}}+\beta\right) \tag{69}$$

For $(\theta_y,\theta_{-y})$, we have

$$\text{NE-gap}(\theta_y,\theta_{-y}) = \max_{y\in Y}\mid\bar{\mathcal{L}}^{n_y}_{reg}(\theta_y,\theta_{-y})-\bar{\mathcal{L}}^{n_y}_{reg}(B(\theta_{-y}),\theta_{-y})\mid \tag{70}$$

$$\leq \max_{y\in\mathcal{Y}}\left\|\nabla_{\theta_y}\bar{\mathcal{L}}^{n_y}_{reg}(\theta_y,\theta_{-y})\right\|^2\|\theta_y-B(\theta_{-y})\|_2^2 \tag{71}$$

Since the strategy space for $\theta_y$ is bounded in norm. We have

$$\text{NE-gap}(\theta_y,\theta_{-y}) \lesssim \max_{y\in\mathcal{Y}}\left\|\nabla_{\theta_y}\bar{\mathcal{L}}^{n_y}_{reg}(\theta_y,\theta_{-y})\right\|^2 \tag{72}$$

$$\implies \min_{\tau\in[T_{train}]}\text{NE-gap}(\theta_y^{\tau},\theta_{-y}^{\tau}) = \mathcal{O}\left(\frac{m^2}{\sqrt{T_{train}}}+\beta\right) \tag{73}$$

## B.7 Monte Carlo Error of the Finite Neural Network

Observe that

$$\mathcal{L}^y(\theta_y) = \frac{1}{2} \sum_{j=1}^N \lambda(t_j)(t_j - t_{j-1}) \mathbb{E}_{X_0} \mathbb{E}_{X_{t_j}|X_0} \left[ \left\| \nabla_{x(t_j)} \log p_t(x_i(t_j)|x_0) - s_{t_j,\theta_y}(x_i(t_j)) \right\|_2^2 \right]$$

$$+ \frac{1}{2} \sum_{j=1}^N \lambda(t_j)(t_j - t_{j-1}) C_{t_j}$$

$$= \frac{1}{2} \sum_{j=1}^N \lambda(t_j)(t_j - t_{j-1}) \mathbb{E}_{x(t_j) \sim p_{t_j}} \left[ \left\| s_{t_j,\theta_y}(x(t_j)) - \nabla_x \log p_{t_j}(x(t_j)) \right\|_2^2 \right]$$

For each $y \in \mathcal{Y}$, $\mathcal{L}^y(\theta_y^*)$ is the optimal loss function for the unregularized version under the current hypothesis class. Let $\mathcal{L}^y(\bar{\theta}_y^*)$ be the optimal unregularized loss function under the continuous version of the random feature model. Then,

$$\mathcal{L}^y(\theta_y^*) = \frac{1}{2} \sum_{j=1}^N \lambda(t_j)(t_j - t_{j-1}) \mathbb{E}_{x(t_j) \sim p_{t_j}} \left[ \left\| s_{t_j,\theta_y^*}(x(t_j)) - \nabla_x \log p_{t_j}(x(t_j)) \right\|_2^2 \right] \tag{74}$$

$$\leq 2 \left( \frac{1}{2} \sum_{j=1}^N \lambda(t_j)(t_j - t_{j-1}) \mathbb{E}_{x(t_j) \sim p_{t_j}} \left[ \left\| \bar{s}_{t,\bar{\theta}_y^*}(x(t)) - \nabla_x \log p_t(x(t)) \right\|_2^2 \right] \tag{75}$$

$$+ \frac{1}{2} \sum_{j=1}^N \lambda(t_j)(t_j - t_{j-1}) \mathbb{E}_{x(t_j) \sim p_{t_j}} \left[ \left\| \bar{s}_{t_j,\bar{\theta}_y^*}(x(t_j)) - s_{t_j,\theta_y^*}(x(t_j)) \right\|_2^2 \right] \right) \tag{76}$$

$$\leq 2 \mathcal{L}^y(\bar{\theta}_y^*) + Err_{MC}(\theta_y^*, \bar{\theta}_y^*; \{t_j\}_{j=1}^N, \{\lambda(t_j)\}_{j=1}^N) \tag{77}$$

**Proposition 2.** *Monte Carlo estimates. Define the Monte Carlo error*

$$Err_{MC}(\theta, \bar{\theta}, \{t_j\}_{j=1}^N, \{\lambda(t_j)\}_{j=1}^N) := \sum_{j=1}^N \lambda(t_j)(t_j - t_{j-1}).$$
$$\mathbb{E}_{x(t_j) \sim p_{t_j}} \left[ \left\| \bar{s}_{t_j,\bar{\theta}_y^*}(x(t_j)) - s_{t_j,\theta_y^*}(x(t_j)) \right\|_2^2 \right] \tag{78}$$

*Suppose that $\|X(0)\|_\infty \leq K$ and the trainable parameter $a$ and embedding functions $W, U, e(.)$ are both bounded. Then. given any $\bar{\theta}$. for any $\delta > 0, \delta \ll 1$, with probability of at least $1 - 2N\delta$, there exists $\theta$ such that*

$$Err_{MC}(\theta, \bar{\theta}, \{t_j\}_{j=1}^N, \{\lambda(t_j)\}_{j=1}^N) \leq \frac{2C_{w,u}^2 B^2 (C_{t_N,\delta} + C_{t_N.e})^2 d^2}{m} \log\left(\frac{2}{\delta}\right) \sum_{j=1}^N \lambda(t_j)(t_j - t_{j-1}) \tag{79}$$

**Proof.** The proof closely along the line of [17]. Fix any $\bar{\theta}$. For notational convenience, we will drop $y$ from $\theta_y$ and $\bar{\theta}_y$. For $k = 1, 2, \cdots d$, define

$$Z_{t,k}(W, U) := \left\| s_{t,\theta,k}(x) - \bar{s}_{t,\bar{\theta},k}(x) \right\|_{L^2(p_t)} = \mathbb{E}_{x \sim p_t}^{1/2} \left[ |s_{t,\theta,k}(x) - \bar{s}_{t,\bar{\theta},k}(x)|^2 \right] \tag{80}$$

$$= \mathbb{E}_{x \sim p_t} \left[ \left| \frac{1}{m} \sum_{i=1}^m a_{i,k} \sigma(w_i^T x + u_i^T e(t)) - \mathbb{E}_{(w,u)} [a_k(w,u) \sigma(w^T x + u^T e(t))] \right|^2 \right] \tag{81}$$

Then, we have

$$\mathbb{E}_{x \sim p_t} \left[ \left\| s_{t,\theta_y}(x) - \bar{s}_{t,\bar{\theta}_y}(x) \right\|_2^2 \right] = \sum_{k=1}^{d} \mathbb{E}_{x \sim p_t} \left[ |s_{t,\theta_y,k}(x) - \bar{s}_{t,\bar{\theta}_y,k}|^2 \right] \tag{82}$$

$$= \sum_{k=1}^{d} Z_{t,k}^2(W,U)$$

$$\leq \sum_{k=1}^{d} \left( |Z_{t,k}(W,U) - \mathbb{E}_{W,U}[Z_{t,k}]| + |\mathbb{E}_{W,U}[Z_{t,k}(W,U)]| \right)^2$$

$$\overset{(a)}{\leq} 2 \sum_{k=1}^{d} \left( |Z_{t,k}(W,U) - \mathbb{E}_{W,U}[Z_{t,k}(W,U)]|^2 \right.$$

$$\left. + \mathbb{E}_{W,U}[Z_{t,k}^2(W,U)] \right) \tag{83}$$

where $(a)$ follows from the fact that $(a+b)^2 \leq 2(a^2+b^2)$ and Jensen's Inequality $\mathbb{E}^2[Z_{t,k}(W,U)] \leq \mathbb{E}_{W,U}[Z_{t,k}^2(W,U)]$. According to Lemma 1. for any $\delta > 0, \delta \ll 1$, w.p. atleast $1 - \delta$, we have

$$\|x(t)\|_\infty \leq C_{t_N,\delta} \tag{84}$$

If $(\tilde{W}, \tilde{U})$ is different from $(W, U)$ at only one component indexed by $i$, we have w.p. $1 - \delta$

$$|Z_{t,k}(W,U) - Z_{t,k}(\tilde{W},\tilde{U})| \tag{85}$$

$$= \left| \left\| s_{t,\theta,k}(x) - \bar{s}_{t,\bar{\theta},k}(x) \right\|_{L^2(p_t)} - \left\| s_{t,\tilde{\theta},k}(x) - \bar{s}_{t,\bar{\theta},k}(x) \right\|_{L^2(p_t)} \right| \tag{86}$$

$$\overset{(a)}{\leq} \left\| s_{t,\tilde{\theta},k}(x) - s_{t,\theta,k}(x) \right\|_{L^2(p_t)} \tag{87}$$

$$= \frac{1}{m} \left\| a_{i,k}\sigma(w_i^T x + u_i^T e(t)) - \tilde{a}_{i,k}\sigma(\tilde{w}_i^T x + \tilde{u}_i^T e(t)) \right\|_{L^2(p_t)} \tag{88}$$

$$\overset{(b)}{\leq} \frac{1}{m} \left( |a_{i,k}| \left\| \sigma(w_i^T x + u_i^T e(t)) \right\|_{L^2(p_t)} + |\tilde{a}_{i,k}| \left\| \sigma(\tilde{w}_i^T x + \tilde{u}_i^T e(t)) \right\|_{L^2(p_t)} \right) \tag{89}$$

$$\overset{(c)}{\leq} \frac{1}{m} \left( |a_{i,k}| \left\| w_i^T x + u_i^T e(t) \right\|_{L^2(p_t)} + |\tilde{a}_{i,k}| \left\| (\tilde{w}_i^T x + \tilde{u}_i^T e(t)) \right\|_{L^2(p_t)} \right) \tag{90}$$

$$\overset{(d)}{\leq} \frac{1}{m} \left( |a_{i,k}|(\|w_i\|_1 C_{t_N,\delta} + \|u_i\|_1 \|e(t)\|_\infty) + |\tilde{a}_{i,k}|(\|\tilde{w}_i\|_1 C_{t_N,\delta} + \|\tilde{u}_i\|_1 \|e(t)\|_\infty) \right) \tag{91}$$

$$\overset{(e)}{\leq} \frac{2}{m} B C_{w,u}(C_{t_N,\delta} + C_{t_N,e}) \tag{92}$$

where $(a)$ and $(b)$ follows from triangle inequality $| \|a\| - \|b\| | \leq \|a - b\|$ and $\|a - b\| \leq \|a\| + \|b\|$, $(c)$ follows from the fact that $|\sigma(y)| \leq |y|$, $(d)$ follows from Lemma 1 and Holder Inequality, $(e)$ follows from the bounds on $\|w_i\|_1, \|u_i\|_1, x, |a_{i,k}|, e(t_j)$.

Thus, $w, p. 1 - \delta$, $Z_{t,k}(W,U)$ has bounded increment property. Using McDiarmid's inequality, $w.p. 1 - 2\delta$, we have

$$|Z_{t,k}(W,U) - \mathbb{E}_{W,U}[Z_{t,k}(W,U)]| \leq \frac{B}{m} C_{w,u}(C_{t_N,\delta} + C_{t_N.e}) \sqrt{d \log\left(\frac{2}{\delta}\right)} \tag{93}$$

Now we compute

$$\mathbb{E}_{W,U}[Z_{t,k}^2(W,U)]$$

$$=\mathbb{E}_{W,U}\left[\mathbb{E}_{x\sim p_t}[|s_{t,\theta,k}(x)-\bar{s}_{t,\bar{\theta},k}(x)|^2]\right]$$

$$\overset{(b)}{=}\mathbb{E}_{x\sim p_t}\left[\mathbb{E}_{W,U}[|s_{t,\theta,k}(x)-\bar{s}_{t,\bar{\theta},k}(x)|^2]\right]$$

$$=\frac{1}{m^2}\mathbb{E}_{x\sim p_t}\left[\mathbb{E}_{W,U}\left[\left|\left|\sum_{i=1}^{m}\left(a_{i,k}\sigma(w_i^T x+u_i^T e(t))-\mathbb{E}_{w,u}[a_k(w,u)\sigma(w^T x+u^T e(t))]\right)\right|\right|^2\right]\right]$$

$$+\frac{1}{m^2}\mathbb{E}_{x\sim p_t}\left[\mathbb{E}_{W,U}\left[\sum_{i\neq j}(a_{i,k}\sigma(w_i^T x+u_i^T e(t))-\mathbb{E}_{w,u}[a_k(w,u)\sigma(w^T x+u^T e(t))])\right.\right.$$

$$\left.\left.\times\,(a_{j,k}\sigma(w_j^T x+u_j^T e(t))-\mathbb{E}_{w,u}[a_k\sigma(w^T x+u^T e(t))])\right]\right]$$

$$\overset{(c)}{=}\frac{1}{m^2}\mathbb{E}_{x\sim p_t}\left[\mathbb{E}_{W,U}\left[\sum_{i=1}^{m}\left(a_{i,k}\sigma(w_i^T x+u_i^T e(t))-\mathbb{E}_{w,u}[a_k(w,u)\sigma(w^T x+u^T e(t))]\right)^2\right]\right]$$

$$+\frac{1}{m^2}\mathbb{E}_{x\sim p_t}\left[\sum_{i\neq j}\mathbb{E}_{w_i,u_i}\left[(a_{i,k}\sigma(w_i^T x+u_i^T e(t))-\mathbb{E}_{w,u}[a_k(w,u)\sigma(w^T x+Ue(t))])\right.\right.$$

$$\left.\left.\times\,\mathbb{E}_{w_j,u_j}\left[(a_{j,k}\sigma(w_j^T x+u_j^T e(t))-\mathbb{E}_{w,u}[a_k(w,u)\sigma(w^T x+Ue(t))])\right]\right]\right]$$

$$\overset{(d)}{=}\frac{1}{m^2}\mathbb{E}_{x\sim p_t}\left[\sum_{i=1}^{m}\mathbb{E}_{W,U}\left[\left(a_{i,k}\sigma(w_i^T x+u_i^T e(t))-\mathbb{E}_{w,u}[a_k(w,u)\sigma(w^T x+u^T e(t))]\right)^2\right]\right]$$

$$\overset{(e)}{\leq}\frac{1}{m}\mathbb{E}_{x\sim p_t}\left[\mathbb{E}_{w,u}\left[(a_k(w,u)\sigma(w^T x+u^T e(t)))^2\right]\right]$$

$$\overset{(f)}{\leq}\frac{1}{m}\mathbb{E}_{x\sim p_t}\left[\mathbb{E}_{w,u}\left[(|a_{y,k}(w,u)|(\|w\|_1 C_{t_N,\delta}+\|u\|_1\|e(t)\|_\infty))^2\right]\right]$$

$$\leq\frac{1}{m}(C_{t_N,\delta}+C_{t_N,e})^2 C_{w,u}^2 B^2$$

where $(b)$ is due to Fubini's theorem, $(c)$ is due to independence of sampling $(w_i,u_i)$ and $(w_j,u_j)$, $(d)$ is due to $a_{j,k}\sigma(w_j^T x+u_j^T e(t))$ being an unbiased estimator of the continuous version of score network, $(e)$ follows from $Var(X)\leq\mathbb{E}[X^2]$, $(f)$ follows from $|\sigma(y)|\leq|y|$ and Holder's inequality. Thus. $w.p.\,1-2\delta$,

$$\mathbb{E}_{x\sim p_t}\left[\left\|s_{t,\theta_y}(x)-\bar{s}_{t,\bar{\theta}_y}(x)\right\|_2^2\right]\leq\frac{2C_{w,u}^2 B^2(C_{t_N,\delta}+C_{t_N.e})^2 d^2}{m}\log\left(\frac{2}{\delta}\right)\tag{94}$$

Finally, we have $w.p.\,1-2N\delta$

$$Err_{MC}(\theta,\bar{\theta},\{t_j\}_{j=1}^N,\{\lambda(t_j)\}_{j=1}^N)\leq\frac{2C_{w,u}^2 B^2(C_{t_N,\delta}+C_{t_N.e})^2 d^2}{m}\log\left(\frac{2}{\delta}\right)\sum_{j=1}^N\lambda(t_j)(t_j-t_{j-1})\tag{95}$$

## B.8 Radamacher Complexity

In this section, we will bound the term related to the generalization bound

$$\sup_{(\theta_y,\theta_{-y})}|\,\mathcal{L}_{reg}^y(\theta_y,\theta_{-y})-\bar{\mathcal{L}}_{reg}^{n_y}(\theta_y,\theta_{-y})\,|\tag{96}$$

The Rademacher complexity of a real valued function class $\mathcal{F}$ is defined as:

$$\mathcal{R}_n(\mathcal{F}) := \mathbb{E}_{x_1,\cdots,x_n}\mathbb{E}_{\sigma_1,\cdots,\sigma_n}\left[\sup_{f\in\mathcal{F}}\frac{1}{n}\sum_{i=1}^{n}\sigma_i f(x_i)\right], \tag{97}$$

The variables $\sigma_1,\cdots,\sigma_m$ are iid Bernoulli random variables that take values $\{+1,-1\}$ with equal probability and are independent of $x_1,\cdots,x_m$. However, for our random feature model, we have a vector valued function class

$$\hat{\mathcal{F}}_{W,U} := \left\{f(x) = \frac{A}{m}\Phi(x,W,U) = \frac{1}{m}\sum_{k=1}^{m}\alpha_k\phi(x,w_k,u_k)\,\middle|\,\|A\|_F \leq B\right\} \tag{98}$$

**Theorem 4.** *[19, Theorem 3] Let $X$ be nontrivial, symmetric and subgaussian. Then there exists a constant $C < \infty$, depending only on the distribution of $X$, such that for any countable set $S$ and functions $\psi_i : S \to \mathbb{R}, \phi_i : S \to l_2, 1 \leq i \leq n$ satisfying*

$$\forall s, s' \in S, \psi_i(s) - \psi(s'_i) \leq \|\phi_i(s) - \phi_i(s')\| \tag{99}$$

*we have*

$$\mathbb{E}\sup_{s\in S}\sum_i \epsilon_i\psi_i(s) \leq C\mathbb{E}\sup_{s\in S}\sum_{i,k} X_{ik}\phi_i(s)_k \tag{100}$$

*where the $X_{ik}$ are independent copies of $X$ for $1 \leq i \leq n$ and $1 \leq k \leq \infty$ and $\phi_i(s)_k$ is the k-th coordinate of $\phi_i(s)$. If $X$ is a Rademacher variable we may choose $C = \sqrt{2}$, if $X$ is a standard normal $C = \sqrt{\frac{\pi}{2}}$.*

**Corollary 2.** *[19, Corollary 4] Let $\mathcal{X}$ be any set, $(x_1,\cdots,x_n) \in \mathcal{X}^n$, let $F$ be a class of functions $f : \mathcal{X} \to l_2$ and let $h_i : l_2 \to \mathbb{R}$ have Lipschitz norm $L$. Then*

$$\mathbb{E}\sup_{f\in F}\sum_i \epsilon_i h_i(f(x_i)) \leq \sqrt{2}L\mathbb{E}\sup_{f\in F}\sum_{i,k}\epsilon_{ik}f_k(x_i) \tag{101}$$

*where $\epsilon_{ik}$ is an independent doubly indexed Rademacher sequence and $f_k(x_i)$ is the k-th component of $f(x_i)$.*

**Lemma 5.** *[19] Consider the function class $\mathcal{F} = \{x \to \frac{A}{m}\phi(x,W,U) : A \in \mathcal{B}(H,\mathbb{R}), \|A\|_F \leq B\}$. Then the empirical Rademacher complexity of $F$ is*

$$\hat{Rad}_n(\mathcal{F}) = \mathbb{E}\sup_{f\in F}\sum_{i,k}\epsilon_{ik}f_k(x_i) \leq \frac{B}{\sqrt{m}}\sqrt{\sum_i\|\phi(x,W,U)\|^2} \tag{102}$$

*Moreover, if $\mathbb{E}_x\|\phi(x,W,U)\|^2 \leq C^2$, the Rademacher Complexity of $\mathcal{F}$ is*

$$\mathcal{R}_n(\mathcal{F}) \leq \frac{BC}{\sqrt{mn}} \tag{103}$$

**Proof:**

$$\hat{Rad}_n(\mathcal{F}) = \mathbb{E}\sup_{f\in F}\sum_{i,k}\epsilon_{ik}f_k(x_i) = \frac{1}{m}\mathbb{E}\sup_{\|A\|_F\leq B}\sum_k\langle a_k, \sum_i\epsilon_{ik}x_i\rangle \tag{104}$$

$$= \frac{1}{m}\mathbb{E}\sup_{\|A\|_F\leq B}tr(D^*A) \leq B\mathbb{E}\|D^*\|_* \tag{105}$$

where $D \in \mathcal{B}(H,\mathbb{R}^K)$ is the random transformation

$$v \to \left(\langle v, \sum_i\epsilon_{i1}x_i\rangle,\cdots,\langle v, \sum_i\epsilon_{iK}x_i\rangle\right) \tag{106}$$

Thus,

$$\mathbb{E}\|D^*\|_* = \mathbb{E}\sqrt{\sum_m\left\|\sum_i\epsilon_{ik}\phi(x_i,W,U)\right\|^2} \leq \sqrt{m\sum_i\|\phi(x_i,W,U)\|^2} \tag{107}$$

Thus,

$$\mathcal{R}_n(\mathcal{F}) = \mathbb{E}_{x_1,\cdots,x_n} \frac{1}{n} \hat{Rad}_n(\mathcal{F}) \leq \frac{B}{\sqrt{mn}} \mathbb{E}_{x_1,\cdots,x_n} \sqrt{\sum_i \|\phi(x_i, W, U)\|^2} \tag{108}$$

$$\leq \frac{B}{n\sqrt{m}} \sqrt{\sum_i \mathbb{E}_{x_1,\cdots,x_n} \|\phi(x_i, W, U)\|^2} \tag{109}$$

$$\leq \frac{BC}{\sqrt{mn}} \tag{110}$$

Suppose $0 < t_1 < \cdots < t_N = T$ are the chosen points of discretization for training, we have from the forward process

$$X(t) = \qquad\qquad e^{-t}X(0) + \sqrt{1 - e^{-2t}}Z, Z \sim N(0, 1) \tag{111}$$

$$\Longrightarrow \mathbb{E}_Z[X^2(t)] = \qquad\qquad e^{-2t}x^2(0) + \frac{1 - e^{-2t}}{2} \tag{112}$$

$$\Longrightarrow \mathbb{E}_{X(0)}\mathbb{E}_Z[X^2(t_j)] \leq \qquad\qquad K^2 + \frac{1 - e^{-2T}}{2}, \forall 0 < t_j < T \tag{113}$$

Using the above bounds along with bounded support of embedding matrices $W, U$ and embedding function $e(t)$ and Assumption 1, it is easy to show that

$$\mathbb{E}_{X_0}\mathbb{E}_{\xi_j}\left[ \|\sigma(Wx(t) + Ue(t))\|_2^2 \right] \leq F_T^2, \forall 0 < t \leq T \tag{114}$$

for some constant $F_T^2$ and $x(t) = e^{-t}x(0) + \sqrt{1 - e^{-2t}}\xi_j, \xi_j \sim \mathcal{N}(0, I)$

**Lemma 6.** *The term*

$$\mathcal{L}_{mut}^y(\theta_y, \theta_{-y}) = \sum_{j=1}^N \omega(t_j)(t_j - t_{j-1})\mathbb{E}_{X_0}\mathbb{E}_{X_{t_j}|X_0}\mathbb{E}_{y' \sim Q}\left[ \left\| s_{t_j, \theta_y}(x_i(t_j)) - s_{t, \theta_{y'}}(x_i(t_j)) \right\|_2^2 \right] \tag{115}$$

*is* $\mathcal{O}\left( F_T B \sum_{j=1}^N \omega(t_j)(t_j - t_{j-1}) \right)$

**Proof:** Using the fact of bounded support of embedding matrices $W, U$ and embedding function $e(t)$, bounded strategy space and Assumption 1 and eq 114, we get the desired bounded.

**Lemma 7.** *Suppose* $L_{C_1} = \bar{\sigma}_{t_j}^2 BF_T + \sqrt{d}\sqrt{\log \frac{2}{\pi\delta^2}})$. *Then, with probability* $1 - \delta$, *the function* $h : \mathcal{A} \subset \mathbb{R}^d \to \mathbb{R}$

$$h(x) = \left\| \bar{\sigma}_{t_j}x + \xi_{ij} \right\|^2 \tag{116}$$

*is Lipschitz in* $x$, *where* $\mathcal{A} = \{x \in \mathbb{R}^d : \|x\|_2 \leq F_T B\}$.

**Proof.** It is sufficient to show the norm of the gradient of $h(x)$ is bounded for $x \in \mathcal{A}$. With probability $1 - \delta$,

$$\|\nabla_x h(x)\|_2 = \bar{\sigma}_{t_j} \left\| \bar{\sigma}_{t_j}x + \xi_{ij} \right\|_2 \leq \bar{\sigma}_{t_j}^2 F_T B + \sqrt{d}\sqrt{\log \frac{2}{\pi\delta^2}}) \tag{117}$$

$$\tag{118}$$

**Lemma 8.** *Suppose* $L_{C_2} = 2F_T B|\mathcal{Y}|$. *Define* $g : \mathcal{A}^{\mathcal{Y}} \subset \mathbb{R}^{d|\mathcal{Y}|} \to \mathbb{R}$ *where*

$$g(x_1, x_2, \cdots, x_{|\mathcal{Y}|}) = \mathbb{E}_{y'}[\|x_i - x_{y'}\|^2], y' \in \{1, 2, \cdots, |\mathcal{Y}|\} - i \tag{119}$$

*is Lipschitz in* $x$, *where* $\mathcal{A} = \{x \in \mathbb{R}^d : \|x\|_2 \leq F_T B\}$.

**Proof:**

$$\nabla_{x_i} g(x_1, x_2, \cdots, x_{|\mathcal{Y}|}) = 2\mathbb{E}_{y'}[(x_i - x_{y'})] \tag{120}$$

$$\nabla_{x_j} g(x_1, x_2, \cdots, x_{|\mathcal{Y}|}) = 2p(x_j)(x_j - x_i), j \neq i \tag{121}$$

$$\|\nabla_x g(x)\| \leq \left\|\nabla_{x_i} g(x_1, x_2, \cdots, x_{|\mathcal{Y}|})\right\| + \sum_{j \neq i} \left\|\nabla_{x_j} g(x_1, x_2, \cdots, x_{|\mathcal{Y}|})\right\| \tag{122}$$

$$\leq 2\mathbb{E}_{y'}[\|x_i - x_{y'}\|] + 2\sum_{k \neq i} \|x_k - x_i\| \leq 2F_T B|\mathcal{Y}| \tag{123}$$

We know

$$\mathcal{L}^y(\theta_y) = \frac{1}{2} \sum_{j=1}^{N} \frac{\lambda(t_j)(t_j - t_{j-1})}{\bar{\sigma}_{t_j}} \mathbb{E}_{X_0} \mathbb{E}_{\xi_j} \left[ \left\| \bar{\sigma}_{t_j} s_{t_j, \theta_y}(x(t_j)) + \xi_j \right\|_2^2 \right] \tag{124}$$

$$+ \frac{1}{2} \sum_{j=1}^{N} \lambda(t_j)(t_j - t_{j-1}) C_{t_j}(y) \tag{125}$$

where $C_t(y) = \mathbb{E}_{X_t} \|\nabla \log p_t(.|y)\|^2 - \mathbb{E}_{X_0} \mathbb{E}_{X_t|X_0} \|\nabla \log p_t(x_t|x_0, y)\|^2$. Let $\bar{C}(y) = \frac{1}{2} \sum_{j=1}^{N} \lambda(t_j)(t_j - t_{j-1}) C_{t_j}(y)$

**Lemma 9.** *With probability* $1 - Nn_y\delta$, *an upper bound for the generalization gap i.e.*

$$\sup_{(\theta_y, \theta_{-y})} |\mathcal{L}^y_{reg}(\theta_y, \theta_{-y}) - \bar{\mathcal{L}}^{n_y}_{reg}(\theta_y, \theta_{-y})| \tag{126}$$

*is*

$$\frac{2\sqrt{2}BF_T}{\sqrt{mn_y}} L_{C_1} \sum_{j=1}^{N} \frac{\lambda(t_j)(t_j - t_{j-1})}{\bar{\sigma}_{t_j}} + \frac{2\sqrt{2}BF_T|\mathcal{Y}|^2}{\sqrt{mn_y}} L_{C_2} \sum_{j=1}^{N} \omega(t_j)(t_j - t_{j-1}) + \bar{\mathcal{C}} \tag{127}$$

*where* $L_{C_1} = \bar{\sigma}_{t_j}^2 BF_T + \sqrt{d}\sqrt{\log \frac{2}{\pi\delta^2}}), L_{C_2} = 2F_T B|\mathcal{Y}|, \bar{\mathcal{C}} = \max_{y \in \mathcal{Y}} |\bar{C}(y)|$

**Proof.** Observe that, we can rewrite Eq. 126 using triangle inequality as

$$\sup_{(\theta_y, \theta_{-y})} |\mathcal{L}^y_{reg}(\theta_y, \theta_{-y}) - \bar{\mathcal{L}}^{n_y}_{reg}(\theta_y, \theta_{-y})| \leq \sup_{\theta_y} |\mathcal{L}^y(\theta_y) - \bar{\mathcal{L}}^{n_y}(\theta_y)|$$

$$+ \beta \sup_{(\theta_y, \theta_{-y})} |\mathcal{L}^y_{mut}(\theta_y, \theta_{-y}) - \bar{\mathcal{L}}^{n_y}_{mut}(\theta_y, \theta_{-y})| \tag{128}$$

Further decomposing them, we get

$$\sup_{\theta_y} |\mathcal{L}^y(\theta_y) - \bar{\mathcal{L}}^{n_y}(\theta_y)| \leq \sum_{j=1}^{N} \frac{\lambda(t_j)(t_j - t_{j-1})}{\bar{\sigma}_{t_j}} \sup_{\theta_y} |\mathcal{L}^y(\theta_y)(j) - \bar{\mathcal{L}}^{n_y}(\theta_y)(j)| + \bar{\mathcal{C}} \tag{129}$$

where $|\mathcal{L}^y(\theta_y)(j) - \bar{\mathcal{L}}^{n_y}(\theta_y)(j)| = | \frac{1}{2n_y} \sum_{i=1}^{n_y} \left[ \left\| \bar{\sigma}_{t_j} s_{t_j, \theta_y}(x_i(t_j)) + \xi_{ij} \right\|_2^2 - \mathbb{E}_{X_0} \mathbb{E}_{\xi_j} \left\| \bar{\sigma}_{t_j} s_{t_j, \theta_y}(e^{-t_j} X_0 + \sqrt{1 - e^{-2t_j}}\xi_j) + \xi_j \right\|^2 \right] |$ and

$$\sup_{(\theta_y, \theta_{-y})} |\mathcal{L}^y_{mut}(\theta_y, \theta_{-y}) - \bar{\mathcal{L}}^{n_y}_{mut}(\theta_y, \theta_{-y})| \leq$$

$$\sum_{j=1}^{N} \omega(t_j)(t_j - t_{j-1}) \sup_{(\theta_y, \theta_{-y})} |\mathcal{L}^y_{mut}(\theta_y, \theta_{-y})(j) - \bar{\mathcal{L}}^{n_y}_{mut}(\theta_y, \theta_{-y})(j)| \tag{130}$$

where

$$|\mathcal{L}_{mut}^y(\theta_y,\theta_{-y})(j) - \bar{\mathcal{L}}_{mut}^{n_y}(\theta_y,\theta_{-y})(j)| = | \frac{1}{2n_y} \sum_{i=1}^{n_y} \left[ \mathbb{E}_{y'\sim Q} \left[ \left\| s_{t_j,\theta_y}(x_i(t_j)) - s_{t,\theta_{y'}}(x_i(t_j)) \right\|_2^2 \right. \right.$$

$$\left. \left. - \mathbb{E}_{X_0} \mathbb{E}_{\xi_j} \mathbb{E}_{y'\sim Q} \left[ \left\| s_{t_j,\theta_y}(e^{-t_j}X_0 + \sqrt{1-e^{-2t_j}}\xi_j) - s_{t_j,\theta_{y'}}(e^{-t_j}X_0 + \sqrt{1-e^{-2t_j}}) \right\|^2 \right] \right] |$$

(131)

Finally using Corollary 2, Lemmas 5 7,8, [19, Section 4.1] we have

$$\sup_{\theta_y} |\mathcal{L}^y(\theta_y) - \bar{\mathcal{L}}^{n_y}(\theta_y)| \le \frac{2\sqrt{2}BF_T}{\sqrt{mn_y}} L_{C_1} \sum_{j=1}^N \lambda(t_j)(t_j - t_{j-1}) + \bar{\mathcal{C}}$$

(132)

and

$$\sup_{(\theta_y,\theta_{-y})} |\mathcal{L}_{mut}^y(\theta_y,\theta_{-y}) - \bar{\mathcal{L}}_{mut}^{n_y}(\theta_y,\theta_{-y})| \le \frac{2\sqrt{2}BF_T|\mathcal{Y}|^2}{\sqrt{mn_y}} L_{C_2} \sum_{j=1}^N \omega(t_j)(t_j - t_{j-1})$$

(133)

## C  Numerical Experiments

**Computing resources.**   The numerical experiments were conducted on a MacBook Air (2023) and Gilbreth. Gilbreth has heterogeneous hardware comprising of Nvidia V100, A100, A10, and A30 GPUs in separate sub-clusters. All the nodes are connected by 100 Gbps Infiniband interconnects. We used sub-cluster B with 16 nodes, 24 cores per node, 192 GB memory per node, 3 A30 (24 GB) per node. For more information follow this link.

### C.1  Gaussian Mixture Models

**Dataset** We perform empirical experiments on synthetic datasets to verify our theoretical findings. The synthetic dataset is randomly generated under the true distribution and fixed. We detail out the underlying distribution on a case by case basis.

**Implementation Details** We employ the random feature model with the width of network $m = 16$, learning rate $\eta_\tau = 10^{-4}, \forall \tau, T_{train} = 5000$ is fixed for Adam optimizer. We set $\lambda(t) = \bar{\sigma}_t, \omega(t) = e^t$, total number of training samples is 50.

**Case one**   We perform more empirical experiments on $d = 1$, imbalance ratio $r = 2.5, \beta = 0.01$. We compute the KL-divergence between the ground truth distribution and the learned model using the procedure in [17]. $P(x|y=1) \sim \mathcal{N}(-\mu, \sigma^2)$ and class 2 is $P(x|y=2) \sim \mathcal{N}(\mu, \sigma^2)$. We observe Fig. 2 the worst case KL divergence for the mutual learning case is lower than the vanilla when we change the distance between mean and the variance of each class label. The performance of head class doesn't worsened for small $\mu$. However, the head class performance suffers for mutual learning case when the distance between the mean increases. This might be because when the support of class distribution are farther apart mutual learning is not advantageous as transfer of knowledge between the class is not useful.

**Case two**   We now consider a case with two classes with imbalance ratio $r = 2.5, \beta = 0.01$. Class 1 itself is a uniform mixture of two Gaussian i.e $P(x|y=1) \sim \frac{1}{2}\mathcal{N}(-4,3) + \frac{1}{2}\mathcal{N}(4,3)$ and class 2 is $P(x|y=2) \sim \mathcal{N}(0,2)$ as in Fig. 3. We observe the Mutual Learning objective with our formulation have lower KL-divergence for both the classes compared to the vanilla diffusion models trained on each class. In this case, mutual learning allows useful transfer of knowledge between the classes increasing the performance for both. We hypothesize that under some notion of similarity between various class distributions, mutual learning is advantageous in improving the performance of all classes.

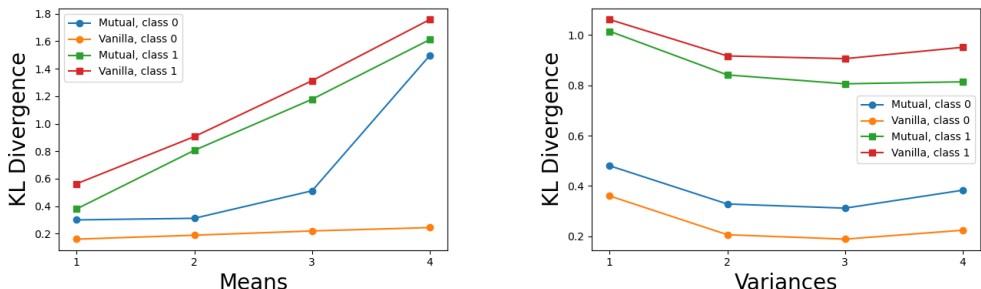

*Figure 2:* Case one: (Left) The first plot shows the KL-divergence for each class with and without mutual learning objective as $\mu$ is varied. (Right) shows the KL-divergence for each class with and without mutual learning objective as $\sigma$ is varied ($\mu = 2$ fixed).

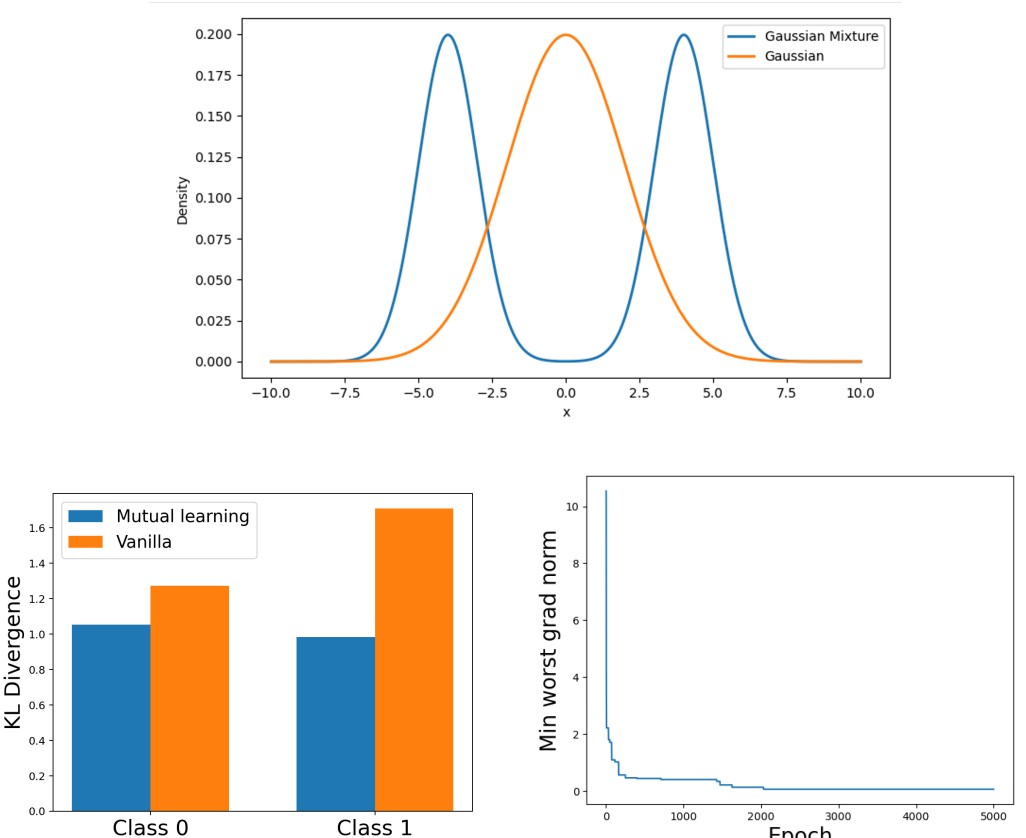

*Figure 3:* Case two: (Top) The first plot shows class 1 as a gaussian mixture with class 2 as Gaussian. (Bottom Left) Shows the KL-divergence for each class with and without mutual learning objective. (Bottom Right) Shows $\min_\tau \max_{y \in \mathcal{Y}} \left\| \nabla \bar{\mathcal{L}}_{reg}^{n_y}(\theta_y^\tau, \theta_{-y}^\tau) \right\|$ decreasing with training epoch.

| Different $\beta$ Values ($\eta = 2 \times 10^{-4}$) | | |
|---|---|---|
| Method | FID($\downarrow$) | IS($\uparrow$) |
| $\beta = 0.0$ | 16.58 | $8.78 \pm 0.15$ |
| $\beta = 0.1$ | 18.61 | $8.94 \pm 0.10$ |
| $\beta = 1.0$ | 16.74 | $8.55 \pm 0.21$ |

| Different $\eta$ Values ($\beta = 0.1$) | | |
|---|---|---|
| Method | FID($\downarrow$) | IS($\uparrow$) |
| $\eta = 2 \times 10^{-4}$ | 18.61 | $8.94 \pm 0.10$ |
| $\eta = 10^{-4}$ | 14.58 | $8.92 \pm 0.19$ |
| $\eta = 10^{-5}$ | 18.62 | $8.62 \pm 0.21$ |

*Figure 4:* FID and IS Scores for Different $\beta$ and $\eta$ Values

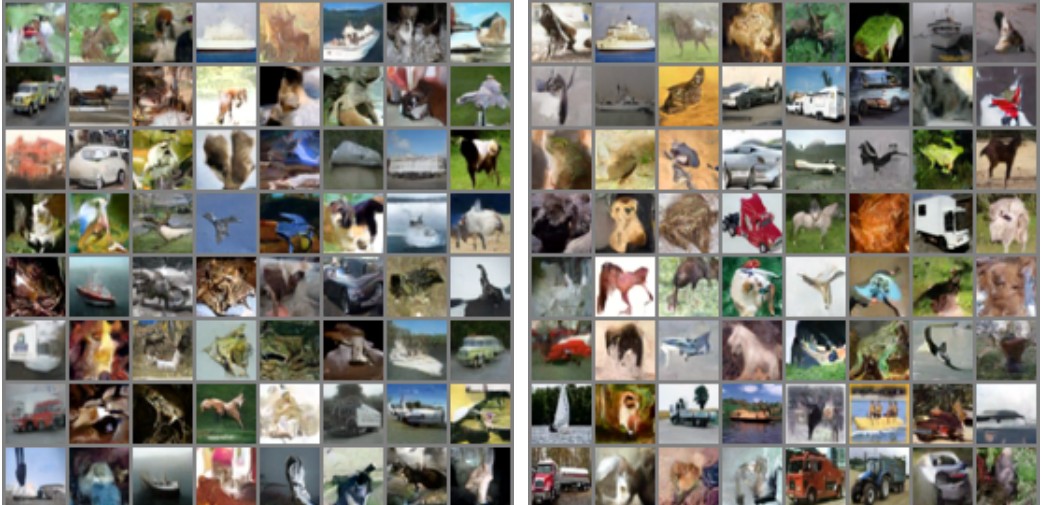

*Figure 5:* Visualization of image generated from Vanilla DDPM ($\beta = 0$) (Left) and Mutual Learning (Right)

## C.2 Experiments on CIFAR10LT

In this section, we present the numerical results for varying hyperparamters $\eta$ and $\beta$ values. Furthermore, for completeness, we provide visualization of the images generated from various methods.

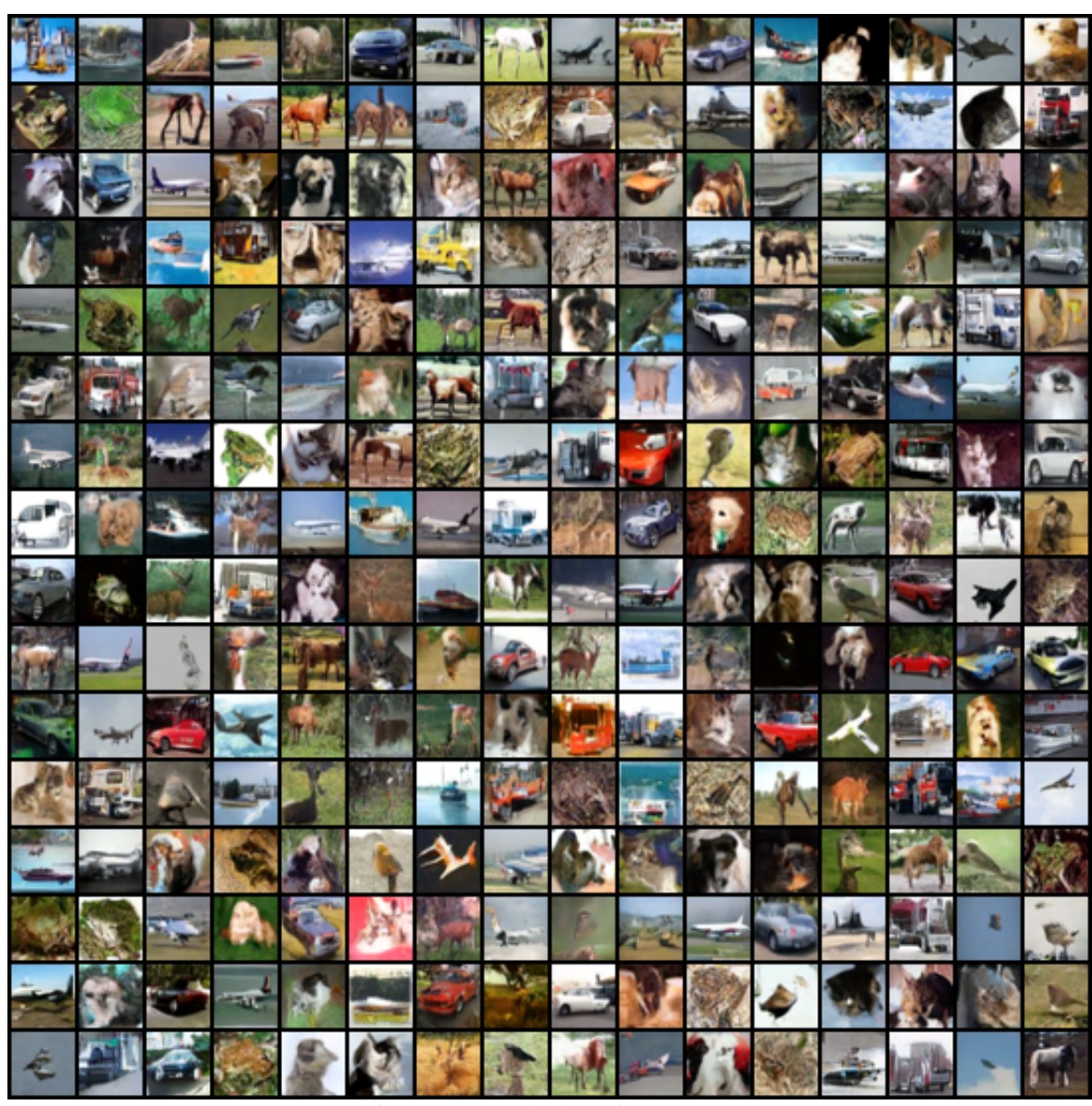

*Figure 6:* Image Visualization of CBDM

