# OpenReview forum: "Principled Long-Tailed Generative Modeling via Diffusion Models"
_NeurIPS.cc/2025/Conference — NeurIPS 2025 poster_

### Official Review · Reviewer_tuny · 2025-06-29

**Clarity:** 2
**Significance:** 1
**Originality:** 1
**Rating:** 3
**Confidence:** 4

**Summary:**

This paper addresses the challenge of training diffusion models on real-world long-tailed datasets where rare classes are underrepresented.
It proposes a novel regularized training objective that combines standard diffusion loss with a mutual learning term, formulated as a multi-player game with a Nash equilibrium solution. The authors provide theoretical convergence guarantees and bounds on sampling error across all classes, including the tails. The framework offers principled insights for improving fairness, generalization, and performance in imbalanced generative modeling scenarios.

**Questions:**

1. The Nash gap you mentioned is: “The Nash gap of the score network obtained from the algorithm is upper bounded by $O(\sqrt{(1/T)} + β)$, where β is the regularizing parameter.” So wouldn’t it be best when β is 0? Then what is the point of the proposed regularization?

2. I understand that class imbalance is an important issue, but in practice, a more difficult and widespread problem arises when there is dataset bias even in the absence of labels. For example, issues addressed in works like *[ICLR 2024] Training Unbiased Diffusion Models From Biased Datasets*. In such cases, do you think your theoretical analysis could be extended to cover these scenarios as well? If extending to that setting is too difficult, it would be helpful to clearly distinguish your work from those in that line of research in the related work section, for better positioning.

**Ethical Concerns:**

["NO or VERY MINOR ethics concerns only"]

**Final Justification:**

Too limitied empirical results.

**Limitations:**

1. While CBDM is used as the closest baseline, the comparison is lacking. Compare the results on CIFAR10LT, CIFAR100LT, and other datasets from CBDM with metrics such as FID, class-wise FID, precision, and recall. What is the meaning of only measuring worst-case FID?

2. My main concern is the potential side effects introduced by the regularization. The fact that the experimental results focus solely on worst-case FID makes me wonder if this is due to such side effects. A comprehensive comparison across multiple metrics and with relevant baselines is, in my view, the most important point.

**Paper Formatting Concerns:**

None.

**Quality:**

2

**Strengths And Weaknesses:**

**Strength**
- Theoretically Grounded Objective: The paper introduces a novel regularized diffusion objective rooted in mutual learning and game theory, with convergence guarantees toward a Nash equilibrium—providing strong theoretical backing for long-tailed generative modeling.

- Tail-Aware Performance: The method explicitly improves generation quality for underrepresented (tail) classes, balancing sample diversity and fidelity across the entire label distribution, which addresses a core limitation of standard diffusion models.

---

> ### Author Rebuttal · Authors · 2025-07-30
>
> Thank you very much for taking the time to review our paper and provide valuable feedback. We appreciate your constructive comments, which have helped us improve the quality of our work. Your insights and suggestions were truly helpful in strengthening our submission.
>
> Thank you again for your effort and contribution to the review process. Please find our response to your questions
>
> 1. The Nash gap you mentioned is: “The N...
>
> Thank you for giving us a chance to respond to this interesting question. It is important to distinguish the empirical version of the objectives $\bar{\mathcal{L_reg}}^{n_y}(\theta_y,\theta_{-y}),\bar{\mathcal{L}}^{n_y}(\theta_y)$ over which training occurs and the population loss $\mathcal{L}^{y}(\theta_y)$ that determines the sampling (generalization) error. The Individual Gradient Descent is employed to find an $\epsilon-$ Nash Equilibrium of the  $|\mathcal{Y}|$-player game $<  \mathcal{Y},(\bar{\mathcal{L_reg}}^{n_y}(\theta_y,\theta_{-y})),(\Theta_y) >$ where the cost function is given by the empirical regularized objectives. When $\beta=0$ or as $\beta \to 0$, $\bar{\mathcal{L_reg}}^{n_y}(\theta_y,\theta_{-y})\approx \bar{\mathcal{L}}^{n_y}(\theta_y)$ and the algorithm finds $\hat{\theta}_y \in argmin \bar{\mathcal{L}}^{n_y}(\theta_y)$ (no mutual learning). This is the classical DDPM training paradigm.
>
> As present in Theorem 1, in such a case, the empirical training objective will be low and the sampling error will be dominated by generalization gap which is inversely proportional to $n_y$. For tail classes that have low training data frequency, this generalization bound will be large resulting in large sampling error. One might feel this can be overcomed by selecting a score network with more hidden neurons (large $m$), however this increases the training error as shown in Theorem 1.
>     When $\beta$ is large, the mutual learning term enjoys more weight during training and . Hence, there is a sweet spot of $\beta$, not too large, not too small (V shaped curve of Figure 1 (right) in our paper) where employing mutual learning is beneficial.
>
>  As we cannot post images or documents in the rebuttal phase, we refer to reviewers to Figure 2(left) in ``Fast Sampling of Diffusion Models with Exponential Integrator"  Zhang et al 2022 arXiv for a  high level intuition. We will ensure a high level intuition for mutual learning applied to diffusion models is included in future revisions/ versions of our work.
>     Diffusion models are trained to learn the score function $\nabla \log{p_t(x_t|x_0)}$ of the forward process which is then used in the reverse SDE to sample/ generate. As it can be seen from Figure 2 (left) the score model works well in the region where $p_t(x)$ is large but suffers from large error in the region where $p_t(x)$ is small. This observation can be explained by examining the training loss on Figure 2 (right). As explained in the above mentioned paper, since the training data is sampled from $p_t(x,y)$ in regions with a low $p_t(x,y)$ value, the learned score network is not expected to work well due to the lack of training data" Figure 2(left).
>      As a consequence, to ensure $\bar{Y}_0$ is close to $x_0$, one need to make sure $\bar{Y}_t$ (as defined in the paper) stays in the high $p_t(x,y)$ region $\forall t \in [0,T]$. Since the tail classes have very few training data, their (tail classes) score do not lie in this high density region in Figure 2 (left). The mutual learning term aligns the score network for the tail classes with the high confidence scores of head classes (in the high noise region, in other words larger $t$) in a way such that the score network of the tail classes stay in the high probability region during sampling/ generation. The above reasoning also justifies the choice of weighting function $\omega(t)$. For $t\approx 0,  p_t \approx p_0   $, we wish the training objective give more wait to fitting to data distribution while for larger $t\gg0$ the training give more weight to mutual learning objective. This implies $\omega(t)$ should be an increasing function of $t\in [0,T]$. This is employed in "Class-Balancing Diffusion Model", but lacked  proper derivation and justification. In short, mutual learning/ regularization improves the generalization capability of the learned score networks by allowing knowledge transfer and preventing overfitting.
>
> 2. I understand that class imbalance is an important issue,...
>
> Although the reviewer raises a very important issue, based on our reading of [ICLR 2024] ``Training Unbiased Diffusion Models From Biased Datasets", we believe the analysis may not be a straightforward extension of our work. The analysis for the theory of unbiased diffusion models will be in line with the works of ``Evaluating the design space of diffusion-based generative models" Wang et al NeuRIPS 2024". We too draw inspiration from the above mentioned works but deviate considerably in terms of problem formulation and analysis. For unbiased diffusion models, there would need to be consideration of additional error terms that arise as a result of estimation error of time-dependent density ratio $$w^{t}_{\phi^*}(x_t)$$, assumption on $$\nabla log w_{\phi^{*}}^t(x_{t})$$, computation of generalization gap  and other terms based on the error decomposition employed. The fundamental difference will be in that mutual learning head classes augment the learning of tail class score functions my making them be close to their score function for high noise region while for the case of importance re-weighting density ratio estimation, the learning process wants to move away from the score of $ p_{bias} $  and move towards the score of $ p_{data} $. .
>
> 3. My main concern is the potential side effect..
>
> We thank the reviewer for raising this concern. We understand your concern and have performed experiments to address these in the supplementary section. Performing statistical analysis on maximum of a sequence of random variable whose distribution are unknown is challenging. To circumvent this, in Appendix C, we directly compute the KL divergence between the distribution characterized by the learnt score network and the ground truth distribution as in Section 4 in [3]. This allows for direct comparison among class labels and within class labels. As can be observed in the Supplementary Material Appendix C, we have provided several empirical studies to verify our theoretical findings . In case 1 (Appendix C), the performance of the head class goes down when mutual learning is used.  However, the worst case KL divergence is still lower for mutual learning. This is because the head class contains enough training samples to learn a good score network and achieves no performance increase from transfer of knowledge from the tail class. Furthermore, in case 2 when the relative position of the class labels in the embedding space provides additional knowledge (information), mutual learning is beneficial for all classes.    We observe improved performance in both the classes (Figure 3 in Appendix C) when mutual learning is employed vs when it is not employed.
>
>     [3] ``On the generalization properties of diffusion models" Li et al NeuRIPS 2023
>
> 4. We took the code from ``Class Balancing Diffusion Model” Qin. et al 2023 and updated the training procedure according to individual gradient descent. Given the time and resource constraint, we were able to run CBDM for 100000 training steps and individual gradient descent for 10000 steps. To perform evaluation, we generate 500 total samples and make the comparison at the 10000 training step mark. We provide FID, IS across various parameter settings. We will include the link to the github repository for the code and images in future revisions, versions of our paper. We provide  tables summarizing the results.
>
> | Method         | FID     | IS          |
> |----------------|---------|-------------|
> | Vanilla, τ=0 (DDPM) | 423.06 | 1.23 (0.02) |
> | CBDM  | 419.82 | 1.22 (0.01) |
> | Mutual Learning, τ=0.1 | 409.50 | 1.22 (0.01) |
>
> ## Table 2: Different τ Values (lr=2e-4)
> | Method         | FID     | IS          |
> |----------------|---------|-------------|
> | τ=0.0, lr=2e-4 | 423.06 | 1.23 (0.02) |
> | τ=0.1, lr=2e-4 | 415.81 | 1.23 (0.02) |
> | τ=1.0, lr=2e-4 | 421.82 | 1.22 (0.01) |
>
> ## Table 3: Different lr Values (τ=0.1)
> | Method         | FID     | IS          |
> |----------------|---------|-------------|
> | τ=0.1, lr=2e-4 | 415.81 | 1.23 (0.02) |
> | τ=0.1, lr=1e-4 | 409.50 | 1.22 (0.01) |
> | τ=0.1, lr=1e-5 | 442.35 | 1.20 (0.01) |
>
> As seen from the above table, CBDM does better than the vanilla case of DDPM trained on individual class labels and mutual learning does better than the two. Mutual learning doing better than Vanilla case highlights the impact of mutual learning. We were not able to replicate the V-shaped curve as in Fig 1 (right) in our paper for real world datasets due to resource and time constraints.
>
> 5. We also performed to replicate case 2 in our supplementary Apeendix C. Class 0: mixture with means [-2,0][2,0] and sigma=1; class 1:single guassian with mean [0,0] and sigma=2).
> | Method         | FD (Head Class) | FD (Tail Class) |
> |----------------|-----------------|-----------------|
> | IGD (τ=0.0)    | 2.766 ± 0.534   | 1.984 ± 0.321   |
> | IGD (τ=0.001)  | 2.751 ± 0.517   | 1.996 ± 0.317   |
> | IGD (τ=0.01)   | 2.720 ± 0.470   | 1.988 ± 0.297   |
> | IGD (τ=0.1)    | 2.467 ± 0.467   | 1.965 ± 0.287   |
> | IGD (τ=1.0)    | 1.954 ± 0.403   | 2.021 ± 0.370   |
>
> We hope the reviewer finds our response satisfactory. Please let us know during the discussion phase if we can provide further clarification regarding our work, notably it's theoretical contributions, novelty, and significance.

---

> ### Author Response · Authors · 2025-08-05
> **Continuation of the Experiemental Results asked by reviewer**
>
> We thank the reviewers for going through our rebuttal. We continued the experiments asked for by the reviewer for larger training and and evaluation steps for a more thorough and detailed comparison with CBDM and various hyper-parameter choices. The Neural network Architecture employed is U-net as in the CBDM paper. We took the code from ``Class Balancing Diffusion Model” Qin. et al 2023 and updated the training procedure according to individual gradient descent. Given the time and resource constraint, we were able to run CBDM for 100000 training steps and individual gradient descent for 50000 steps on CIFAR10LT. To perform the evaluation, we generate 1500 samples per class label as in CBDM paper and make the comparison at the 50000 training step mark. We provide FID and IS across various parameter settings. We will include the link to the github repository for the code and images in future revisions and versions of our paper. We provide tables summarizing the results below.
>
> **Table 1: Best FID Scores**
>
> | Method                | FID    | IS           |
> |-----------------------|--------|--------------|
> | IGD, τ=0, lr=2e-4     | 19.46  | 8.62 (0.15)  |
> | IGD, τ=0.1, lr=1e-4   | 17.63  | 8.70 (0.17)  |
> | CBDM, lr=2e-4         | 19.10  | 7.92 (0.18)  |
>
> **Table 2: Different τ Values (lr=2e-4)**
>
> | Method                  | FID    | IS           |
> |-------------------------|--------|--------------|
> | IGD, τ=0.0, lr=2e-4     | 19.46  | 8.62 (0.15)  |
> | IGD, τ=0.1, lr=2e-4     | 19.23  | 8.53 (0.21)  |
> | IGD, τ=1.0, lr=2e-4     | 19.96  | 8.50 (0.21)  |
>
> **Table 3: Different lr Values (τ=0.1)**
>
> | Method                  | FID    | IS           |
> |-------------------------|--------|--------------|
> | IGD, τ=0.1, lr=2e-4     | 19.23  | 8.53 (0.21)  |
> | IGD, τ=0.1, lr=1e-4     | 17.63  | 8.70 (0.17)  |
> | IGD, τ=0.1, lr=1e-5     | 22.25  | 8.33 (0.15)  |
>
>
>
> As seen from the above tables, CBDM does better than the vanilla case of DDPM trained on individual class labels and mutual learning does better than both of them. Mutual learning doing better than the Vanilla case highlights the impact of mutual learning. We hope that these experiments clarify the queries raised by the reviewer.
>
> We would like to also respectfully highlight that the paper’s main contribution is of a theoretical nature, which is why we have chosen “theory” as the primary area. We hope the experiments provided are sufficient for a reader to build confidence in our theoretical findings.

---

### Official Review · Reviewer_HV1G · 2025-06-30

**Clarity:** 2
**Significance:** 3
**Originality:** 2
**Rating:** 4
**Confidence:** 3

**Summary:**

The paper proposes using Mutual learning to address the problem of class imbalance (i.e, minimizing the KL can lead to a class with class conditional KL way worst than the class that is more represented) in learning a generative model. It proposes mainly a theoretical analysis of the convergence of the proposed algorithm to the Nash equilibrium for the proposed Game in a simplified setting. Then, it illustrates briefly in a toy example the performance of the model.

**Questions:**

Questions:
1. I don't understand the phrase in lines 157-158 "The above objective is sound when the marginal density of the classes...". As I reckon the above objective is always "valid", but maybe what the authors mean is that they would not lead to an Egalitarian solution?
2. At what point do you really nead to train different networks for each class (suggested by the notation $\theta_{-y}$? This is specially important, because one of  the alternatives of the current method would be to estimate the marginal weights and this is of course difficult in the cases with a large number of classes. But training different networks would be as well.
3. How adding the regularization term helps ? What is the fundamental difference between doing so and just learning one score for each class? It would be interesting to compare to such method  in the numerical section.

Some typos I found:
Typos:
1. But we not (line 188)
2. p_t(x_t | x_0) is always a Gaussian with the forward process defined as previously.
3. Line 177: "To obtain a regularized version of the DM objectinve function" $\textbf{\mathcal{L}^{y}{conti}}$ I suppose.

**Ethical Concerns:**

["NO or VERY MINOR ethics concerns only"]

**Final Justification:**

Strong theoretical work with limited evaluation.

**Limitations:**

yes

**Paper Formatting Concerns:**

It seems that the space between section headers and text has been reduced using a negative vspace, against the recommendation in the original neurips style files ("Do not change any aspects of the formatting parameters in the style files."). I believe that if those reductions would are removed, the paper would go over the 9 page limit.

**Quality:**

3

**Strengths And Weaknesses:**

Strengths:
The paper does a rather complete analysis of the convergence of the proposed method to the Nash equilibrium. Most of the hypothesis used are minimal, with two exception being the depth of the neural net and the way the parameters are defined (which leads to a convex objective), compacity of the support of the data and differentiability of the score, the latter being easily eased by the so called "early stopping" of the backward diffusion. Compacity is another aspect that is restrictive but we might consider it ok in mostly current applications of generative modelling (images for example).

Weaknesses:
The main weakness is the rather lacking numerical experiments section. While I understand that the paper is mainly theoretical, one would still appreciate a comparison with other methods and not only the vanilla one. Also, some of the graphs are not properly explained. Namely what do the bars in Worst FID score represent?

---

> ### Author Rebuttal · Authors · 2025-07-30
>
> Thank you very much for taking the time to review our paper and provide valuable feedback. We appreciate your constructive comments, which have helped us improve the quality of our work. Your insights and suggestions were truly helpful in strengthening our submission.
>
> Thank you again for your effort and contribution to the review process. Please find our response to your questions..
>
> 1. I don't understand the phrase in lines 157-158 ...  We apologize for the confusion. Yes, if the marginal densities of the class labels are not uniformly distributed, training on the total (over all class labels $y\in \mathcal{Y}$) empirical loss will lead to a bias by putting more weight to the head classes during the training process. This will not lead to an egalitarian solution as shown in Figure 1 of   Yiming Qin, et al. “Class-balancing diffusion models.” CVPR 2023. We will clarify this in future versions of our paper.
>
> 2.  At what point do you really need to train......
>
> (a) If the reviewer is asking when the parameters $\theta_{-y}$ are trained within the algorithm, then the answer is that all parameters $\theta_y, y \in \mathcal{Y}$ are  trained simultaneously, as can be seen in the for loop over $y=0,\cdots|\mathcal{Y}|$ in the algorithm.(b) Case of training a single neural network for all classes: This is a special case in our problem setting which is similar to the objective in CBDM [2]. The error analysis for such a case will follow from arguments underlying gradient descent convergence properties for convex, smooth loss functions. However, in such a case we can only derive $E_{y}[ D_{KL}(P_0(.|y)||P_{0,\theta}(.|y))]$ and deriving a worst-case sampling error, a desideratum made possible by our formulation, will not be possible.  (c) Application to Federated Learning: If the question is regarding when does such a scenario arise where one needs to train separate score network for each class label, then Federated Learning is a classic example where our problem formulation will thrive. Consider the following scenario, each class label $y \in \mathcal{Y}$ is thought of as a client that holds private training data with variable number of training sample points. Individual Gradient Descent then represents local training of score network with global sharing of updated score network parameters while preserving the privacy of local client data. This allows fair learning and generalization among all classes and prevents overfitting (memorization) for class labels with low training data frequency. In Continual Learning, multiple score networks can also be thought as different subspaces over which training occurs and each subspace corresponding to a class label. This idea stems from ``Continual Learning in Low-rank Orthogonal Subspaces" Chaudhry et al NeuRIPS 2020. Mutual learning allows for transfer of knowledge among task ``Meta-learn" during training within these subspaces while preventing catastrophic forgetting. However, such a concept is a new research direction in itself, and one has to employ manifold optimization theory which is beyond the scope of our work.
>
> 3. How adding the regularization term ...
>
> Thank you for giving us a chance to respond to this interesting question. It is important to distinguish the empirical version of the objectives $\bar{\mathcal{L_reg}}^{n_y}(\theta_y,\theta_{-y}),\bar{\mathcal{L}}^{n_y}(\theta_y)$ over which training occurs and the population loss $\mathcal{L}^{y}(\theta_y)$ that determines the sampling (generalization) error. The Individual Gradient Descent is employed to find an $\epsilon-$ Nash Equilibrium of the  $|\mathcal{Y}|$-player game $<  \mathcal{Y},(\bar{\mathcal{L_reg}}^{n_y}(\theta_y,\theta_{-y})),(\Theta_y) >$ where the cost function is given by the empirical regularized objectives. When $\beta=0$ or as $\beta \to 0$, $\bar{\mathcal{L_reg}}^{n_y}(\theta_y,\theta_{-y})\approx \bar{\mathcal{L}}^{n_y}(\theta_y)$ and the algorithm finds $\hat{\theta}_y \in \arg\min \bar{\mathcal{L}}^{n_y}(\theta_y)$ (no mutual learning). This is the classical DDPM training paradigm.
>
> As present in Theorem 1, in such a case, the empirical training objective will be low, and the sampling error will be dominated by generalization gap which is inversely proportional to $n_y$. For tail classes that have low training data frequency, this generalization bound will be large resulting in large sampling error. One might feel this can be overcome by selecting a score network with more hidden neurons (large $m$); however, this increases the training error as shown in Theorem 1.
>     When $\beta$ is large, the mutual learning term enjoys more weight during training. Hence, there is a sweet spot of $\beta$, not too large, not too small (V shaped curve of Figure 1 (right) in our paper) where employing mutual learning is beneficial.
>
>  As we cannot post images or documents in the rebuttal phase, we refer to reviewers to Figure 2(left) in ``Fast Sampling of Diffusion Models with Exponential Integrator" Zhang et al 2022 arXiv for high-level intuition. We will ensure a high-level intuition for mutual learning applied to diffusion models is included in future revisions/ versions of our work.
>
> Diffusion models are trained to learn the score function $\nabla \log{p_t(x_t|x_0)}$ of the forward process which is then used in the reverse SDE to sample/ generate. As it can be seen from Figure 2 (left) the score model works well in the region where $p_t(x)$ is large but suffers from large error in the region where $p_t(x)$ is small. This observation can be explained by examining the training loss on Figure 2 (right). As explained in the paper mentioned above, since the training data is sampled from $p_t(x,y)$ in regions with a low $p_t(x,y)$ value, the learned score network is not expected to work well due to the lack of training data" Figure 2(left).
>
> As a consequence, to ensure $\bar{Y}_0$ is close to $x_0$, one need to make sure $\bar{Y}_t$ (as defined in the paper) stays in the high $p_t(x,y)$ region $\forall t \in [0,T]$. Since the tail classes have very few training data, their (tail classes) score do not lie in this high density region in Figure 2 (left). The mutual learning term aligns the score network for the tail classes with the high confidence scores of head classes (in the high noise region, in other words larger $t$) in a way such that the score network of the tail classes stays in the high probability region during sampling/ generation. The above reasoning also justifies the choice of weighting function $\omega(t)$. For $t\approx 0,  p_t \approx p_0   $, we wish the training objective would give more wait to fitting to data distribution while for larger $t\gg0$ the training give more weight to mutual learning objective. This implies $\omega(t)$ should be an increasing function of $t\in [0,T]$. This is employed in "Class-Balancing Diffusion Model" but lacked proper theoretical derivation and justification. In short, mutual learning/ regularization improves the generalization capability of the learned score networks by allowing knowledge transfer and preventing overfitting.
>
> 4. We took the code from ``Class Balancing Diffusion Model” Qin. et al 2023 and updated the training procedure according to individual gradient descent. Given the time and resource constraint, we were able to run CBDM for 100000 training steps and individual gradient descent for 10000 steps. To perform evaluation, we generate 500 total samples and make the comparison at the 10000 training step mark. We provide FID, IS across various parameter settings. We will include the link to the github repository for the code and images in future revisions, versions of our paper. We provide  tables summarizing the results.
>
> | Method         | FID     | IS          |
> |----------------|---------|-------------|
> | Vanilla, τ=0 (DDPM) | 423.06 | 1.23 (0.02) |
> | CBDM  | 419.82 | 1.22 (0.01) |
> | Mutual Learning, τ=0.1 | 409.50 | 1.22 (0.01) |
>
> ## Table 2: Different τ Values (lr=2e-4)
> | Method         | FID     | IS          |
> |----------------|---------|-------------|
> | τ=0.0, lr=2e-4 | 423.06 | 1.23 (0.02) |
> | τ=0.1, lr=2e-4 | 415.81 | 1.23 (0.02) |
> | τ=1.0, lr=2e-4 | 421.82 | 1.22 (0.01) |
>
> ## Table 3: Different lr Values (τ=0.1)
> | Method         | FID     | IS          |
> |----------------|---------|-------------|
> | τ=0.1, lr=2e-4 | 415.81 | 1.23 (0.02) |
> | τ=0.1, lr=1e-4 | 409.50 | 1.22 (0.01) |
> | τ=0.1, lr=1e-5 | 442.35 | 1.20 (0.01) |
>
> As seen from the above table, CBDM does better than the vanilla case of DDPM trained on individual class labels and mutual learning does better than the two. Mutual learning doing better than Vanilla case highlights the impact of mutual learning. We were not able to replicate the V-shaped curve as in Fig 1 (right) in our paper for real world datasets due to resource and time constraints.
>
> 5. We also performed to replicate case 2 in our supplementary Apeendix C. Class 0: mixture with means [-2,0][2,0] and sigma=1; class 1:single guassian with mean [0,0] and sigma=2).
> | Method         | FD (Head Class) | FD (Tail Class) |
> |----------------|-----------------|-----------------|
> | IGD (τ=0.0)    | 2.766 ± 0.534   | 1.984 ± 0.321   |
> | IGD (τ=0.001)  | 2.751 ± 0.517   | 1.996 ± 0.317   |
> | IGD (τ=0.01)   | 2.720 ± 0.470   | 1.988 ± 0.297   |
> | IGD (τ=0.1)    | 2.467 ± 0.467   | 1.965 ± 0.287   |
> | IGD (τ=1.0)    | 1.954 ± 0.403   | 2.021 ± 0.370   |
>
> We hope the reviewer finds our response satisfactory. Please let us know during the discussion phase if we can provide further clarification regarding our work, notably it's theoretical contributions, novelty, and significance.

---

> > ### Comment · Reviewer_HV1G · 2025-08-05
> >
> > I would like to thank the authors for the responses, which have properly addressed my questions. I don't have any further questions. I intend to keep my grade though, as I think this is solid theoretical work but with not enough empirical evidence, even though I appreciate the experiment added in the rebuttal. This opinion seem to also be shared amongst the other reviewers.

---

> ### Author Response · Authors · 2025-08-05
> **Continuation of Experiemental Results**
>
> We thank the reviewers for going through our rebuttal. We continued the experiments asked for by the reviewer for larger training and and evaluation steps and perform a detailed comparison with CBDM and various hyper-parameter choices. The Neural network Architecture employed is U-net as in CBDM paper. We took the code from ``Class Balancing Diffusion Model” Qin. et al 2023 and updated the training procedure according to individual gradient descent. Given the time and resource constraint, we were able to run CBDM for 100000 training steps and individual gradient descent for 50000 steps on CIFAR10LT. To perform the evaluation, we generate 1500 samples per class label as in CBDM paper and make the comparison at the 50000 training step mark. We provide FID, IS across various parameter settings. We will include the link to the github repository for the code and images in future revisions as well as versions of our paper. We provide tables summarizing the results below.
>
> **Table 1: Best FID Scores**
>
> | Method                | FID    | IS           |
> |-----------------------|--------|--------------|
> | IGD, τ=0, lr=2e-4     | 19.46  | 8.62 (0.15)  |
> | IGD, τ=0.1, lr=1e-4   | 17.63  | 8.70 (0.17)  |
> | CBDM, lr=2e-4         | 19.10  | 7.92 (0.18)  |
>
> **Table 2: Different τ Values (lr=2e-4)**
>
> | Method                  | FID    | IS           |
> |-------------------------|--------|--------------|
> | IGD, τ=0.0, lr=2e-4     | 19.46  | 8.62 (0.15)  |
> | IGD, τ=0.1, lr=2e-4     | 19.23  | 8.53 (0.21)  |
> | IGD, τ=1.0, lr=2e-4     | 19.96  | 8.50 (0.21)  |
>
> **Table 3: Different lr Values (τ=0.1)**
>
> | Method                  | FID    | IS           |
> |-------------------------|--------|--------------|
> | IGD, τ=0.1, lr=2e-4     | 19.23  | 8.53 (0.21)  |
> | IGD, τ=0.1, lr=1e-4     | 17.63  | 8.70 (0.17)  |
> | IGD, τ=0.1, lr=1e-5     | 22.25  | 8.33 (0.15)  |
>
>
>
> As seen from the above tables, CBDM does better than the vanilla case of DDPM trained on individual class labels and mutual learning does better than both of them. Mutual learning doing better than the Vanilla case highlights the impact of mutual learning. We hope that these experiments clarify the queries raised by the reviewer.
>
> We would also like to respectfully highlight that the paper’s main contribution is of a theoretical nature, which is why we have chosen “theory” as the primary area. We hope the experiments provided are sufficient for a reader to build confidence in our theoretical findings.

---

### Official Review · Reviewer_9nrw · 2025-07-02

**Clarity:** 2
**Significance:** 2
**Originality:** 3
**Rating:** 4
**Confidence:** 1

**Summary:**

This paper proposes a long-tailed learning framework via diffusion models. This paper introduces a regularized training objective that combines the standard diffusion loss with a mutual learning term, addressing the issue of poor generation performance for tail classes under long-tailed data distributions in diffusion models. The paper analyzes the convergence and generalization performance of this method from a theoretical perspective, and verifies its effectiveness through experiments.

**Questions:**

See above weakness. Since I am not familar with the theoretical part in this paper, authors need to address the above weakness well: 1) Comparsions 2) hyperparameters choosen 3) more indicators 4) More visual results.

**Ethical Concerns:**

["NO or VERY MINOR ethics concerns only"]

**Final Justification:**

Since I am not familar with the theoretical part in this paper, although the authors have addressed partial of my concerns, but other reviewers ' question seems not to be solved now. So I would like to keep the score.

**Limitations:**

Yes

**Quality:**

3

**Strengths And Weaknesses:**

Strength：

1.This paper combines deep mutual learning and diffusion models and proposes a regularized training objective that balances the generation performance of all categories.

2.The paper gives an upper bound of Nash gap of the score network obtained from the algorithm and worst case error bound. Furthermore, the theoretical analysis is comprehensive and in-depth.

Weakness:

1.This paper should give sufficient comparisons of quantitative and qualitative experimental results.

2.This paper points out that the choice of hyperparameters has a significant impact on model performance, but there is a lack of in-depth discussion on the impact of these hyperparameters on performance results

3.This paper should include results of more indicators, such as Recall and IS used in CBDM [1].

4.This paper gives enough formulas, but lacks necessary visualization, which creates obstacles to understanding the methods and effects.

[1] Yiming Qin, et al. “Class-balancing diffusion models.” CVPR 2023

---

> ### Author Rebuttal · Authors · 2025-07-30
>
> Thank you very much for taking the time to review our paper and provide valuable feedback. We appreciate your constructive comments, which have helped us improve the quality of our work. Your insights and suggestions were truly helpful in strengthening our submission.
>
> Thank you again for your effort and contribution to the review process. Please find our response to your questions
>
>
> 1.This paper should give sufficient comparisons of quantitative and qualitative experimental results.
>
> 2.This paper points out that the choice of hyperparameters has a significant impact on model performance, but there is a lack of in-depth discussion on the impact of these hyperparameters on performance results
>
> In Corollary 1, we provide the hyper-parameters as a function of the network parameters, training iterations, sampling time duration, dimension of data, etc. to achieve and $\mathcal{O}(\epsilon+1) $ sampling error.  Although our analysis is focused on two layer random ReLUs,  We strongly believe our approach can be extended to deeper neural network architectures with more hidden layers in line with  ``Evaluating the design space of diffusion-based generative models" Wang et al NeuRIPS 2024. The objective functions satisfy nice properties when the score function is parametrized through a random feature model, enabling the computation of an $\epsilon-$ Nash Equilibrium. For deeper neural networks, we present a high level approach:
>
> (a) Prove Semi-Smoothness of the regularized objective function: Observe that $\bar{\mathcal{L_reg}}^{n_y}(\theta_y,\theta_{-y})=\bar{\mathcal{L}}^{n_y}(\theta_y)+ \beta \bar{\mathcal{L_mut}}^{n_y}(\theta_y,\theta_{-y})$. Lemma 9 [1] proves the Semi-smoothness of   $\bar{\mathcal{L}}^{n_y}(\theta_y)$ with high probability. This can also be proved for $\bar{\mathcal{L_mut}}^{n_y}(\theta_y,\theta_{-y})$ in line with Theorem 3 in [2].
>
> (b) PL type inequality: Then next step would be to derive a PL like inequality as Lemma 2 in [1] with high probability. As the mutual learning learning term takes a quadratic form, we believe after some work this property to continue to hold for deeper neural networks.
>
> (c)  Lipschitzness of mutual learning term: With high probability the gradient of norm $\bar{\mathcal{L_reg}}^{n_y}(\theta_y,\theta_{-y})$ w.r.t to $\theta_y$ is bounded which implies $\bar{\mathcal{L_mut}}^{n_y}(\theta_y,\theta_{-y})$ is lipschitz in $\theta_y$.
>
> Combining the above results will allow one to extend our framework to deeper neural networks. Our work forms the stepping stone and motivation for future work in Generative AI, Deep Learning and Optimization to be reformulated as games that would allow for their expansion to Federated, distributed setting and much more.
>
>         [1] Wang et al NeuRIPS 2024 ``Evaluating the design space of diffusion-based generative models"
>
>         [2] Zhu et al ``A Convergence Theory for Deep Learning via Over-Parametrization"
>
>
>
> 3.This paper should include results of more indicators, such as Recall and IS used in CBDM [1].
>
> 4.This paper gives enough formulas, but lacks necessary visualization, which creates obstacles to understanding the methods and effects.
>
> As we cannot post images or documents in the rebuttal phase, we refer the reviewers to Figure 2(left) in ``Fast Sampling of Diffusion Models with Exponential Integrator"  Zhang et al 2022 arXiv for high-level intuition for now. We will ensure a high-level intuition and diagram for mutual learning applied to diffusion models is included in future revisions/ versions of our work.
>     Intuition: Diffusion models are trained to learn the score function $\nabla \log{p_t(x_t|x_0)}$ of the forward process which is then used in the reverse SDE to sample/ generate. As it can be seen from Figure 2 (left) [4] the score model works well in the region where $p_t(x)$ is large but suffers from large error in the region where $p_t(x)$ is small. This observation can be explained by examining the training loss on Figure 2(right). As explained in the paper mentioned above, since the training data is sampled from $p_t(x,y)$ in regions with a low $p_t(x,y)$ value, the learned score network is not expected to work well due to the lack of training data" Figure 2(left).
>      As a consequence, to ensure $\bar{Y}_0$ is close to $x_0$, one need to make sure $\bar{Y}_t$ (as defined in the paper) stays in the high $p_t(x,y)$ region $\forall t \in [0,T]$ during the generation/sampling process. Since the tail classes have very few training data, their (tail classes) score do not lie in this high-density region in Figure 2 (left). The mutual learning term aligns the score network for the tail classes with the high confidence scores of head classes (in the high noise region, in other words larger $t$) in a way such that the score network of the tail classes stays in the high probability region during sampling/ generation. The above reasoning also justifies the choice of weighting function $\omega(t)$. For $t\approx 0,  p_t \approx p_0
>      $, we wish the training objective would give more wait to fitting to data distribution while for larger $t\gg0$ the training give more weight to mutual learning objective. This implies $\omega(t)$ should be an increasing function of $t\in [0,T]$. This is employed in "Class-Balancing Diffusion Model" but lacked proper derivation and justification.
>
>    [4] Fast Sampling of Diffusion Models with Exponential Integrator"  Zhang et al 2022 arXiv
>
> 5. We took the code from ``Class Balancing Diffusion Model” Qin. et al 2023 and updated the training procedure according to individual gradient descent. Given the time and resource constraint, we were able to run CBDM for 100000 training steps and individual gradient descent for 10000 steps. To perform evaluation, we generate 500 total samples and make the comparison at the 10000 training step mark. We provide FID, IS across various parameter settings. We will include the link to the github repository for the code and images in future revisions, versions of our paper. We provide  tables summarizing the results.
>
> | Method         | FID     | IS          |
> |----------------|---------|-------------|
> | Vanilla, τ=0 (DDPM) | 423.06 | 1.23 (0.02) |
> | CBDM  | 419.82 | 1.22 (0.01) |
> | Mutual Learning, τ=0.1 | 409.50 | 1.22 (0.01) |
>
> ## Table 2: Different τ Values (lr=2e-4)
> | Method         | FID     | IS          |
> |----------------|---------|-------------|
> | τ=0.0, lr=2e-4 | 423.06 | 1.23 (0.02) |
> | τ=0.1, lr=2e-4 | 415.81 | 1.23 (0.02) |
> | τ=1.0, lr=2e-4 | 421.82 | 1.22 (0.01) |
>
> ## Table 3: Different lr Values (τ=0.1)
> | Method         | FID     | IS          |
> |----------------|---------|-------------|
> | τ=0.1, lr=2e-4 | 415.81 | 1.23 (0.02) |
> | τ=0.1, lr=1e-4 | 409.50 | 1.22 (0.01) |
> | τ=0.1, lr=1e-5 | 442.35 | 1.20 (0.01) |
>
> As seen from the above table, CBDM does better than the vanilla case of DDPM trained on individual class labels and mutual learning does better than the two. Mutual learning doing better than Vanilla case highlights the impact of mutual learning. We were not able to replicate the V-shaped curve as in Fig 1 (right) in our paper for real world datasets due to resource and time constraints.
>
> 6. We also performed to replicate case 2 in our supplementary Apeendix C. Class 0: mixture with means [-2,0][2,0] and sigma=1; class 1:single guassian with mean [0,0] and sigma=2).
> | Method         | FD (Head Class) | FD (Tail Class) |
> |----------------|-----------------|-----------------|
> | IGD (τ=0.0)    | 2.766 ± 0.534   | 1.984 ± 0.321   |
> | IGD (τ=0.001)  | 2.751 ± 0.517   | 1.996 ± 0.317   |
> | IGD (τ=0.01)   | 2.720 ± 0.470   | 1.988 ± 0.297   |
> | IGD (τ=0.1)    | 2.467 ± 0.467   | 1.965 ± 0.287   |
> | IGD (τ=1.0)    | 1.954 ± 0.403   | 2.021 ± 0.370   |
>
> We hope the reviewer finds our response satisfactory. Please let us know during the discussion phase if we can provide further clarification regarding our work, notably it's theoretical contributions, novelty, and significance.

---

### Official Review · Reviewer_aRKS · 2025-07-03

**Clarity:** 2
**Significance:** 2
**Originality:** 3
**Rating:** 4
**Confidence:** 2

**Summary:**

This paper tackles the underexplored challenge of training **diffusion models on long-tailed datasets**, where many classes have very few examples. The authors propose a **game-theoretic framework** that treats training diffusion models as a multi-player game over conditional score networks, using **Deep Mutual Learning (DML)** and a **regularized objective**. The solution is sought at a **Nash equilibrium**, and the authors derive convergence guarantees for gradient descent toward such an equilibrium. Theoretical bounds on **worst-case sampling error** across all classes are provided, along with architectural and training details using ReLU networks.

**Questions:**

- How would this approach scale to larger architectures?
- How does this approach experimentally compare to other long-tailed methods?
- How is the quality of the samples? Would the quality still be maintained for larger generative tasks?
- Could there be a diagram or primer on mutual learning?

- please fix Line 108 typo in section header

**Ethical Concerns:**

["NO or VERY MINOR ethics concerns only"]

**Final Justification:**

Significant theoretical contributions and adequate experiments. Theoretical contributions could go further however.

**Limitations:**

Yes

**Quality:**

3

**Strengths And Weaknesses:**

Strengths:
- High importance: long-tailed class imbalance is a critical bottleneck in real-world generative applications
- Novel game-theoretic formulation with Nash equilibrium applied to conditional diffusion
- Theoretical guarantees (non-asymptotic convergence + worst-case bounds)


Weaknesses:
- experiments only show FID to measure the quality of the samples
- Could benefit from more high-level intuition—especially why mutual learning is the right approach/ what mutual learning even is
- Small architectures used in experiments
- no experimental comparison with other long-tailed methods

---

> ### Author Rebuttal · Authors · 2025-07-30
>
> Thank you very much for taking the time to review our paper and provide valuable feedback. We appreciate your constructive comments, which have helped us improve the quality of our work. Your insights and suggestions were truly helpful in strengthening our submission.
>
> Thank you again for your effort and contribution to the review process. Please find our response to your questions..
>
> 1.  ...scale to larger architectures?
>
> On the Theory side: We strongly believe our approach can be extended to deeper neural network architectures with more hidden layers in line with  ``Evaluating the design space of diffusion-based generative models" Wang et al NeuRIPS 2024. The objective functions satisfy nice properties when the score function is parametrized through a random feature model, enabling the computation of an $\epsilon-$ Nash Equilibrium. For deeper neural networks, we present a high level approach:
>
> (a) Prove Semi-Smoothness of the regularized objective function: Observe that $\bar{\mathcal{L_reg}}^{n_y}(\theta_y,\theta_{-y})=\bar{\mathcal{L}}^{n_y}(\theta_y)+ \beta \bar{\mathcal{L_mut}}^{n_y}(\theta_y,\theta_{-y})$. Lemma 9 [1] proves the Semi-smoothness of   $\bar{\mathcal{L}}^{n_y}(\theta_y)$ with high probability. This can also be proved for $\bar{\mathcal{L_mut}}^{n_y}(\theta_y,\theta_{-y})$ in line with Theorem 3 in [2].
>
> (b) PL type inequality: Then next step would be to derive a PL like inequality as Lemma 2 in [1] with high probability. As the mutual learning learning term takes a quadratic form, we believe after some work this property continues to hold for deeper neural networks.
>
> (c)  Lipschitzness of mutual learning term: With high probability the gradient of norm $\bar{\mathcal{L_reg}}^{n_y}(\theta_y,\theta_{-y})$ w.r.t to $\theta_y$ is bounded which implies $\bar{\mathcal{L_mut}}^{n_y}(\theta_y,\theta_{-y})$ is lipschitz in $\theta_y$.
>
> Combining the above results will allow one to extend our framework to deeper neural networks.
>
> 2. ..experimentally compare to other long-tailed methods?
>
> There are various methods that are used to address the problem of Deep learning task in the presence of long-tailed distribution. Most of the long-tailed methods employed in machine learning task are yet to be extended to Diffusion models. In our work, we take a more theoretical approach to extend mutual learning to the problem of Diffusion generative modeling in the presence of tong-tailed distributions. As we take a more theoretical approach, we provide comparison with empirical data distributions such as non-uniform mixture of gaussian where each Gaussian distribution represents a class label.  These empirical studies allow us to verify our theoretical findings and understand scenarios when mutual learning is useful and when it is not. We understand your concern and have performed experiments to address these in the supplementary section. Performing statistical analysis on maximum of a sequence of random variable whose distribution are unknown is challenging. To circumvent this, in Appendix C, we directly compute the KL divergence between the distribution characterized by the learned score network and the ground truth distribution as in Section 4 in [3]. This allows for direct comparison among class labels and within class labels. As you will observe in Supplementary Material Appendix C, we have provided more empirical studies to verify our theoretical findings. In case 1 (Appendix C), the performance of the head class goes down when mutual learning is used.  However, the worst case KL divergence is still lower for mutual learning. This is because the head class contains enough training samples to learn a good score network and achieves no performance increase from transfer of knowledge from the tail class. Furthermore, in case 2 when the relative position of the class labels in the embedding space provides additional knowledge (information); mutual learning is beneficial for all classes.    We observe improved performance in both the classes (Figure 3 in Appendix C) when mutual learning is employed vs when it is not employed. This raises interesting questions in representation learning of the input class label data into embedding spaces where the relative positions of classes enable efficient learning through mutual learning. Our work opens the door to multiple future research directions.
>
>
>
> 3. ..Could there be a diagram or primer on mutual learning?
>
> As we cannot post images or documents in the rebuttal phase, we refer the reviewers to Figure 2(left) in ``Fast Sampling of Diffusion Models with Exponential Integrator"  Zhang et al 2022 arXiv for a  high level intuition for now. We will ensure a high level intuition and diagram  for mutual learning applied to diffusion models is included in future revisions/ versions of our work.
>     Intuition: Diffusion models are trained to learn the score function $\nabla \log{p_t(x_t|x_0)}$ of the forward process which is then used in the reverse SDE to sample/ generate. As it can be seen from Figure 2 (left) [4] the score model works well in the region where $p_t(x)$ is large but suffers from large error in the region where $p_t(x)$ is small. This observation can be explained by examining the training loss on Figure 2(right). As explained in the above mentioned paper, since the training data is sampled from $p_t(x,y)$ in regions with a low $p_t(x,y)$ value, the learned score network is not expected to work well due to the lack of training data" Figure 2(left).
>      As a consequence, to ensure $\bar{Y}_0$ is close to $x_0$, one need to make sure $\bar{Y}_t$ (as defined in the paper) stays in the high $p_t(x,y)$ region $\forall t \in [0,T]$ during the generation/sampling process. Since the tail classes have very few training data, their (tail classes) score do not lie in this high density region in Figure 2 (left). The mutual learning term aligns the score network for the tail classes with the high confidence scores of head classes (in the high noise region, in other words larger $t$) in a way such that the score network of the tail classes stay in the high probability region during sampling/ generation. The above reasoning also justifies the choice of weighting function $\omega(t)$. For $t\approx 0,  p_t \approx p_0
>      $, we wish the training objective give more wait to fitting to data distribution while for larger $t\gg0$ the training give more weight to mutual learning objective. This implies $\omega(t)$ should be an increasing function of $t\in [0,T]$. This is employed in "Class-Balancing Diffusion Model", but lacked  proper derivation and justification.
>
> 4. Please note that we provide the limitations of our work in Section 7. Regarding the potential societal consequences, given the theoretical nature of our work we do not foresee any direct implications. However, mitigating the adverse effects of the long-tailed phenomena could lead to more robust, secure, and fair generative models.
>
>         [1] Wang et al. NeuRIPS 2024 ``Evaluating the design space of diffusion-based generative models"
>
>         [2] Zhu et al. ``A Convergence Theory for Deep Learning via Over-Parametrization"
>
>         [3] Li et al. NeuRIPS 2023 ``On the generalization properties of diffusion models"
>
>         [4] Zhang et al. 2022 arXiv ``Fast Sampling of Diffusion Models with Exponential Integrator"
>
>
>
>
>
> 4. We took the code from ``Class Balancing Diffusion Model” Qin. et al 2023 and updated the training procedure according to individual gradient descent. Given the time and resource constraint, we were able to run CBDM for 100000 training steps and individual gradient descent for 10000 steps. To perform evaluation, we generate 500 total samples and make the comparison at the 10000 training step mark. We provide FID, IS across various parameter settings. We will include the link to the github repository for the code and images in future revisions, versions of our paper. We provide  tables summarizing the results.
>
> | Method         | FID     | IS          |
> |----------------|---------|-------------|
> | Vanilla, τ=0 (DDPM) | 423.06 | 1.23 (0.02) |
> | CBDM  | 419.82 | 1.22 (0.01) |
> | Mutual Learning, τ=0.1 | 409.50 | 1.22 (0.01) |
>
> ## Table 2: Different τ Values (lr=2e-4)
> | Method         | FID     | IS          |
> |----------------|---------|-------------|
> | τ=0.0, lr=2e-4 | 423.06 | 1.23 (0.02) |
> | τ=0.1, lr=2e-4 | 415.81 | 1.23 (0.02) |
> | τ=1.0, lr=2e-4 | 421.82 | 1.22 (0.01) |
>
> ## Table 3: Different lr Values (τ=0.1)
> | Method         | FID     | IS          |
> |----------------|---------|-------------|
> | τ=0.1, lr=2e-4 | 415.81 | 1.23 (0.02) |
> | τ=0.1, lr=1e-4 | 409.50 | 1.22 (0.01) |
> | τ=0.1, lr=1e-5 | 442.35 | 1.20 (0.01) |
>
> As seen from the above table, CBDM does better than the vanilla case of DDPM trained on individual class labels and mutual learning does better than the two. Mutual learning doing better than Vanilla case highlights the impact of mutual learning. We were not able to replicate the V-shaped curve as in Fig 1 (right) in our paper for real world datasets due to resource and time constraints.
>
> 5. We also performed to replicate case 2 in our supplementary Apeendix C. Class 0: mixture with means [-2,0][2,0] and sigma=1; class 1:single guassian with mean [0,0] and sigma=2).
> | Method         | FD (Head Class) | FD (Tail Class) |
> |----------------|-----------------|-----------------|
> | IGD (τ=0.0)    | 2.766 ± 0.534   | 1.984 ± 0.321   |
> | IGD (τ=0.001)  | 2.751 ± 0.517   | 1.996 ± 0.317   |
> | IGD (τ=0.01)   | 2.720 ± 0.470   | 1.988 ± 0.297   |
> | IGD (τ=0.1)    | 2.467 ± 0.467   | 1.965 ± 0.287   |
> | IGD (τ=1.0)    | 1.954 ± 0.403   | 2.021 ± 0.370   |
>
> We hope the reviewer finds our response satisfactory. Please let us know during the discussion phase if we can provide further clarification regarding our work, notably it's theoretical contributions, novelty, and significance.

---

> > ### Comment · Reviewer_aRKS · 2025-08-06
> >
> > The authors have adequately responded to certain points. However, this reviewer still has reservations regarding:
> >
> > 1. ...scale to larger architectures?
> >
> > While the authors proposed approach is believable, the reviewer's intuition is that many theoretical results for small networks are quite difficult to extend to deeper networks. At present, this reviewer does not believe the response is adequate, given this uncertainty. Would it be possible to point to specific proof techniques?
> >
> > 2. ..experimentally compare to other long-tailed methods?
> >
> > The authors do not provide comparisons to other long-tailed methods, even those that are not diffusion-based. Why can we not compare to these non-diffusion methods?
> >
> > 4. societal impacts
> >
> > Typically a good acknowledgement of societal consequences for a theoretical work should be along the lines of:
> >
> > > We do not know of particular risks or negative impacts of this work beyond risks of generative models in general.
> >
> > I would suggest the authors amend their societal consequences section to have less of a techno-chauvinist tone about their results.
> >
> > Furthermore the following was unaddressed:
> >
> > > How is the quality of the samples? Would the quality still be maintained for *larger generative tasks*?

---

> ### Author Response · Authors · 2025-08-07
> **Response to Reviewer questions**
>
> 1. While the authors proposed approach is believable....
>
> Although extensions to deeper networks are difficult, it is not impossible. Here we cite two works [1], [2] and [3]. [1] proves why stochastic gradient descent (SGD) can find global minima on the training objective of Deep Neural Networks (DNN) in polynomial time using only two assumptions the inputs are non-degenerate and the network is over-parameterized i.e.  the network width is sufficiently large: polynomial in L, the number of layers and in n, the number of samples. [2] extends the works of [1] to determine the training complexity for diffusion models and determine the generalization error of sampling with DNNs (Corollary 1 in [2]). Please allow us to elaborate our approach for Mutual Learning for DNNs. [3] provides the generalization bound using Rademacher Complexity for DNNs with ReLU activation function.
>
> (a) Prove Semi-Smoothness of the regularized objective function: Observe that $\bar{\mathcal{L_reg}}^{n_y}(\theta_y,\theta_{-y})=\bar{\mathcal{L}}^{n_y}(\theta_y)+ \beta \bar{\mathcal{L_mut}}^{n_y}(\theta_y,\theta_{-y})$. Lemma 9 [2] proves the Semi-smoothness of $\bar{\mathcal{L}}^{n_y}(\theta_y)$ with high probability.  The Semi-Smoothness of  $\bar{\mathcal{L_mut}}^{n_y}(\theta_y,\theta_{-y})$ follows directly from  Theorem 3 in [1]. The only thing that one needs to compute how the various constants depend on the hyper-parameters such network with, depth, dimensions, etc.
>
> (b)  Lipschitzness of mutual learning term: With high probability the gradient of norm $\bar{\mathcal{L_reg}}^{n_y}(\theta_y,\theta_{-y})$ w.r.t to $\theta_y$ is bounded which implies $\bar{\mathcal{L_mut}}^{n_y}(\theta_y,\theta_{-y})$ is lipschitz in $\theta_y$.
>
>
> (c) PL type inequality: The next step would be to derive a PL like inequality as Theorem 3 in [1] and Lemma 1 (Appendix D.1 in [2]) with high probability. Step (c) is the challenging step.Lemma 1 in [2] provides a proof sketch for the case without mutual learning.  Even though $\bar{\mathcal{L}}^{n_y}(\theta_y)$ and $\bar{\mathcal{L_mut}}^{n_y}(\theta_y,\theta_{-y})$ individually satisfy this PL like inequality, their sum $\bar{\mathcal{L_reg}}^{n_y}(\theta_y,\theta_{-y})=\bar{\mathcal{L}}^{n_y}(\theta_y)+ \beta \bar{\mathcal{L_mut}}^{n_y}(\theta_y,\theta_{-y})$ may or may not. This is a non-trivial issue and needs to be rigorously proven. We leave this as a conjecture allowing for future works to pursue.
>
> Combining the above results will allow one to extend our framework to deeper neural networks.
>
>
>         [1]  Zhu et al ``A Convergence Theory for Deep Learning via Over-Parametrization"
>
>         [2] Wang et al NeuRIPS 2024 ``Evaluating the design space of diffusion-based generative models"
>
>         [3] Lan V. Truong ``On Rademacher Complexity-based Generalization Bounds for Deep Learning"
>
> 2. quality of sample and long-tailed methods ...
> We took the code from ``Class Balancing Diffusion Model” Qin. et al 2023 (closest established neighbour to our work) and updated the training procedure according to individual gradient descent. The Neural network Architecture employed is U-net as in the CBDM paper.  To perform the evaluation, we generate 1500 samples per class label as in CBDM paper and make the comparison at the 50000 training step mark for CIFAR10LT dataset. We provide FID and IS across various parameter settings. FID measures the quality of samples. A low FID score represents better sample quality.
>
> **Table 1: Best FID Scores**
>
> | Method                | FID    | IS           |
> |-----------------------|--------|--------------|
> | IGD, τ=0, lr=2e-4     | 19.46  | 8.62 (0.15)  |
> | IGD, τ=0.1, lr=1e-4   | 17.63  | 8.70 (0.17)  |
> | CBDM, lr=2e-4         | 19.10  | 7.92 (0.18)  |
>
> **Table 2: Different τ Values (lr=2e-4)**
>
> | Method                  | FID    | IS           |
> |-------------------------|--------|--------------|
> | IGD, τ=0.0, lr=2e-4     | 19.46  | 8.62 (0.15)  |
> | IGD, τ=0.1, lr=2e-4     | 19.23  | 8.53 (0.21)  |
> | IGD, τ=1.0, lr=2e-4     | 19.96  | 8.50 (0.21)  |
>
> **Table 3: Different lr Values (τ=0.1)**
>
> | Method                  | FID    | IS           |
> |-------------------------|--------|--------------|
> | IGD, τ=0.1, lr=2e-4     | 19.23  | 8.53 (0.21)  |
> | IGD, τ=0.1, lr=1e-4     | 17.63  | 8.70 (0.17)  |
> | IGD, τ=0.1, lr=1e-5     | 22.25  | 8.33 (0.15)  |
>
> 3. Societal ...  W
>
> We apologize for the language used under that section. We will amend it according to your suggestion in future revisions.
>
> An experimental comparison of mutual learning with other long tailed methods for classification task is provided in [4]. We would like to also respectfully highlight that the paper’s main contribution is of a theoretical nature, which is why we have chosen “theory” as the primary area.  We hope the experiments provided are sufficient for a reader to build confidence in our theoretical findings.
>
> [4] Park et al Mutual Learning for Long-Tailed Recognition WACV 2023

---

> > ### Author Response · Authors · 2025-08-07
> > **Additional Points**
> >
> > As few points slipped our mind while writing the above response. We would like to add the following points
> >
> > quality of sample and long-tailed methods ...
> >
> > As seen from the table the Mutual Learning performs better than CBDM in terms of generation quality and IS (larger the better).
> >
> > A theoretical comparison among various non-diffusion methods would itself be a research direction to pursue owing to the domain specific performance metric which motivated a unified performance metric across various machine learning, deep learning, generative modelling to compare various methods.

---

> > > ### Comment · Reviewer_aRKS · 2025-08-08
> > >
> > > This reviewer would like to thank the authors for their responses. This reviewer believes the responses are adequate. Furthermore, as this is a theory paper, the experiments seem sufficient.
> > >
> > > The theoretical results are significant, though could be extended for deeper networks. The reviewer will raise the score to a borderline accept.

---

### Note · Authors · 2025-08-12

Dear AC and reviewers,

We thank the AC for managing the review process. We thank the reviewers from the bottom of our heart for providing constructive feedback and engaging in a fruitful discussion that enabled us to strengthen our work.

  1.  While our theoretical contributions have been acknowledged by the reviewers, some concerns centered on the number of experiments were raised. In response to the reviewer requests, we conducted substantial additional experiments. As reported during the rebuttal and discussion phases, these experiments include evaluations on real-world datasets (e.g., CIFAR10-LT),  comparison with CBDM, and tests on synthetic benchmarks (e.g., Gaussian Mixture Models) with varying hyper-parameters such as regularizing parameter, learning rate.
  2.  Across various settings, we highlight that these results consistently align with and reinforce our theoretical predictions. These supplemental results along with the corresponding code will be integrated into the future versions within the main text and the appendix to ensure clarity, reproducibility, and accessibility. We believe that the combination of rigorous theory and well-targeted empirical validation meets the intended scope of the Theory track and addresses the key concerns of the reviewers.
  3.  We have chosen theory as the submission track for our paper as our contribution is primarily theoretical. The experiments—both those presented in the original submission and the additional ones reported in response to reviewer requests—are only intended to verify and support the main theoretical contributions and we believe they have done so.

We appreciate your consideration in the final decision.



Warm Regards,

Anonymous Authors

---

### Decision · Program_Chairs · 2025-09-17

**Decision:**

Accept (poster)

**Comment:**

This work considers diffusion models for long-tail distributions through a game-theoretic approach. Overall the method is quite interesting with solid empirical performance. Through the discussion phase, the authors have also addressed most concerns raised by the reviewers. Hence, the meta-reviewer recommends this work for acceptance.